# Lie Algebra Canonicalization: Equivariant Neural Operators under arbitrary Lie Groups

**Zakhar Shumaylov**[*,1]**, Peter Zaika**[*,1]**, James Rowbottom**[1]**, Ferdia Sherry**[1]**,
**Melanie Weber**[2]**, Carola-Bibiane Schönlieb**[1]

[1]University of Cambridge, [2]Harvard University,
[*]Equal contributions

## Abstract

The quest for robust and generalizable machine learning models has driven recent interest in exploiting symmetries through equivariant neural networks. In the context of PDE solvers, recent works have shown that Lie point symmetries can be a useful inductive bias for Physics-Informed Neural Networks (PINNs) through data and loss augmentation. Despite this, directly enforcing equivariance within the model architecture for these problems remains elusive. This is because many PDEs admit non-compact symmetry groups, oftentimes not studied beyond their infinitesimal generators, making them incompatible with most existing equivariant architectures. In this work, we propose Lie aLgebrA Canonicalization (LieLAC), a novel approach that exploits only the action of infinitesimal generators of the symmetry group, circumventing the need for knowledge of the full group structure. To achieve this, we address existing theoretical issues in the canonicalization literature, establishing connections with frame averaging in the case of continuous non-compact groups. Operating within the framework of canonicalization, LieLAC can easily be integrated with unconstrained pre-trained models, transforming inputs to a canonical form before feeding them into the existing model, effectively aligning the input for model inference according to allowed symmetries. LieLAC utilizes standard Lie group descent schemes, achieving equivariance in pre-trained models. Finally, we showcase LieLAC's efficacy on tasks of invariant image classification and Lie point symmetry equivariant neural PDE solvers using pre-trained models.

## 1 Introduction

The deep learning revolution of the 2010s has demonstrated the remarkable effectiveness of deep learning (DL) models in various domains, from text generation to computer vision. This widespread success has led to the adoption of DL in numerous scientific fields, including biology (Abramson et al., 2024), physics (Merchant et al., 2023) and engineering (Degrave et al., 2022). The training of such models often requires access to large amounts of high-quality data and computing resources, which can be challenging to obtain in practice. However, many tasks in these domains involve structured data, including data with symmetries induced by fundamental physical principles. *Physics-informed neural networks* (PINNs) seek to leverage such structure to mitigate the aforementioned challenges.

DL models have been observed to significantly benefit from the inductive bias of symmetries. Notable examples include translation invariance in imaging data and time-series, translation and rotation symmetries of molecules, resulting in position and momentum conservation. Incorporating these symmetries into the design of neural networks can enhance their performance and generalization capabilities. This has been analyzed empirically (Esteves et al., 2023; Wang et al., 2020) and through the lens of learning-theoretic trade-offs (Mei et al., 2021; Bietti et al., 2021; Kiani et al., 2024b).

Prior work on equivariant neural networks focuses on "simple" groups, resulting in frameworks that are often not rich enough to encode the complex geometric structure found in scientific applications. For example, the heat equation in Appendix D.3 was recently shown by Koval & Popovych (2023) to admit a group of point symmetries $\mathrm{SL}(2, \mathbb{R}) \ltimes_\phi \mathrm{H}(1, \mathbb{R})$, which is not only non-compact, it also does not act in a representation on the underlying jet space. This renders most of the currently available equivariant architectures unusable. To overcome this, we introduce a novel framework that is able to encode a wider range of symmetries, not requiring a priori knowledge of the full group structure. This flexibility allows for integration with existing models, notably pre-trained models, with the potential of leveraging the benefits of geometric inductive biases beyond classical equivariant architectures.

## 1.1 EQUIVARIANCE IN NEURAL NETWORKS

Incorporating group symmetries inside neural networks, in the form of *equivariance*, is one of the fundamental tasks in *geometric deep learning* (Bronstein et al., 2021). The translational equivariance of convolutional neural networks (CNNs) (Fukushima & Miyake, 1982; LeCun et al., 1989) has been suggested as an important reason for their successes in tasks such as image classification (Krizhevsky et al., 2012; He et al., 2016). Because datasets may have symmetries more intricate than just translations, the past decade has seen a push towards generalizing this approach to allow for other (Lie group) symmetries, starting from works on roto-translationally equivariant CNNs (Dieleman et al., 2015; 2016; Cohen & Welling, 2016), to more recent methods, efficiently utilizing knowledge of the Lie algebra for more general groups (Kumar et al., 2024; MacDonald et al., 2022; Finzi et al., 2020a; McNeela, 2023) and computationally efficient methods (Mironenco & Forré, 2024). Notably, convolution-based approaches to equivariance satisfy a type of local equivariance too (Weiler & Cesa, 2019): with compactly supported filters, if features in the data are sufficiently separated, these networks are equivariant to transformations localized to the features. Despite all of the work in this direction, construction of efficient and accurate equivariant architectures is not simple, see Appendix A.

More recently, there has been a drive towards publicly available foundational models (Bommasani et al., 2021) for various tasks, pretrained on massive datasets. These models, however, are often not equivariant. For example, Alphafold 3 (Abramson et al., 2024) no longer utilises equivariant architectures, unlike the initial version (Jumper et al., 2021). For this reason a direction of research has been investigating the possibility of utilizing the generalization offered by equivariance, while being able to use pre-trained models. The main two approaches to achieve this are *canonicalization*[1] Kaba et al. (2023); Mondal et al. (2023) and *frame averaging* Puny et al. (2022); Basu et al. (2023) (also referred to as symmetrization). However, there still remain a number of open theoretical questions regarding the possibility of continuous canonicalization (Dym et al., 2024), connections between frame averaging and canonicalizations Ma et al. (2024) and constructibility of frames Dym et al. (2024); Puny et al. (2022).

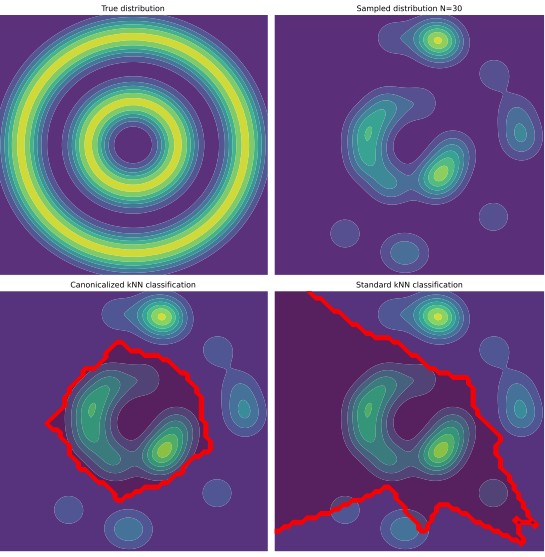

Figure 1: Effect of canonicalization on decision boundaries in $k$-NN classification for separating the inner and the outer rings Section 4.1.

## 1.2 PHYSICS-INFORMED MACHINE LEARNING

In a separate direction, deep neural networks are being increasingly used in scientific computing, particularly in simulating physical systems modeled by partial differential equations (PDEs). This surge in interest is largely driven by limitations of traditional numerical solvers, as they often struggle with demanding scenarios, such as those involving non-linear PDEs on complex domains necessitating fine discretization, or high-dimensional problems where the curse of dimensionality kicks in (Han et al., 2018). This is further exacerbated if there is a need to solve PDEs across varying domain geometries, parameters, and initial/boundary conditions, requiring independent simulations for each unique configuration.

The main prominent approach that has emerged is based on integrating physical laws and constraints into the learning process to solve PDEs (Raissi et al., 2019), called physics-informed neural networks (PINNs), bypasses the need for direct discretization. However, classical PINNs are not without

---

[1]The idea of canonicalization in many ways is not new, see Appendix D.1.

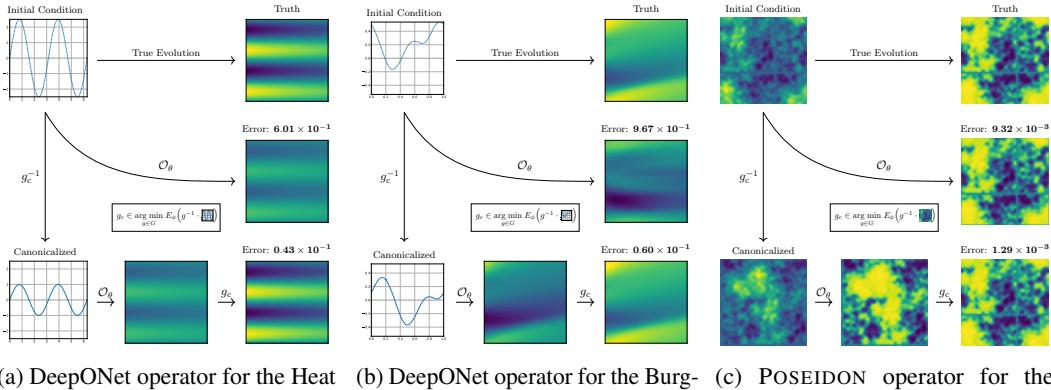

(a) DeepONet operator for the Heat equation Appendix D.3.

(b) DeepONet operator for the Burgers' equation Appendix D.4.

(c) POSEIDON operator for the Allen–Cahn equation Section 4.3.1.

Figure 2: Canonicalization pipeline for numerical PDE evolution Section 4.3.

limitations (Grossmann et al., 2023; Krishnapriyan et al., 2021). See Appendix A for a further discussion. To address the challenge of solving PDEs across diverse scenarios, Neural Operator models have emerged as a promising solution (Li et al., 2021a; Raonic et al., 2024). Unlike standard DNNs, which approximate functions over finite-dimensional vector spaces, neural operators learn mappings between infinite-dimensional function spaces. These learned operators map parameters or IBCs to corresponding PDE solutions. Combining such Neural Operator models with a physics-informed loss has resulted in physics-informed neural operators (Li et al., 2021b; Wang et al., 2021b), oftentimes improving generalization and physical validity, compared to their physics un-informed variants. In what follows, we concern ourselves precisely with this class of approaches.

## 1.3 EQUIVARIANCE IN PINNs

Based on the discussions above, a natural question arises - *can generalization of physics-informed neural networks be improved by utilizing symmetries arising in the study of PDEs?* A few strides have been made towards understanding ways in which to take advantage of the Lie point symmetries (LPS) of differential equations in neural solvers. The simplest ways to add symmetry information into neural PDE solvers is through data augmentation (Brandstetter et al., 2022; Li et al., 2022), resulting in more expensive training, or loss augmentation (Akhound-Sadegh et al., 2023; Li et al., 2023; Zhang et al., 2023), resulting in more difficult training. Both of these, however, oftentimes result in improved generalization. Similar benefits have also been observed in Gaussian process based solvers Dalton et al. (2024). Alternatively, exact equivariance can be sought. As an example, in Wang et al. (2021a), neural PDE solvers were combined with standard approaches to designing equivariant architectures for simple symmetries such as translations, roto-translations and scalings. Still, differential equations may have significantly more complicated Lie point symmetry groups than the groups for which there are off-the-shelf equivariant architectures. The concept of moving frames has been applied to the problem of using PINNs to solve ODEs with LPS in Arora et al. (2024). Here, the approach taken was to invariantize the ODE in question, and apply a PINN to this equation, before mapping back to an (approximate) solution of the original ODE. Similarly, Lagrave & Tron (2022) uses equivariant neural networks to first approximate differential invariants, before passing to a classical numerical scheme for evolving the solution. Overall, the literature so far has illustrated that Lie point symmetries can provide useful inductive biases for Neural PDE solvers, and from a practical point of view can even be approximated from observed data (Gabel et al., 2024). However it still remains an open question, whether it is possible to make existing physics-informed approaches equivariant. In this paper we directly tackle this question. Preliminaries are provided in Appendix A.

## 1.4 OVERVIEW OF CONTRIBUTIONS

Motivated by limitations of existing equivariant architectures, we turn to *energy-based canonicalization* as a means for inducing equivariance in existing models. This approach was first introduced in Kaba et al. (2023), but remains largely unexplored. In this paper we revisit and extend this approach, and explore its utility in scientific machine learning applications. Noting that many existing

definitions and results in the prior literature do not extend readily to the general setting, we extend and refine existing definitions to handle non-discrete and non-compact groups. Our key contribution is an energy-based canonicalization approach that provides a unifying theoretical framework for constructing frames and canonicalizations (Section 2.2). To define energy-based canonicalization we introduce a generalization of standard canonicalization, allowing use with non-compact Lie groups, which we call weighted canonicalization. This new framework encapsulates prior work of Dym et al. (2024) on weighted frames, and our energy-based canonicalizations live naturally in this framework.

We emphasize that our proposed approach is generic, in that it has the potential to apply to a wide range of models and learning tasks. The energy canonicalization framework developed here is readily applicable for an ***arbitrary*** group with an ***arbitrary*** action, including those not acting linearly, therefore being extendable to gauge and local symmetries. For Lie groups with smooth actions, one only requires knowledge of the action of some basis of the Lie algebra, without requiring a global parametrization of the group.

This flexibility stems from the choice of energy functional, which directly influences empirical performance and the size of resulting canonicalizations. We offer guidelines for constructing energy functionals and for performing Lie group optimization in Section 3. We showcase the effectiveness of our method on image classification tasks under homography and affine group invariance in Section 4.2. We further discuss turning neural operators Lie point symmetry equivariant on three examples, including the pre-trained POSEIDON foundation model (Herde et al., 2024) in Section 4.3.

## 2 GENERAL FRAMEWORK FOR CANONICALIZATIONS AND FRAMES

For this section let $G$ be a Lie group acting smoothly on a manifold $X$. Let $C_b^0(X)$ be the bounded continuous $\mathbb{R}$-valued functions on $X$. Let $\mathrm{PMeas}(X)$ be the set of Radon probability measures on $X$, with respect to the Borel $\sigma$-algebra on $X$. We will set $\phi \in C_b^0(X)$ some function to be acted on.

The two main ways of imposing equivariance on arbitrary models: *frame averaging* and *canonicalization*, both arise from the study of the Reynolds operator. For finite groups $G$, defined as

$$\mathcal{R}_G(\phi)(x) = \frac{1}{|G|} \sum_{g \in G} g \cdot \phi(g^{-1} \cdot x).$$

Evaluating this equivariant function $\mathcal{R}_G(\phi)$ can become prohibitively expensive, requiring summation over the entire group for every input $x$. Therefore one seeks ways to restrict the range of this summand while preserving the equivariance of the resulting function. *Frames* are defined, for each element $x \in X$, as a subset of the group $G$ for which to average over, with an equivariance condition detailed below to ensure equivariance of the resulting function. *Canonicalizations*[2] are the opposite perspective, for each element $x \in X$, directly providing a subset of $X$ over which to directly average the function over with an invariance condition to ensure the resulting function is equivariant. The presentation here was reduced to fit the page limit; for a full description see Appendix H.

### 2.1 FRAMES

**Definition 2.1.** *Frames* are functions $\mathcal{F} : X \to 2^G \setminus \{\varnothing\}$ such that for any $x \in X$ and $g \in G$: $\mathcal{F}(gx) = g\mathcal{F}(x)$. We call the space of frames $\mathrm{Fra}_G(X)$.

A frame $\mathcal{F}$ acts on $\phi$ by frame averaging $\mathcal{F}(\phi)(x) = \frac{1}{|\mathcal{F}(x)|} \sum_{g \in \mathcal{F}(x)} \phi(g^{-1}x)$. As $\mathcal{F}(x) = G_x\mathcal{F}(x)$ for any frame $\mathcal{F}$, frame averaging is inherently limited to the case that the frames are finite at each point, i.e. that the set $\mathcal{F}(x)$ is finite $\forall x \in X$. For frames this can only be the case when the group acts with finite stabilizers at all points. This leads to the following definition of Dym et al. (2024). A weighted frame $\mu_x$ acts on $\phi$ similarly by averaging $\mu(\phi)(x) = \int_G \phi(g^{-1} \cdot x) \, d\mu_x(g)$.

**Definition 2.2.** *(Weakly)-Equivariant Weighted Frames* are functions $\mu_{[\cdot]} : X \to \mathrm{PMeas}(G)$ s.t. $\forall x \in X, g \in G : \mu_{gx} = g_*\mu_x$ (respectively: $(\pi_{gx})_*\mu_{gx} = (\pi_{gx})_*g_*\mu_x$ where $\pi_{gx} : G \to G/G_{gx}$ is the quotient map. See Lemma F.2). We call the space of weakly-equivariant weighted frames $\mathrm{WFra}_G(X)$.

---

[2]Original proposal of Kaba et al. (2023) relied on these being singletons, requiring authors to introduce relaxed equivariance. Treating these as sets turns out to be natural for both energy canonicalizations (Section 2.3), and for providing (equivalence) isomorphism of (weighted) canonicalizations with (weighted) frames (Theorem G.3 and Ma et al. (2024)).

## 2.2 CANONICALIZATIONS

We briefly recall the definitions of (finite) canonicalizations from Ma et al. (2024).

**Definition 2.3.** A *canonicalization* is a function $\mathcal{C} : X \to 2^X \setminus \{\varnothing\}$ s.t. $\forall x \in X, g \in G : \mathcal{C}(gx) = \mathcal{C}(x)$. We call the space of canonicalizations $\mathrm{Can}_G(X)$.

**Definition 2.4.** An *orbit canonicalizations* is a canonicalization s.t. $\forall x \in X, \mathcal{C}(x) \subseteq Gx$. We call the space of orbit canonicalizations $\mathrm{OCan}_G(X)$.

A canonicalization $\mathcal{C}$ acts on $\phi$ similarly to frames by $\mathcal{C}(\phi)(x) = \frac{1}{|\mathcal{C}(x)|} \sum_{y \in \mathcal{C}(x)} \phi(y)$. By the orbit-stabilizer theorem, orbit canonicalizations are equivalent to frames. In this isomorphism, frames are always larger by a factor equal to the size of the stabilizer group. Thus from a computational perspective orbit canonicalizations are prefered, and we will consider the canonicalization perspective from now on. We shall also see that this is indeed the correct perspective to work with when $G$ is a non-compact Lie group.

## 2.3 ENERGY-BASED CANONICALIZATION

We are now in a position to introduce the definition of our energy-based canonicalization method. We start with an energy function $E : X \to [0, +\infty]$ which we then minimize over the orbits of the group action $Gx$. First, in the case where $G$ is finite, every orbit in $X$ necessarily has a minimum of $E$, and we note that energy minimization naturally results in a frame (Proposition G.1):

$$\mathcal{F}_E(x) = \underset{g \in G}{\arg\min}\, E(g^{-1}x) \tag{1}$$

Moving onto canonicalizations, we want to note that intuitively the best energy would be some approximation of the negative log likelihood for the sampled distribution. Therefore finding the minimizers of the energy within an orbit has the effect of finding those examples in the orbit which are most likely to have been trained on. In keeping with this intuition, we would like to define the energy minimizing canonicalization to be:

$$\mathcal{C}_E(x) = \underset{y \in Gx}{\arg\min}\, E(y) \tag{2}$$

In the non-compact cases we wish to consider, this definition isn't suitable for two main reasons: the set of minima may not be finite or there may be no minima. Dealing with the first naturally leads us to defining weighted canonicalizations, in analogy with weighted frames:

**Definition 2.5.** A *weighted canonicalization* is a $G$-invariant function $\kappa_{[\cdot]} : X \to \mathrm{PMeas}(X)$. We call the space of weighted canonicalizations $\mathrm{WCan}_G(X)$.

**Definition 2.6.** A *weighted orbit canonicalization* is a weighted canonicalization $\kappa$ such that for every $x \in X$, $\mathrm{supp}\,\kappa_x \subseteq Gx$. We denote the set of weighted orbit canonicalizations as $\mathrm{WOCan}_G(X)$.

A weighted canonicalization $\kappa_x$ acts on $\phi$ similarly as $\kappa(\phi)(x) = \int_X \phi \, d\kappa_x$. As in the non-weighted case, we have a similar equivalence between weighted orbit canonicalizations and weakly-equivariant weighted frames given by Theorem G.3. Weighted canonicalizations do not address the second issue however, as orbits of a non-compact Lie group are not necessarily compact or even closed. The non-compactness of these orbits is easy to solve by making the standard assumption that $E$ is topologically coercive: for any $N \in \mathbb{R}^+$ there is a compact $K \subseteq X$ such that $E|_{X \setminus K} > N$. The non-closedness of orbits is more subtle, as shown by the following example.

*Example.* Let $X = \mathbb{R}^2$ and $G = \mathbb{R}_{>0}$ with its standard multiplicative Lie group structure. Note that $G$ is a non-compact Lie group. Let $t \in G$ act by $t \cdot x = tx$. Then one sees that the orbits of this action are exactly the non-closed rays $\mathbb{R}_{>0}x$ emanating from origin as well as simply the origin. In practice, this might represent a scale-invariant learning problem.

Now consider the energy minimization problem defined by taking the energy to be $E(x) = |x|^2$. The global minima of this function is clearly the origin, which isn't in the domain. On the other hand, any of the non-closed rays have the origin as their limit points, therefore $E$ does not have a minima on any of these orbits.

To get around the non-closedness issue we introduce the following notion. This may seem like an artificial solution to the non-closedness problem, but in fact weighted closed canonicalizations is the most natural extension, as they are simply limits of weighted orbit canonicalizations by Theorem G.5.

**Definition 2.7.** A *weighted closed canonicalization* is a weighted canonicalization $\kappa : X \to$ PMeas$(X)$ such that for every $x \in X$, supp $\kappa_x \subseteq \overline{Gx}$. Call the set of weighted closed canonicalizations WCCan$_G(X)$.

By taking the closures of orbits, we solve the problem of $E$ not having minima, so we are naturally led to constructing weighted closed canonicalizations. First, we define the energy-minimizing set, $\mathcal{M}_E(x) = \arg\min_{y \in \overline{Gx}} E(y)$, set of minima of $E$ on the closure of the orbit of $x$.

**Proposition 2.1.** *(Proposition G.2)* $\mathcal{M}_E(x)$ *is $G$-invariant and non-empty.*

If $\mathcal{M}_E(x)$ is finite for each $x \in X$, then we define the (non-weighted) closed canonicalization $\kappa_E \in$ WCCan$_G(X)$ by taking $\kappa_E(x)$ to be the normalized counting measure on $\mathcal{M}_E(x)$ as usual.

For a general $E$, we may not be able to put any reasonable probability measure on the set $\mathcal{M}_E(x)$, for example if $\mathcal{M}_E(x)$ has highly fractal geometry. We therefore must assume that $E$ is such that for all $x$, the Hausdorff measure of $\mathcal{M}_E(x)$ is non-zero. As each $\mathcal{M}_E(x)$ is compact, the Hausdorff measure is finite, hence combined with this assumption we may normalize it to obtain a probability measure on $\mathcal{M}_E(x)$ which is a weighted canonicalization $\kappa_E(x) \in$ WCCan$_G(X)$. Refer to Appendix E for more discussion on the Hausdorff measure and on why this assumption is reasonable. Note that if $\mathcal{M}_E(x)$ is finite, then the normalized Hausdorff measure constructed agrees with the normalized counting measure. Hence this construction is a generalization of the purely finite case.

## 2.4 Continuity Preserving Canonicalizations

Using the setup above we may now write down our main diagram connecting all the different notions of canonicalization and framing together, shown in Figure 3.

In practice, it is often desired for continuity to be preserved under canonicalization/frame averaging. In Dym et al. (2024) it is shown that there are many topological obstructions to constructing continuous canonicalization functions. In analogy with Dym et al. (2024), we may now ask whether energy canonicalizations presented here take continuous functions to continuous functions. In our framework this is a simple condition to add.

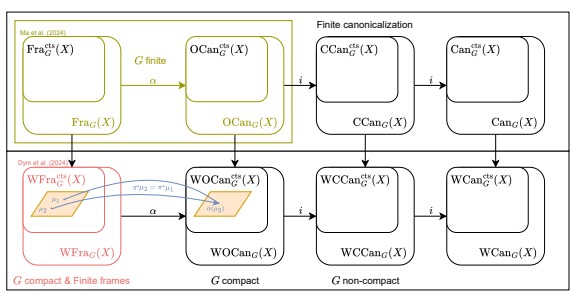

Figure 3: Connections between various notions introduced previously and in this work. Top row: finite frames and canonicalizations, mapping vertically into weighted versions via normalized counting measures. Inside each, contained a sequentially closed subspace of those that preserve continuity.

**Definition 2.8.** A weighted canonicalization $\kappa$ is *continuous* if for every $\phi \in C_b^0(X)$, $\kappa(\phi)$ is continuous. Call the space of continuous weighted canonicalizations WCan$_G^{\text{cts}}(X)$. We similarly define continuous weighted (orbit) canonicalizations and weighted frames.

**Proposition 2.2.** *A weighted canonicalization $\kappa$ is continuous if and only if for every convergent sequence $x_n \to x$ in $X$, the sequence of measures $\kappa_{x_n}$ converges weakly to $\kappa_x$.*

It turns out, that for energy canonicalizations this completely provides a characterization of which energies give rise to continuous weighted canonicalizations.

**Lemma 2.3.** *(Lemma G.7) Let $E$ be a lower semi-continuous energy function. Then $\kappa_E$ is continuous if and only if for every convergent sequence $x_n \to x$ we have* Li $\mathcal{M}_E(x_n) = \mathcal{M}_E(x)$.

## 3 PRACTICAL CONSIDERATIONS

In Section 2.3 we proposed energy canonicalization as a constructive way of achieving equivariance, but a question remains - how to choose the energy[3]? Unfortunately there is no unique way to choose the energy or the optimization routine to find the minima. In this section we instead outline key considerations to guide these choices, with Section 4.1 serving as a guiding example. In complex settings of Sections 4.2 and 4.3.1 we choose it to be a neural network, as explained in Appendix B, while for Section 4.1 and appendices D.3 and D.4, the energy is hand-crafted.

As we may not directly have access to parameterizations of the orbits for Equation (2), we instead implement group descent algorithms for Equation (1), e.g. illustrated in Algorithm 1, and then consider the resulting canonicalizations, as this are much smaller than frames and often.

In what follows, we clarify the interplay between the four primary problem aspects. We assume to be given a specific choice of group and action on a domain, as well as information about the domain itself (either through samples or in the case of synthetic data, full knowledge is available). Based on these, we need to construct two things: (1) the energy function, based on the domain information; and (2) the group-based optimization routine for the problem in Equation (2), based on the domain information and the energy

---

**Algorithm 1:** Canonicalization with a global retraction

**Data:** Non-canonical input $x$
**parameters :** N, steps $\eta_i$, $\xi_0 \in \mathfrak{g}$, Energy $E : \mathcal{X} \to \mathbb{R}$,
       Retraction $\tau : \mathfrak{g} \to G$
**Result:** Canonicalized input $\hat{x} = g^{-1} \cdot x$; inverse $g^{-1}$;
       canonicalizing group element $g$.
`# Do gradient descent on` $\xi \in \mathfrak{g}$
$\xi \leftarrow \xi_0$;
**for** $i = 0 \dots N$ **do**
    | $\xi \leftarrow \xi - \eta_i \nabla_\xi E(\tau(\xi) \cdot x)$
**end**
**return** $\tau(\xi) \cdot x; \tau(\xi); [\tau(\xi)]^{-1}$

---

choice (potentially leading to problem relaxation as e.g. in Section 4.3.1 or improved initialization for iterative optimization as e.g. in Appendix D.3).

**Domain information**    Models trained on a specific domain oftentimes do not generalize beyond their training data. To ensure the canonicalized input remains within the domain of the trained model, domain-specific information should be directly incorporated into the energy function. To give a concrete example, consider a setting in which some neural network is trained on samples living on a compact set $\mathcal{M}$. In such a setting we would only be interested in canonicalizations, which are either close to the set, or live on the set itself. Thus, choosing the energy function to include the characteristic function of the set $\chi_\mathcal{M}$ is particularly suitable when the action is regular (or transitive). Otherwise, there may not be any elements in the set $\mathcal{M}$ that can be mapped to, in which case a relaxation using the distance function $d_\mathcal{M}(x) = \min_{y \in \mathcal{M}} \|x - y\|$ can be used. Such choices turn out to be beneficial, especially in the setting of Section 4.3.

In the case of data sampled from an underlying distribution $x \sim p_\mathcal{X}$, one natural choice would be to consider as energy function the negative log likelihood $E(x) = -\log p_\mathcal{X}(x)$, which may be available in closed form if trained on synthetic data. Otherwise, an approximation can be used. For this, there exist quite a few choices, including normalizing flows (Dinh et al., 2016) or score-based diffusion models (Song et al., 2020) to approximate the true likelihood or its gradient. In this paper, for examples in Sections 4.2 and 4.3.1 we opt for a simpler variational autoencoder (Kingma, 2013), using the ELBO, being a lower bound, instead of the likelihood for the energy.

**Non-convexity and Flatness**    Even with a convex energy, minimization with respect to $g \in G$ introduces non-convexity. The orbit $Gx$ is often non-convex, e.g. see Figure 5, where this is a circle. Invoking a parameterization of a group element through its Lie algebra and considering its action on the space leads to further non-convexity. In the worst cases, such as those considered in PDE examples in Section 4.3, we may not have access to a global group parameterization, significantly complicating the process. What is worse, the energy may exhibit many flat regions, with the gradients being noisy, as in Figure 5.

Thus the problem requires the use of non-convex optimization methods. We find, that in all examples in Section 3, utilizing a multi-initialization strategy with a fixed number of gradient steps is sufficient for convergence. Altenatively, relaxation to the problem in Equation (2) to be performed over $x$, subject to an orbit distance constraint can be used.

---

[3]We note that unlike Kaba et al. (2023), there is no canonicalization network. Constructing a canonicalization network requires one to use equivariant architectures, which may not exist (or may be too expensive) for the groups considered in this paper. Instead, when needed, the energy functional is parameterized as a network.

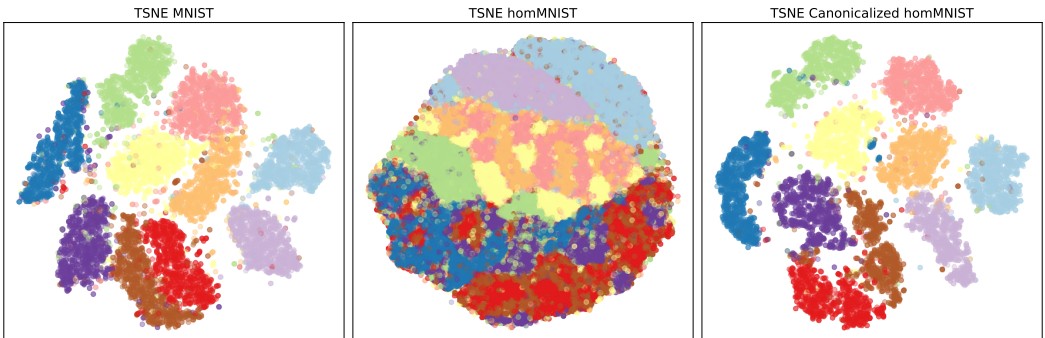

Figure 4: $t$-SNE plots: For MNIST, homNIST and canonicalized homNIST (Section 4.2).

To address the problem of flat zero-likelihood regions, alternative energy functions can be considered, e.g. empirical distance $E(x) = \mathbb{E}_{X \sim p_\mathcal{X}} d(X, x)$. However, in Sections 4.2 and 4.3.1 we make a simpler choice to utilize extra regularization. Due to the difficulty in handcrafting regularization for various specific cases, we utilize learned convex regularization, trained adversarially on real and partially transformed data, based on the recent proposals in adversarial regularization Mukherjee et al. (2024); Shumaylov et al. (2024; 2023). Ideally, the regularization would also ensure uniqueness of minimizers to the problem in Equation (2), however it is not clear how to achieve this. Generically, we would want to construct an energy such that level sets are transverse to group orbits, similar to Panigrahi & Mondal (2024), which is indeed achieved by utilizing adversarial regularization.

**Non-parameterizability** Moving from compact groups to non-compact presents further challenges. Even with access to the exponential mapping, it may not be surjective, making global parameterization of the group using the Lie algebra potentially impossible, if the group is not *Noetherian*. However, Lie algebras of many PDE symmetry groups are solvable, which can be used for group parameterizations that simplify the optimization problem: if the Lie algebra is solvable, any group element can be decomposed using the derived subalgebras (Varadarajan, 2013, Theorem 3.18.11) as a train of exponentials. For example, for the 1D heat equation (Appendix D.3), we can represent any element of the group as $\exp(a_6 v_6) \ldots \exp(a_1 v_1)$, reducing optimization over the group to optimization over $(a_1, \ldots, a_6) \in \mathbb{R}^{\dim(\mathfrak{g})}$. In this case one can utilize Algorithm 1.

When the algebra is not solvable, one has to revert to Lie algebra based descent schemes. However, for Lie point symmetry groups, oftentimes only actions of specific basis vectors of the Lie algebra are available, making it necessary to use a coordinate descent (CD) approach. CD has received considerable attention in classical optimization (see Wright (2015)) and has recently been studied in geometric settings such as Riemannian manifolds and Lie groups (Han et al., 2024). The algorithms for both of these cases can be found in Algorithms 2 and 3. If a global parameterization is available, then classical schemes may be utilized in the same way (Lezcano Casado, 2019; Absil et al., 2008).

## 4 EXPERIMENTS

### 4.1 2D TOY EXAMPLE

To illustrate the universality of the approach, we consider one simple two dimensional example without any machine-learned components. We consider the problem of classification under inherent rotation invariance illustrated on Figure 5. To mimic a realistic setting, we assume to have samples only. In this case, we can approximate the density using kernel density estimation with a Gaussian kernel, illustrated in the second column on Figure 5. As one would expect, without knowledge of the symmetries of the distribution, the approximation is far from the ground truth. In what follows, the energy is chosen to be the negative log-likelihood of the approximated distribution.

This example illustrates the main issue with energy based canonicalization, which also extensively arises in more complex examples: even when the energy is convex, it is not convex when viewed as a function of $g$. This results in a non-convex optimization problem with (potentially many) local

minima, which one can get stuck in when running standard descent schemes, and as discussed in Section 3, one has to revert to more complex optimization routines. In this case, as shown in Figure 5, it is possible to find global minimizers, despite the non-convexity, by running gradient descent for a fixed number of iterations, but for a number of different initializations. The power of canonicalization can then be seen in Figure 1, turning a non-equivariant kNN classifier rotation equivariant.

## 4.2 INVARIANT CLASSIFICATION

To illustrate the effectiveness of LieLAC in achieving invariance of pre-trained models, we first test it on invariant image classification with respect to the groups of affine and homography transformations. We take MacDonald et al. (2022) as the baseline for comparison due to the similarity in the required knowledge of the group required for implementation.

Here, we take a pre-trained convolutional classifer, trained on the original MNIST dataset. The combination with the canonicalizer, as described in Section 3, is then tested on the affine-perturbed MNIST

Table 1: MNIST test accuracy for Affine and Homography groups, compared with affConv and homConv of MacDonald et al. (2022).

| Name | MNIST | affNIST |
|---|---|---|
| CNN | **0.985** | 0.629 |
| LieLAC [CNN] | 0.979 | **0.972** |
| affConv | 0.982 | 0.943 |

| Name | MNIST | homNIST |
|---|---|---|
| CNN | **0.985** | 0.644 |
| LieLAC [CNN] | 0.982 | **0.960** |
| homConv | 0.980 | 0.927 |

(affNIST) (Gu & Tresp, 2020) and the homography-perturbed MNIST (homNIST) (MacDonald et al., 2022). The overall performance results for this example are reported in Table 1. We can see that the canonicalization based approach achieves improved equivariance compared to the baseline, as illustrated by improved test accuracy on the datasets. We can also directly analyse achieved equivariance, as it appears only through canonicalization. We provide examples, visualising achieved equivariance in Figure 6. We can also analyze the effect of canonicalization globally by considering a $t$-SNE plot (Van der Maaten & Hinton, 2008) in Figure 4, showing that energy canonicalization turns the dataset into the usual MNIST form, clearly separating the classes, unlike homNIST.

## 4.3 EQUIVARIANT PDE EVOLUTION

In analogy with Akhound-Sadegh et al. (2023), we first consider two simple examples with non-trivial symmetry groups: the 1D Heat and Burgers' equations, which also admit global parameterisations of the group elements (Koval & Popovych, 2023). Due to space restrictions, evaluation can be found in Appendix D.3 for the Heat and Appendix D.4 for the Burgers' equation. Full description of the experimental setting, further visualisations and theoretical background can also be found in the appendix. Examples of canonicalized solutions are shown in Figure 2.

We explicitly emphasize that in this paper we only consider symmetries for initial value problems, where there is no dependence of $g \cdot x$ and $g \cdot t$ on $u$. I.e. for $(x', t', u') = g \cdot (x, t, u)$, we have $x'(x, t, u) = x'(x, t)$. In this case the action on $x, t$ can be separated from the action on $u$, making the desired equivariance for the physics-informed neural operator to be:

$$\mathcal{O}_\theta[u_0, x_0, t_0](x_f, t_f) = g \cdot \mathcal{O}_\theta[g^{-1} \cdot u_0, g^{-1} \cdot x_0, g^{-1} \cdot t_0](g^{-1} \cdot x_f, g^{-1} \cdot t_f) \tag{3}$$

### 4.3.1 ALLEN–CAHN EQUATION

We test the POSEIDON (Herde et al., 2024) foundation model, fine-tuned on the Allen–Cahn equation (ACE) on data it was trained on, and on transformed data (using Lie-point symmetry group of ACE).

$$u_t = u_{xx} + u_{yy} - \epsilon^2 u \left( u^2 - 1 \right) \tag{4}$$

We consider solving Equation (4) on a 2D domain with periodic boundary conditions for $t \in [0, 1], x \in \Omega = [0, 1]^2$. This equation admits a 4-dimensional Lie algebra, see Appendix D.5, with $SE(2)$ as the point symmetry group.

**Choice of Energy**    In order to justify the choice of energy, we need to understand the POSEIDON (Herde et al., 2024) model itself. Firstly, $x_0$ and $t_0$ in Equation (3) are implicitly defined in the model to be fixed to the standard domain $x_0 = \Omega$, and $t_0 = 0$. As such, the only input is the initial condition $u_0|_\Omega$. Therefore, we are going to have to only consider transformations, that preserve $x_0$ and $t_0$. This

can be imposed directly within the energy canonicalization framework as described in Section 3 by considering the characteristic functions as follows:

$$E_{\mathrm{AC}}(x_0, t_0, u_0, x_f, t_f) = \chi_{\{\Omega\}}(x_0) + \chi_{\{0\}}(t_0) + \chi_{[0,1]}(t_f) + \chi_\Omega(x_f) + E(u_0) \qquad (5)$$

Note, that this is able to significantly reduce the computational cost of optimization, as the group generated by $v_4$ in 28, resulting in rotations of the underlying domain, can be reduced to a discrete subgroup $C_4 \leq \mathrm{SO}(2)$, as the only transformations such that $g^{-1}\Omega = \Omega$. Further information on the training and testing data is available in Appendix D.5. The overall performance is reported in Table 2.

**Finetuning** As seen in prior works (Mondal et al., 2023), the canonicalized data may end up misaligned with the original data. For this reason finetuning of the model/canonicalization may be desired.

Table 2: ACE Test error evaluated over 90 trajectories for in-dsitribution (ID) and out-of-distribution (OOD). All errors are ($\times 10^{-4}$).

| Name ($\times 10^{-4}$) | ID Error eq. (29) | OOD Error eq. (30) | Avg |
|---|---|---|---|
| POSEIDON | $\mathbf{6.93 \pm 2.83}$ | $75.76 \pm 6.80$ | 41.35 |
| LieLAC [POSEIDON] | $16.69 \pm 5.42$ | $29.19 \pm 5.64$ | 22.94 |
| POSEIDON + ft (can). | $8.45 \pm 2.87$ | $20.09 \pm 3.32$ | 14.27 |
| LieLAC [POSEIDON+ ft.] | $10.23 \pm 3.36$ | $\mathbf{11.34 \pm 2.92}$ | $\mathbf{10.79}$ |
| POSEIDON + ft. (data aug) | $8.41 \pm 3.48$ | $8.50 \pm 3.11$ | 8.46 |

It is important to emphasize that finetuning is not necessary, and even without finetuning the out of distribution errors go down significantly, as seen in Table 2. All of the examples above, including kNN (Section 4.1), MNIST (Section 4.2), Heat and Burgers (Appendices D.3 and D.4) did not require finetuning. That is due to tasks not being as complex, and not requiring the same level of precision. POSEIDON, being pre-trained in a regressive manner, turns out to be very good to begin with, and finetuning is performed to illustrate that baseline accuracy of data-augmentation (Brandstetter et al., 2022) can be matched.

While it may seem that finetuning the resulting operator on canonicalized data is as hard as augmenting the original data, it turns out to not be, as fine-tuning can be done on a very small data-pool (in this case only 100 trajectories). This stems from the energy minima in Equation (2) residing near, but not necessarily on, training data. If the orbits are well-sampled in training, finetuning is not necessary, e.g. seen in Section 4.2. However, based on the training data in eq. (29), the orbits under $C_4$ are well-sampled, while the orbits under $\mathrm{SE}(2)$ are not, illustrated in Table 2, wherein the error falls 2/3-fold after finetuning. An example is shown on Figure 2c, with further illustrations in Figure 13.

## 5 CONCLUSION

In this paper we provided a unified theoretical framework for defining frames and canonicalizations over non-compact and infinite groups. We showed that energy-based canonicalization can be used constructively to obtain both, and provide a Lie algebra based approach for turning any pre-trained network equivariant. We showcased that the proposed method is effective in turning image classifiers invariant, as well as neural operators Lie point symmetry equivariant.

**Future work** Our example in Section 4.3.1 illustrates the promise of the proposed framework for fine-tuning foundation models. Due to the typically high cost of (re-)training foundation models, further exploration of such applications is an important direction for future work.

Of particular interest are "realistic" scientific machine learning applications: In such applications collecting sufficiently large training data sets may be prohibitively expensive or technically impossible, i.e., available data not sufficient for retraining existing models/ using standard fine-tuning approaches. Encoding symmetries explicitly allows for more efficient fine-tuning, which may be more accessible in settings with little data.

**Limitations** Energy-based canonicalization suffers from slow inference (resulting in $5 - 30\times$ slowdown). Improvements to the optimization are possible via non-convex optimization techniques (e.g., simulated annealing (Baudoin et al., 2008; Rutenbar, 1989), particle swarm (Huang et al., 2023), consensus-based (Totzeck, 2021; Fornasier et al., 2024)), particularly applicable as Lie groups are oftentimes low-dimensional, while the ambient space is not. The majority of these, however, are not easily adaptable to the setting of Lie groups.

Implementing Lie point symmetries in existing neural operators is difficult due to hard-coded boundary conditions. Future work should focus on foundation models handling initial/boundary conditions and parameters, or explore alternative PDE symmetries (Brandstetter et al., 2022). This could include studying infinite-dimensional symmetry groups, which would also naturally arise when studying equivariance to local transformations.

ACKNOWLEDGMENTS

The authors would like to thank Olga Obolenets, Lucy Richman, Hong Ye Tan and Yury Korolev for useful discussions and feedback. ZS acknowledges support from the Cantab Capital Institute for the Mathematics of Information, Christs College and the Trinity Henry Barlow Scholarship scheme. FS acknowledges support from the EPSRC advanced career fellowship EP/V029428/1. MW was supported by NSF Awards DMS-2406905 and CBET-2112085, and a Sloan Research Fellowship. CBS acknowledges support from the Philip Leverhulme Prize, the Royal Society Wolfson Fellowship, the EPSRC advanced career fellowship EP/V029428/1, EPSRC Grants EP/S026045/1 and EP/T003553/1, EP/N014588/1, EP/T017961/1, the Wellcome Innovator Awards 215733/Z/19/Z and 221633/Z/20/Z, the European Union Horizon 2020 research and innovation programme under the Marie Skłodowska-Curie Grant agreement No. 777826 NoMADS, the Cantab Capital Institute for the Mathematics of Information and the Alan Turing Institute.

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

# Appendices

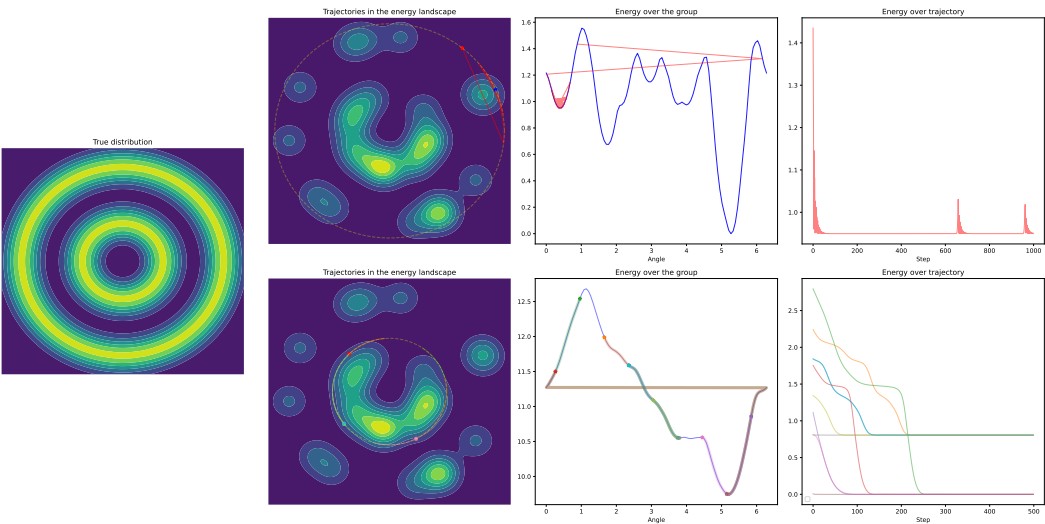

Figure 5: LieLAC optimised sample from a mix of Gaussian distributions.

# A   EXTENDED BACKGROUND

## A.1   EQUIVARIANT CONVOLUTIONS

An important fact that comes up repeatedly in the design of equivariant neural networks is that ordinary convolutions on Euclidean spaces can be generalized to any group on which we can integrate against an invariant measure (as is the case for any Lie group, for example), giving rise to equivariant linear maps in the form of *group convolutions*. Furthermore, converse results can be established, showing that a linear equivariant map (under general conditions) must take the form of a group convolution. This is the approach taken in Cohen & Welling (2016) for discrete groups of roto-translations acting on images. Since group convolutions are applied to signals defined on the group, it is necessary to apply an initial equivariant linear lifting operation to convert a signal on the original domain to a signal on the group. Incompatibility of continuous groups and grid-based discretizations of signals is a major issue in methods that take this approach, and for general groups, it is not necessarily clear how one should discretize the integral defining a group convolution. For Lie groups, there have been various proposals for approximation methods for these integrals based on finite samples, including the works in Bekkers (2020); Finzi et al. (2020b) and works following them, which focus more heavily on using the Lie algebra for approximating group convolutions (MacDonald et al., 2022; Kumar et al., 2024). The corresponding methods have been shown to result in more efficient use of training data and more robustness to transformations not seen during training, and have been applied successfully in areas such as medical image analysis (Bekkers et al., 2018; Winkels & Cohen, 2019), computational molecular sciences (Batzner et al., 2022; Batatia et al., 2022) and inverse problems in imaging (Celledoni et al., 2021; Chen et al., 2023). Beyond construction of equivariant neural networks, information about the symmetries can be extracted directlty from data (Moskalev et al., 2022; Gruver et al.).

It is worth noting that some specific groups allow for construction of more efficient equivariant architectures. For example, recent work on geometric and clifford algebra networks Zhdanov et al. (2024); Ruhe et al. (2024) provides an efficient parameterization for groups of symmetries of vector spaces, like Euclidean (Brehmer et al., 2024) and Minkowski groups (Spinner et al., 2024).

## A.2   PHYSICS-INFORMED MACHINE LEARNING

In a separate direction, deep neural networks are being increasingly used in scientific computing, particularly in simulating physical systems modeled by partial differential equations (PDEs). This surge in interest is largely driven by limitations of traditional numerical solvers, as they often struggle with demanding scenarios, such as those involving non-linear and non-smooth PDEs necessitating

fine discretization, or high-dimensional problems where the curse of dimensionality kicks in (Han et al., 2018). This is further exacerbated if there is a need to solve PDEs across varying domain geometries, parameters, and initial/boundary conditions, requiring independent simulations for each unique configuration.

The main prominent approach that has emerged is based on integrating physical laws and constraints into the learning process to solve PDEs (Raissi et al., 2019). These neural networks have been dubbed PINNs, and have been shown to be useful for solving forward and inverse problems for PDEs (Raissi et al., 2019; Sun et al., 2020; Zhu et al., 2019; Karumuri et al., 2020; Sirignano & Spiliopoulos, 2018), and regularizing learning (Zhang et al., 2024).

PINNs are able to handle irregular domains and complex geometries, overcoming some of the challenges of classical methods. However, classical PINNs are not without limitations (Grossmann et al., 2023; Krishnapriyan et al., 2021).The original method is unable to handle varying parameters or input and boundary conditions (IBCs), and often faces challenges with training due to unbalanced gradients and non-convex loss landscapes, requiring careful hyperparameter tuning or preconditioning Müller & Zeinhofer (2024).

To address the challenge of solving PDEs across diverse scenarios, neural operator models have emerged as a promising solution. Unlike standard DNNs, which approximate functions over finite-dimensional vector spaces, neural operators learn mappings between infinite-dimensional function spaces. These learned operators map parameters or IBCs to corresponding PDE solutions, overcoming the computational burden of performing independent simulations for each scenario. Combining such neural operator models with a physics informed loss has resulted in physics-informed neural operators (Li et al., 2021b; Wang et al., 2021b), oftentimes improving generalization and physical validity, compared to their physics un-informed variants. In what follows, we concern ourselves precisely with this class of approaches.

### A.3   WHY LIE POINT SYMMETRIES?

What is appealing about LPS, is that for a given PDE, under certain regularity conditions, all such (simply connected) symmetries can be derived in the form of one-parameter transformation groups (Olver, 1993). For many classic PDEs these have been derived and can e.g. be found in (Ibragimov, 1995). There further exists symbolic software packages, e.g. Baumann (1998), to derive symmetries for general PDEs.

A.4 SYMMETRIES OF PARTIAL DIFFERENTIAL EQUATIONS

Partial differential equations (PDEs) are the most generic mathematical model used to describe dynamics of various systems in sciences. In what follows, we will be working primarily with open subsets of euclidean space, but these ideas generically extend to more complicated spaces. The formal presentation here closely follows Olver (1993), and an interested reader is referred there for more in-depth discussions of the concepts.

This subsection introduces the concept of PDE point symmetries – transformations that map solutions of a given PDE to other solutions of the same PDE. To analyze these symmetries, traditionally focus is shifted to Jet spaces, where the problem of studying PDE symmetries is transformed into the simpler task of studying symmetries of algebraic equations. By employing one-parameter groups of transformations, it becomes possible to systematically derive all such symmetries by examining their infinitesimal actions.

Suppose we are considering a system $\Delta$ of $n$-th order differential equations involving $p$ independent variables $x = \left(x^1, \ldots, x^p\right) \in X = \mathbb{R}^p$, and $q$ dependent variables $u = \left(u^1, \ldots, u^q\right) \in U = \mathbb{R}^q$. PDEs define relations between various derivatives of the function, for denoting which we use the following multi-index notation for $J = (j_1, \ldots, j_k)$, with $\partial_J f(x) = \frac{\partial^k f(x)}{\partial x^{j_1} \partial x^{j_2} \cdots \partial x^{j_k}}$.

**Jet spaces and Prolongations:** Consider $f : X \to U$ smooth and let $U_k$ be the Euclidean space, endowed with coordinates $u_J^\alpha = \partial_J f^\alpha(x)$ so as to represent the above derivatives. Furthermore, set $U^{(n)} = U \times U_1 \times \cdots \times U_n$ to be the Cartesian product space, whose coordinates represent all the derivatives of functions $u = f(x)$ of all orders from 0 to $n$. The total space $X \times U^{(n)}$, whose coordinates represent the independent variables, the dependent variables and the derivatives of the dependent variables up to order $n$ is called the $n$-th order jet space of the underlying space $X \times U$. Given a smooth function $u = f(x)$, there is an induced function $u^{(n)} = \mathrm{pr}^{(n)} f(x)$, called the $n$-th prolongation of $f$, defined by the equations $u_J^\alpha = \partial_J f^\alpha(x)$. Thus $\mathrm{pr}^{(n)} f$ is a function from $X$ to the space $U^{(n)}$, and for each $x$ in $X$, $\mathrm{pr}^{(n)} f(x)$ is a vector representing values of $f$ and all its derivatives up to order $n$ at the point $x$.

**PDE Symmetry groups:** With the notation defined, we can now write down the PDE, given as a system of equations $\Delta\left(x, u^{(n)}\right) = 0$, with $\Delta : X \times U^{(n)} \to \mathbb{R}^l$ smooth. The differential equations themselves tell where the given map $\Delta$ vanishes on $X \times U^{(n)}$, and thus determine a subvariety

$$S_\Delta = \left\{ \left(x, u^{(n)}\right) : \Delta\left(x, u^{(n)}\right) = 0 \right\} \subset X \times U^{(n)}$$

Solutions of the system will be of the form $u = f(x)$. A symmetry group of the system $\Delta$ will be a local group of transformations, $G^\Delta$, acting on some open subset $M \subset X \times U$ in such that "$G$ transforms solutions of $\Delta$ to other solutions of $\Delta$".

**Prolongations of group actions and vector fields** Now suppose $G$ is a local group of transformations acting on an open subset $M \subset X \times U$ of the space of independent and dependent variables. There is an induced local action of $G$ on the $n$-jet space $M^{(n)}$, called the $n$-th prolongation of $G$ denoted $\mathrm{pr}^{(n)} G$. This prolongation is defined so that it transforms the derivatives of functions $u = f(x)$ into the corresponding derivatives of the transformed function $\tilde{u} = \tilde{f}(\tilde{x})$. Consider now v a vector field on $M$, with corresponding (local) one-parameter group $\exp(\varepsilon \mathrm{v})$. The $n$-th prolongation of v, denoted $\mathrm{pr}^{(n)} \mathrm{v}$, will be a vector field on the $n$-jet space $M^{(n)}$, and is defined to be the infinitesimal generator of the corresponding prolonged one-parameter group $\mathrm{pr}^{(n)}[\exp(\varepsilon \mathrm{v})]$. In other words,

$$\mathrm{pr}^{(n)} \Big|_{\left(x, u^{(n)}\right)} = \frac{d}{d\varepsilon}\Big|_{\varepsilon=0} \mathrm{pr}^{(n)}[\exp(\varepsilon \mathbf{v})]\left(x, u^{(n)}\right)$$

for any $\left(x, u^{(n)}\right) \in M^{(n)}$. These can be derived using Olver (1993, Theorem 2.36).

**Theorem A.1** (2.31 in Olver (1993)). *Suppose $\Delta\left(x, u^{(n)}\right) = 0$, is a system of differential equations of maximal rank defined over $M \subset X \times U$. If $G$ is a local group of transformations acting on $M$, and*

$$\mathrm{pr}^{(n)} \mathrm{v} \left[\Delta\left(x, u^{(n)}\right)\right] = 0, \quad \text{whenever} \quad \Delta\left(x, u^{(n)}\right) = 0$$

*for every infinitesimal generator v of $G$, then $G$ is a symmetry group of the system.*

## A.5 DEEP LEARNING FOR PDEs

The abstract discussion of symmetries of PDEs above only concerned itself with transformations leaving the set of solutions invariant. In practice, we are often interested in problems, for which the solution is unique due to either initial or boundary conditions (IBCs). In this paper, we primarily concern ourselves with temporal PDEs in the form of

$$\Delta = u_t + \mathcal{D}_x[u] = 0, \text{ and } u(0, x) = f(x), \quad x \in \Omega, \quad t \in [0, T]$$
$$u(t, x) = g(t, x), \quad x \in \partial\Omega, \quad t \in [0, T]$$

where $u(t, x) \in \mathbb{R}^{\dim u}$ is the solution to the PDE, $t$ denotes the time, and $x$ is a vector of spatial coordinates. $\Omega \subset \mathbb{R}^{\dim x}$ is the domain and $\mathcal{D}_x$ is a (potentially non-linear) differential operator. $f(x)$ is known as the initial condition function and $g(t, x)$ describes the boundary conditions. As discussed in Section 1.2, we are interested in physics-informed approaches, wherein $u(t, x)$ is parameterised as a neural network $u_\theta(t, x)$ with parameters $\theta$, and the network is trained by minimizing a loss comprised of two parts: $\mathcal{L}(\theta) = \mathcal{L}_{\text{PDE}} + \mathcal{L}_{\text{Data}}$. The first term encourages the network to satisfy the PDE, while data-fit is a supervised loss, enforcing IBC and ensuring a good fit with the training data. Specific forms of the losses, and further discussion can be found in Appendix D.2.

## A.6 LIE GROUPS

In this section we briefly recall some necessary notions from Lie group theory. For a more comprehensive introduction, see Kirillov (2008). A Lie group $G$ is a group with a smooth manifold structure such that both the inverse and multiplication maps are smooth. A Lie algebra is a finite dimensional vector space $V$ with an alternating bilinear map $[\cdot, \cdot] : V \times V \to V$ called the Lie bracket, satisfying the Jacobi identity. Given a Lie group $G$, the tangent space at the identity $\mathfrak{g} = T_e G$ naturally has the structure of a Lie algebra. A 1-parameter subgroup of $G$ is simply a group homomorphism $\mathbb{R} \to G$, where $\mathbb{R}$ is a group under addition. For every Lie algebra vector $v$ there is a unique 1-parameter subgroup $\phi_v$ such that $\phi_v(0)$ is the identity and $\phi_v'(0)(\partial_t) = v$. There is a smooth map $\exp : \mathfrak{g} \to G$ which sends $v$ to $\phi_v(1)$. This has image contained inside $G_0$, the connected component of the identity $e \in G$. If $G_0$ is compact then $\exp$ surjects onto $G_0$, but in the non-compact case it doesn't necessarily surject.

For a Lie group $G$ acting smoothly on a manifold $X$ we have a version of the orbit-stabilizer theorem.

**Theorem A.2.** *The stabilizer $G_x \subseteq G$ is a closed Lie subgroup, and the natural map $G/G_x \to X$ is an injective immersion with image $Gx$.*

## B  CHOOSING THE ENERGY

Based on discussion in Section 3, we wish to construct an energy, which contains information about the domain (or dataset) and the energy levels are transverse to orbits. The first we achieve by using a variational autoencoder, and choosing the ELBO as the energy itself; and second by training an appropriate (convex) adversarial regularizer. Thus the overall energy $E = \mathcal{L} + \alpha R$, for some constant $\alpha > 0$ and terms below. We emphasize once again that this is simply *a construction*, which have observed to work well in practice. Effectively all that is required is to align minimas of such an energy with the training set - doing this directly is not that simple and should be a fruitful direction for future work.

**Variational Autoencoders**  Variational Autoencoders (VAEs) (Kingma, 2013) are a class of generative models that learn the underlying probability distribution of a dataset. VAEs consist of two main components:

1. **Encoder:** This component maps an input data point $x$ to a lower-dimensional latent representation $z$. Instead of directly mapping to a single point in latent space, the encoder outputs parameters for a probability distribution, typically a Gaussian distribution, $q(z|x) = \mathcal{N}(z; \mu(x), \sigma^2(x))$. Here, $\mu(x)$ and $\sigma^2(x)$ are the mean and variance of the latent distribution, respectively, and are outputs of the encoder network.

2. **Decoder:** This component takes a point in the latent space $z$ and maps it back to the original data space, reconstructing the input data $\hat{x}$. This is achieved through a probability distribution $p(x|z)$, also often modeled as a Gaussian.

VAEs are trained by maximizing a lower bound on the data log-likelihood, known as the Evidence Lower Bound (ELBO):

$$\mathcal{L}(x) = \mathbb{E}_{q_\theta(z|x)}[\log p_\theta(x|z)] - D_{\mathrm{KL}}(q_\theta(z|x)||p(z)) \tag{6}$$

The first term in the ELBO encourages the decoder to reconstruct the input data accurately. The second term is the Kullback-Leibler (KL) divergence between the learned latent distribution $q(z|x)$ and a prior distribution $p(z)$, typically a standard normal distribution. This term acts as a regularizer, encouraging the latent space to be well-structured and preventing overfitting.

**Adversarial Regularization**  Adversarial regularization Lunz et al. (2018); Mukherjee et al. (2024) is a framework for solving inverse problems that incorporates a discriminator network to learn and impose data-driven regularization. The discriminator network is trained to distinguish between unregularized reconstructions and real solutions obtained from ground truth data. The regularizer is trained with the objective of favoring solutions that are similar to the ground-truth images in the training dataset and penalizing reconstructions with artifacts.

The adversarial loss function for the regularizer can be formulated as:

$$\mathcal{L}_{\mathrm{adv}}(R) = \mathbb{E}_{p_X(x)}[R(x)] - \mathbb{E}_{p_n(\hat{x})}[R(\hat{x})] \tag{7}$$

where $R$ is the regularizer, $x$ is a ground truth image, $\hat{x}$ is a noisy measurement, $p_X$ and $p_n$ are the data distributions. This loss encourages the regularizer to produce a small output when a true image is given as input and a large output when it is presented with an unregularized reconstruction.

In our setting, the noisy measurement distribution is chosen to be $\hat{x} = g \cdot x$, for $x \sim p_X$ and $g \in G$ sampled randomly. Thus the regularizer learns to discern between true looking images and group transformed ones - leading to level sets lying transverse to the orbits.

Adversarial regularization has been shown to provides a flexible data-driven approach to impose priors in inverse problems, which lends it readily to be used in our problem.

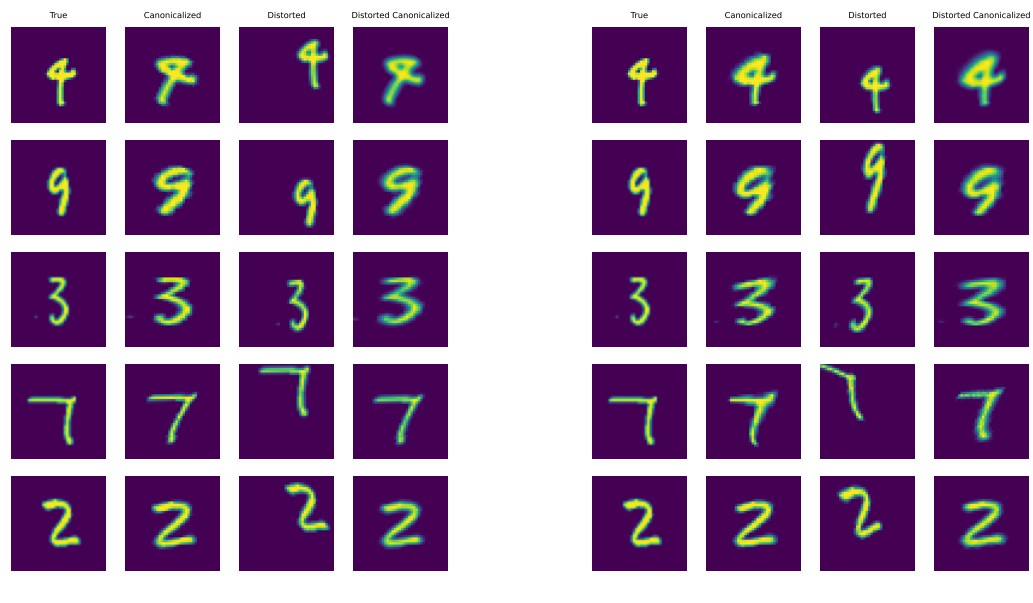

(a) Affine MNIST examples  (b) Homography MNIST examples

Figure 6: MNIST canonicalization images for both Affine and Homography groups, as described in Section 4.2. From left to right: Original MNIST image; Canonicalization of the original image; Original image distorted by a random homography transformation; Canonicalization of the distorted image. This figure illustrates the equivariance properties of energy canonicalization.

## C  FURTHER MNIST VISUALISATION

Of particular interest are the failure modes of canonicalization, warranting further theoretical investigation. We present the failure modes in Figure 7c, and as we can see from the examples provided - in majority of the cases the reason why failure modes exists is due to existence of group transformations resulting in completely different digits! As such, the overall distribution is *not* exactly equivariant.

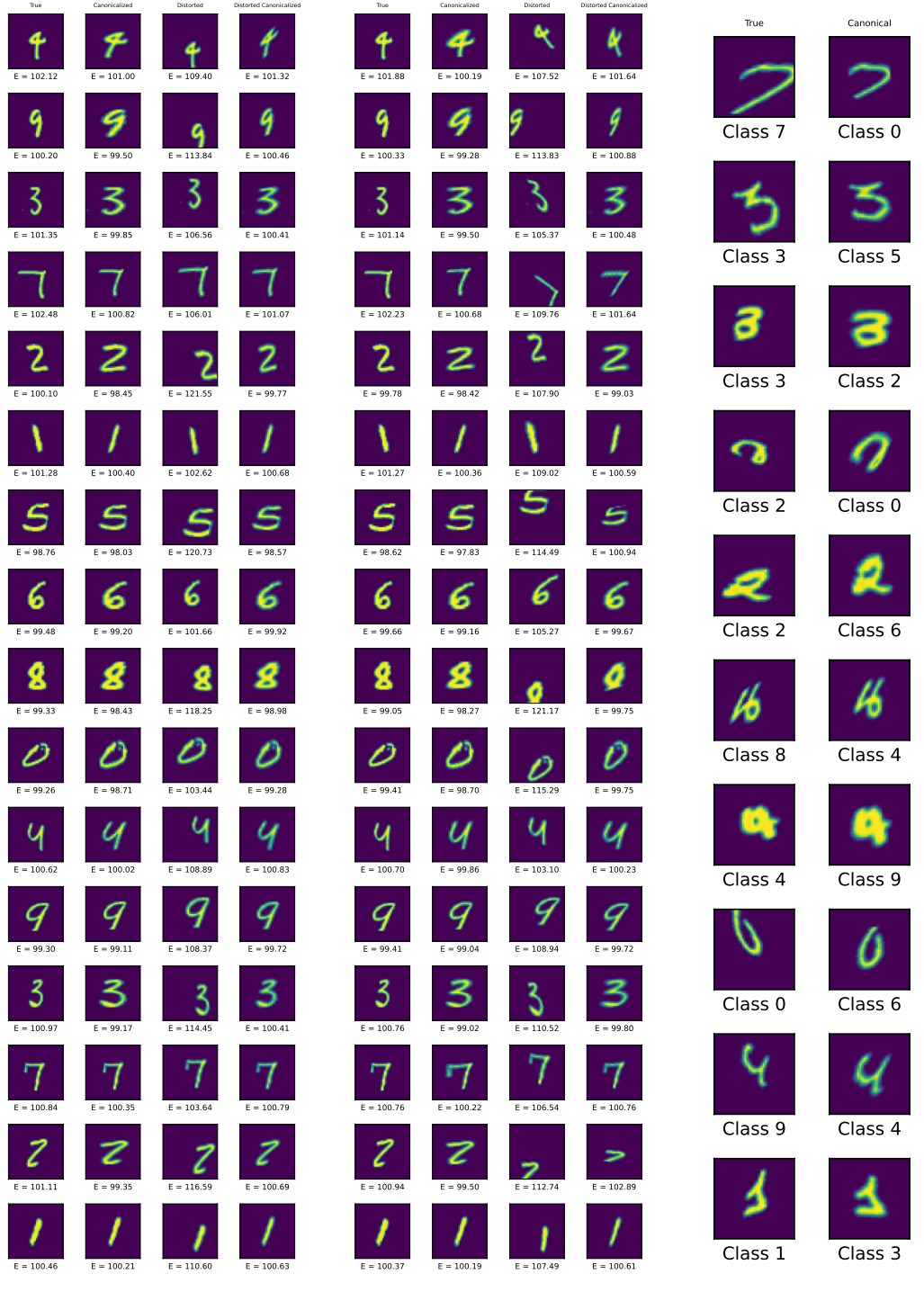

(a) Affine MNIST examples      (b) Homography MNIST examples      (c) Misclassification examples

Figure 7: Expanded Visualisations

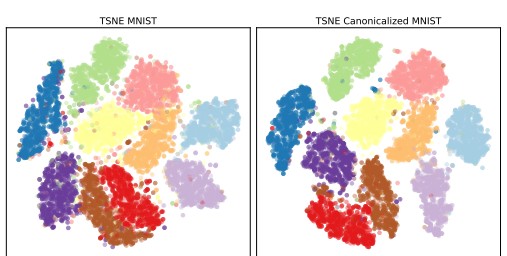 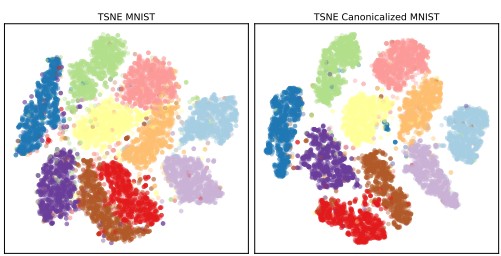

(a) TSNE post-affine canonicalization    (b) TSNE post-homography canonicalization

Figure 8: TSNE plots

## D FURTHER PDE DISCUSSION

### D.1 CANONICALIZATION AS A UNIFYING VIEW

The idea of putting the evolutionary equation and initial into a canonical form prior to evolving is not novel, and in fact quite standard in all numerical analysis textbooks. The first step in solving an ODE numerically is to normalize the equations and remove units and normalize the domain. For example, when solving a time evolution equation on the domain $x \in [0, 15]$, one would first normalize the $x$ variable by rescaling it to be in $[0, 1]$, and solve on this domain instead. Floating point representations are much coarser for smaller values and as such, this can lead to lower numerical errors.

Another example is the recent work on achieving exact unit equivariance - which is precisely scaling group canonicalization (Villar et al., 2023). A similar problem also often arises in the problems of image registration (Kanter & Lellmann, 2022) with an infinite-dimensional Lie algebra of diffeomorphisms. Another particular example being Procrustes analysis (Gower, 1975), when considering only the homography group.

In this work, we present that canonicalization can be seen as the unifying approach, especially through the lens of energy canonicalization. Insights from the problems above can then be used to improve the optimization, and used for PDE symmetry groups.

### D.2 ON DEEP LEARNING FOR PDES

In what follows we demonstrate the effectiveness of canonicalization across three example PDEs Heat, Burgers, Allen–Cahn with baseline models taken from three recent state of the art physics-informed machine learning papers Akhound-Sadegh et al. (2023); Wang et al. (2021b); Herde et al. (2024) respectively. In each case we take the pre-trained neural PDE solver and demonstrate a failure mode when evaluating on out-of-distribution initial conditions. We then optimize for a group action derived from the Lie algebra based energy minimization and report results after the canonicalization technique is applied. Results and additional experimental details for each PDE are provided in the following sections. In PINNs, the PDE solution $u(t, x)$ is parameterised as a neural network $u_\theta(t, x)$ with parameters $\theta$. The loss function is then compromised of two parts, with different weightings:

$$\mathcal{L}(\theta) = \alpha_{\text{PINN}} \mathcal{L}_{\text{PDE}} + \gamma_{\text{data}} \mathcal{L}_{\text{data-fit}}$$

The first term is the physics-informed objective of Raissi et al. (2019), ensuring the neural network satisfies the PDE. The PDE loss is the calculated as the residual, with derivatives are calculated using automatic differentiation, as calculated on a finite set of points $(t, x)_{1:N_{\text{dom}}}$ which are sampled from inside the domain $[0, T] \times \Omega$ to obtain the PDE loss:

$$\mathcal{L}_{\text{PDE}} = \frac{1}{N_{\text{dom}}} \sum_{i=1}^{N_{\text{dom}}} \|\partial_t u_\theta(t_i, x_i) + \mathcal{D}_x \left[ u_\theta(t_i, x_i) \right]\|_2^2 \tag{8}$$

The second term, is a supervised loss enforcing initial and boundary conditions:

$$\mathcal{L}_{\text{data-fit}} = \frac{1}{N_{\text{ic}}} \sum_{i=1}^{N_{\text{ic}}} \left\| u_\theta\left(0, x_i^0\right) - f\left(x_i^0\right) \right\|_2^2 + \frac{1}{N_{\text{bc}}} \sum_{i=1}^{N_{\text{bc}}} \left\| u_\theta\left(t_i^b, x_i^b\right) - g\left(t_i^b, x_i^b\right) \right\|_2^2, \tag{9}$$

with $\left(x^0\right)_{1:N_{\text{ic}}} \in \Omega$ sample at which the initial condition function $f$ is evaluated. $\left(t^b, x^b\right)_{1:N_{\text{bc}}}$ are $N_{\text{bc}}$ points sampled on the boundary $[0, T] \times \partial\Omega$. The two loss terms are then weighted by $\alpha_{\text{PINN}}$ for PDE loss equation 8 and $\gamma_{\text{data}}$ for data-fit equation 9 to be consistent with Akhound-Sadegh et al. (2023). Within the field of neural operators, it is desirable that the trained network generalizes across various initial conditions, and the above mentioned loss is evaluated for a number of initial conditions $N_{f,\text{train}}$.

### D.3 HEAT EQUATION

This and following sections primarily recap the relevant facts regarding the symmetry groups of the PDEs considered, which act to motivate the need for new equivariant architectures due to the non-compactness of the resulting symmetry groups, and the difficulty of constructing global parametrisations of either the action or the group considered.

#### D.3.1 POINT SYMMETRIES

Simplest PDE example is the heat equation, admitting a non-trivial group of Lie point symmetries, beyond space isometries.

$$u_t - \nu u_{xx} = 0 \tag{10}$$

This equation admits the following 6-dimensional Lie algebra (note the typo in Akhound-Sadegh et al. (2023)), see Olver (1993); Koval & Popovych (2023):

$$v_1 = \partial_x, \quad v_2 = \partial_t, \quad v_3 = \frac{1}{\nu} u \partial_u, \quad v_4 = 2t\partial_t + x\partial_x - \frac{1}{2} u\partial_u, \tag{11}$$

$$v_5 = t\partial_x - \frac{1}{2\nu} xu\partial_u, \quad v_6 = t^2\partial_t + tx\partial_x - \frac{1}{4\nu}\left(x^2 + 2\nu t\right) u\partial_u. \tag{12}$$

The resulting one parameter flows are:

$$
\begin{array}{llll}
\exp[v_1](\epsilon): & \tilde{t} = t, & \tilde{x} = x + \epsilon, & \tilde{u} = u, \\
\exp[v_2](\epsilon): & \tilde{t} = t + \epsilon, & \tilde{x} = x, & \tilde{u} = u, \\
\exp[v_3](\epsilon): & \tilde{t} = t, & \tilde{x} = x, & \tilde{u} = e^{\epsilon/\nu} u, \\
\exp[v_4](\epsilon): & \tilde{t} = e^{2\epsilon} t, & \tilde{x} = e^{\epsilon} x, & \tilde{u} = e^{-\frac{1}{2}\epsilon} u, \\
\exp[v_5](\epsilon): & \tilde{t} = t, & \tilde{x} = x + \epsilon t, & \tilde{u} = e^{-\frac{1}{4\nu}\left(\epsilon^2 t + 2\epsilon x\right)} u, \\
\exp[v_6](\epsilon): & \tilde{t} = \frac{t}{1-\epsilon t}, & \tilde{x} = \frac{x}{1-\epsilon t}, & \tilde{u} = \sqrt{|1 - \epsilon t|} e^{\frac{\epsilon x^2}{4\nu(\epsilon t - 1)}} u,
\end{array}
$$

The only non-zero Lie brackets are:

$$[v_4, v_2] = -2v_2, \quad [v_4, v_6] = 2v_6, \quad [v_2, v_6] = v_4, \tag{13}$$

$$[v_2, v_5] = v_1, \quad [v_4, v_5] = v_5, \quad [v_4, v_1] = -v_1, \quad [v_6, v_1] = -v_5, \tag{14}$$

$$[v_5, v_1] = \frac{1}{2} v_3. \tag{15}$$

In particular, these would be enough for LieLAC, however in this particular example it is possible to show more. To be precise, as shown in Koval & Popovych (2023), the global structure, as explained below, of the symmetry group (the finite simply connected component) is $G^H = \text{SL}(2, \mathbb{R}) \ltimes_\phi \text{H}(1, \mathbb{R})$, where $\text{H}(1, \mathbb{R})$ is the rank one polarized Heisenberg group.

**Theorem D.1** (Koval & Popovych (2023) for $\nu \neq 1$). *The point symmetry pseudogroup $G$ of the $(1 + 1)$-dimensional linear heat equation is constituted by the point transformations of the form*

$$\tilde{t} = \frac{\alpha t + \beta}{\gamma t + \delta}, \quad \tilde{x} = \frac{x + \lambda_1 t + \lambda_0}{\gamma t + \delta},$$

$$\tilde{u} = \sigma\sqrt{|\gamma t + \delta|} \exp\left(\frac{\gamma\left(x + \lambda_1 t + \lambda_0\right)^2}{4\nu(\gamma t + \delta)} - \frac{\lambda_1}{2\nu} x - \frac{\lambda_1^2}{4\nu} t\right)(u + h(t, x)),$$

*where $\alpha, \beta, \gamma, \delta, \lambda_1, \lambda_0$ and $\sigma$ are arbitrary constants with $\alpha\delta - \beta\gamma = 1$ and $\sigma \neq 0$, and $h$ is an arbitrary solution of the heat equation.*

We now describe the product on $G$. First we note that there is a subgroup $R$ consisting of those elements for which $\lambda_1 = \lambda_0 = 0$ and $\sigma = 1$. This is evidently isomorphic to $\mathrm{SL}(2, \mathbb{R})$. Therefore we shall group the $\alpha, \beta, \gamma, \delta$ terms into a matrix $A = \begin{pmatrix} \alpha & \beta \\ \gamma & \delta \end{pmatrix}$. Further we note that there are two connected components of this group, corresponding to sign of $\sigma$, and the group can be written as a direct product corresponding to this sign

$$G = G' \times \mathbb{Z}_2 \tag{16}$$

Thus we may simply describe the product on $G'$. For this we may assume that the two elements we multiply have $\sigma > 0$. For clarity, we shall write out the product rule in terms of the logarithm of $\sigma$. The product can now be written as

$$(A', \lambda_1', \lambda_0', \ln \sigma') \cdot (A, \lambda_1, \lambda_0, \ln \sigma) = (A'A, \lambda_1 + \alpha\lambda_1' + \gamma\lambda_0', \lambda_0 + \beta\lambda_1' + \delta\lambda_0',$$
$$\ln \sigma + \ln \sigma' + \frac{1}{4}(\lambda_0'\lambda_1' - (\alpha\lambda_1' + \gamma\lambda_0')(\beta\lambda_1' + \delta\lambda_0')) - \frac{\lambda_0}{2}(\alpha\lambda_1' + \gamma\lambda_0'))$$

We see that the subgroup $F$ consisting of those elements for which $\alpha = \delta = 1, \beta = \gamma = 0$ and $\sigma > 0$ is isomorphic to the rank one polarized Heisenberg group $\mathrm{H}(1, \mathbb{R})$ via

$$(\lambda_1, \lambda_0, \ln \sigma) \mapsto \begin{pmatrix} 1 & -\frac{1}{2}\lambda_1 & \ln \sigma \\ 0 & 1 & \lambda_0 \\ 0 & 0 & 1 \end{pmatrix} \tag{17}$$

where we use the standard embedding of $\mathrm{H}(1, \mathbb{R})$ into the group of $3 \times 3$ real valued matrices. Note, that while the parameterization here looks the same as the one from Koval & Popovych (2023), it is different (resulting in different $\phi$), as in that paper, the Heisenberg group and the polarized Heisenberg group are conflated, making the product rule unclear. Here we fix a specific form of the Heisenberg group as a subgroup of the group of $3 \times 3$ real matrices, with standard matrix multiplication as the product rule. The product formula can then be described in terms of $F$ and $R$ as well. Let $\phi : \mathrm{SL}(2, \mathbb{R}) \to \mathrm{Aut}(\mathrm{H}(1, \mathbb{R}))$ be the antihomomorphism defined by

$$\phi(A) = \begin{pmatrix} \lambda_1 \\ \lambda_0 \\ \ln \sigma \end{pmatrix} \mapsto \begin{pmatrix} \alpha\lambda_1 + \gamma\lambda_0 \\ \beta\lambda_1 + \delta\lambda_0 \\ \ln \sigma + \frac{1}{4}\lambda_0\lambda_1 - \frac{1}{4}(\alpha\lambda_1 + \gamma\lambda_0)(\beta\lambda_1 + \delta\lambda_0) \end{pmatrix} \tag{18}$$

where write $\mathrm{H}(1, \mathbb{R})$ using the identification $F \cong \mathrm{H}(1, \mathbb{R})$. Then we can write

$$G' = \mathrm{SL}(2, \mathbb{R}) \ltimes_\phi \mathrm{H}(1, \mathbb{R}) \tag{19}$$

As the semidirect product with an antihomomorphism is rarely written out in literature, we write out explicitly what this entails. We have that the underlying set of $G'$ is $\mathrm{SL}(2, \mathbb{R}) \times \mathrm{H}(1, \mathbb{R})$ and the product is

$$(A', H') \cdot (A, H) = (A'A, \phi(A)(H')H) \tag{20}$$

In order to decanonicalize, we also require access to the inverse group element. Particularly, for our implementation we can either find the action of the inverse of $\exp(e_6 v_6) \ldots \exp(e_1 v_1)$ as $\exp(-e_1 v_1) \ldots \exp(-e_6 v_6)$, but given access to a global parametrization of the group, we can write down an inverse of a given element in the form of $A, H$. This is $\left(A^{-1}, \phi(A^{-1})\left(H^{-1}\right)\right)$. I.e. for $\alpha, \beta, \gamma, \delta, \lambda_0, \lambda_1, \sigma$, the inverse is

$$A^{-1} = \begin{pmatrix} \delta & -\beta \\ -\gamma & \alpha \end{pmatrix} \quad H^{-1} = \begin{pmatrix} 1 & \frac{1}{2}\lambda_1 & -\ln \sigma - \frac{1}{2}(\lambda_1\lambda_0) \\ 0 & 1 & -\lambda_0 \\ 0 & 0 & 1 \end{pmatrix}$$

Writing this out in coordinates this becomes

$$\delta, -\beta, -\gamma, \alpha, \beta\lambda_1 - \alpha\lambda_0, -\delta\lambda_1 + \gamma\lambda_0, -\ln \sigma - \frac{1}{4}\lambda_0\lambda_1 - \frac{1}{4}(-\delta\lambda_1 + \gamma\lambda_0)(\beta\lambda_1 - \alpha\lambda_0)$$

| Model | Heat (Original) | | Heat (+ data aug.) | | Burgers (Original) | |
|---|---|---|---|---|---|---|
| | ID (21) | OOD (21) | ID (21) | OOD (21) | ID (D.4.2) | OOD (D.4.2) |
| DeepONet | $0.0498 \pm 0.0072$ | $0.6572 \pm 0.1235$ | $0.0504 \pm 0.0014$ | $0.0687 \pm 0.0044$ | $\mathbf{0.0832 \pm 0.0547}$ | $0.8369 \pm 0.0987$ |
| LieLAC [DeepONet] | $\mathbf{0.0443 \pm 0.0027}$ | $\mathbf{0.0435 \pm 0.0017}$ | $\mathbf{0.0500 \pm 0.0003}$ | $\mathbf{0.0500 \pm 0.0003}$ | $0.0916 \pm 0.0632$ | $\mathbf{0.1006 \pm 0.0637}$ |

Table 3: $L_2$ relative error for Heat equation and Burgers' equation averaged over $[0, T]$. Original refers to model trained on in-distribution (ID) data. For heat equation this corresponds to with fixed $A_k^{\text{Train}} = 1$ in Equation (21). Data aug refers to model trained on out of distribution data achieved via symmetry transformations, resulting in $A_k^{\text{Train}} \in \{0.5, 5.0\}$. For testing ID refers to $A_k^{\text{Test}} \in \{0.95, 1.05\}$, while OOD refers to $A_k^{\text{Test}} \in \{0.5, 5.0\}$. For Burgers' equation, ID corresponds to gaussian random fields as in Appendix D.4.2, while OOD refers to ID plus a constant shift.

### D.3.2 NUMERICAL RESULTS

For the numerical experiments, we consider solving Equation (10) with diffusivity $\nu = 0.1$ on the spatial domain $[0, 2\pi]$, with periodic boundary conditions for $t \in [0, 16]$. We train a physics informed DeepONet (Lu et al., 2019; Wang et al., 2021b) with initial conditions of the form following Akhound-Sadegh et al. (2023); Brandstetter et al. (2022)

$$u_0(x) = \sum_{i=1}^{K} A_k \sin\left(2\pi l_k x/L + \phi_k\right). \tag{21}$$

To focus on the effects of data augmentation and canonicalization, we train the model under two regimes. In the first regime (which we refer to as in-distribution), the amplitude parameter is fixed to $A_k = 1$, with the other parameters as $K = 1$, $l_k = 2$, $\phi_k = 0$, analogous to PINN training. In the second regime, the amplitude parameter is sampled from $A \sim \mathcal{U}[0.5, 5.0]$, representing a broader operator training distribution.

The model is then trained using the physics loss with sampling weights of $\alpha_{\text{PINN}} = 150$, $\gamma_{\text{data}} = 20$, we use a learning rate of 0.001 and batch size of 8 and train for 100k epochs following Akhound-Sadegh et al. (2023). Training data consists of $N_{f,\text{train}}$ initial condition functions, evaluated at $N_{\text{ic}} = 200$ points at $t = 0$ to evaluate $u_0$, $N_{\text{bc}} = 100$ points on the boundaries $x = 0$ or $x = 2\pi$ to enforce the periodic boundary conditions and finally $N_{\text{dom}} = 500$ points in the interior of the domain where the residual physics loss is enforced. The official implementation of DeepONet from the code and examples provided by the library DeepXDE (Lu et al., 2021) was used.

We report results for both training regimes, averaged over 10 random seeds, in Table 3. The first rows in both tables show the accuracy of the standard DeepONet. In both cases, the first column shows a slight improvement with LieLAC canonicalizing to the average of the tight distribution. In Table 3 the second column shows DeepONet's failure to generalize when tested on $Nf_{test} = 10$ test functions sampled $A \sim \mathcal{U}[0.5, 5.0]$, which is outside of the training range. Applying LieLAC, following the canonicalization pipeline shown in Figure 2a, restores test accuracy to in-distribution levels. In Table 3, extending the training range to $A \sim \mathcal{U}[0.5, 5.0]$ improves generalization by exposing the model to a broader range of amplitudes. However, even in this setting, LieLAC outperforms data augmentation by leveraging the canonicalizing group action.

**Choice of Energy** Given the initial conditions in Equation (21), we see that the distribution of initial conditions can be described by the bounding box of the jet. The energy is therefore chosen to be the distance to the trained domain. Letting $\text{jet}_0 = (u_0, x_0, t_0)$ and $\text{jet}_\Omega = (1, \Omega, 0)$, the energy is:

$$E_{\text{Heat}}(x_0, t_0, u_0, x_f, t_f) = \text{dist}\left[\max(\text{jet}_0), \max(\text{jet}_\Omega)\right] + \text{dist}\left[\min(\text{jet}_0), \min(\text{jet}_\Omega)\right] \tag{22}$$
$$+ \text{dist}\left[x_f, \Omega\right] + \text{dist}\left[t_f, 16\right]$$

### D.3.3 FURTHER HEAT ILLUSTRATIONS

Figure 9 shows canonicalicalisation of the initial conditions, out-of-distribution and canonicalized DeepONet predictions.

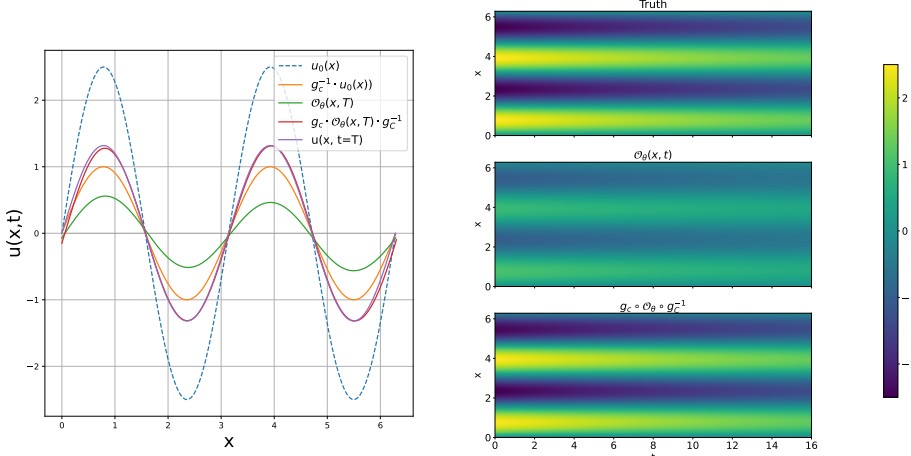

Figure 9: Canonicalization for the Heat equation given out of distribution $u_0(x)$ and improved equivariant prediction

### D.4 BURGERS' EQUATION

#### D.4.1 POINT SYMMETRIES

Burgers' Equation combines diffusion and non-linear advection, which can be related to the linear heat equation above via the Cole–Hopf transformation. The nonlinearity of the equation results in more complex dynamics, e.g. via shock formation, while still admitting non-trivial Lie point symmetries.

$$u_t + uu_x - \nu u_{xx} = 0 \tag{23}$$

This equation admits the following 5-dimensional Lie algebra (note the typo in Akhound-Sadegh et al. (2023)), see Freire (2010); Koval & Popovych (2023):

$$v_1 = \partial_x, \quad v_3 = 2t\partial_t + x\partial_x - u\partial_u, \quad v_5 = t^2\partial_t + tx\partial_x + (x - tu)\partial_u, \tag{24}$$
$$v_2 = \partial_t, \quad v_4 = t\partial_x + \partial_u.$$

The resulting one parameter flows, for $\epsilon$ arbitrary constant, are:

$$
\begin{aligned}
\exp[v_1](\epsilon): \quad & \tilde{t} = t, & \tilde{x} = x + \epsilon, & \quad \tilde{u} = u, \\
\exp[v_2](\epsilon): \quad & \tilde{t} = t + \epsilon, & \tilde{x} = x, & \quad \tilde{u} = u, \\
\exp[v_3](\epsilon): \quad & \tilde{t} = e^{2\epsilon}t, & \tilde{x} = e^{\epsilon}x, & \quad \tilde{u} = e^{-\epsilon}u, \\
\exp[v_4](\epsilon): \quad & \tilde{t} = t, & \tilde{x} = x + \epsilon t, & \quad \tilde{u} = u + \epsilon, \\
\exp[v_5](\epsilon): \quad & \tilde{t} = \frac{t}{1 - \epsilon t}, & \tilde{x} = \frac{x}{1 - \epsilon t}, & \quad \tilde{u} = u(1 - \epsilon t) + \epsilon x,
\end{aligned}
$$

The non-zero brackets are:

$$[v_3, v_2] = -2v_2, \quad [v_3, v_5] = 2v_5, \quad [v_2, v_5] = v_3, \tag{25}$$
$$[v_2, v_4] = v_1, \quad [v_3, v_4] = v_4, \quad [v_3, v_1] = -v_1, \quad [v_5, v_1] = -v_4 \tag{26}$$

Once again, this would be enough for the LieLAC formulation, however in this particular example, we can say more:

**Theorem D.2** (Koval & Popovych (2023) for $\nu \neq 1$)**.** *The point symmetry group $G^{\mathrm{B}}$ of the Burgers' equation consists of the point transformations of the form*

$$\tilde{t} = \frac{\alpha t + \beta}{\gamma t + \delta}, \quad \tilde{x} = \frac{x + \lambda_1 t + \lambda_0}{\gamma t + \delta}, \quad \tilde{u} = (\gamma t + \delta)u - \gamma x + \lambda_1 \delta - \lambda_0 \gamma,$$

*where $\alpha, \beta, \gamma, \delta, \lambda_1$ and $\lambda_0$ are arbitrary constants with $\alpha\delta - \beta\gamma = 1$.*

And similar to before, the global structure of the group can be inferred, with the group $G^{\mathrm{B}}$ isomorphic to the group $\mathrm{SL}(2, \mathbb{R}) \ltimes_{\breve{\varphi}} (\mathbb{R}^2, +)$, where $\breve{\varphi} : \mathrm{SL}(2, \mathbb{R}) \to \mathrm{Aut}\left((\mathbb{R}^2, +)\right)$ is the group antihomomorphism defined by $\breve{\varphi}(\alpha, \beta, \gamma, \delta) = (\lambda_1, \lambda_0) \mapsto (\lambda_1, \lambda_0)\varrho_1(\alpha, \beta, \gamma, \delta)$. Thus, the group $G^{\mathrm{B}}$ is connected.

$$\varrho_1 = (\alpha, \beta, \gamma, \delta)_{\alpha\delta-\beta\gamma=1} \mapsto \begin{pmatrix} \alpha & \beta \\ \gamma & \delta \end{pmatrix}, \quad (\lambda_1, \lambda_0) \mapsto (\lambda_1, \lambda_0).$$

The standard conjugacy action of the subgroup $\breve{F} < G^{\mathrm{B}}$ on the normal subgroup $\breve{R} \triangleleft G^{\mathrm{B}}$ is given by $(\tilde{\lambda}_1, \tilde{\lambda}_0) = (\lambda_1, \lambda_0)\,\varrho_1(\alpha, \beta, \gamma, \delta)$. The inverse can be found similar to Appendix D.3.

### D.4.2 NUMERICAL RESULTS

For the numerical experiments, we consider solving Equation (23) on the domain $x \in \Omega = [0, 1]$, with periodic boundary conditions on $t \in [0, 1]$. An example of a canonicalized solution is shown in Figure 2, with further illustrations in Figures 10 and 11.

For Burgers' equation we take the code and pretrained Physics Informed DeepONet (PI-DeepONet) model weights provided by Wang et al. (2021b). This includes a model trained on $N_{f,\mathrm{train}} = 1000$ initial conditions sampled from a Gaussian random field (GRF) parameterised as $\mathcal{N}\left(0, 25^2(-\Delta + 5^2 I)^{-4}\right)$ on the periodic domain $\Omega = [0, L] \times [0, T] = [0, 1] \times [0, 1]$ discretized by a meshgrid with $101 \times 101$ points.

In order to sample out-of-distribution initial conditions from the GRF we apply a uniform 0.2 shift to $u_0$ away from the mean of 0. Figure 11 shows surprisingly even this small perturbation is enough to break the pre-trained PI DeepONet. Interestingly in both in and out of distribution cases LieLAC learns a group action that is able to redistribute around zero mean, leading to a more accurate prediction.

**Choice of Energy** Given the slightly different form of initial conditions to the heat equation, now sampled as GRFs we adjust the energy to focus on the bounding box of the domain and mean of $u_0$. The energy is therefore chosen to be the distance to the training domain. Letting $\mathrm{trunk}_0 = (x_0, t_0)$ and $\mathrm{trunk}_\Omega = (\Omega, 0)$, the energy is:

$$\begin{aligned} E_{\mathrm{Burgers}}(x_0, t_0, u_0, x_f, t_f) = {} & \mathrm{dist}\left[\max(x_0) - \min(x_0), 1)\right] \\ & + \mathrm{dist}\left[t_0, \Omega\right] + \mathrm{dist}\left[\mathrm{mean}[u_0], 0\right] \\ & + \mathrm{dist}\left[x_f, \Omega\right] + \mathrm{dist}\left[t_f, \Omega\right] \end{aligned} \tag{27}$$

### D.4.3 FURTHER BURGERS' ILLUSTRATIONS

Figure 10 shows canonicalicalisation of the initial conditions, out-of-distribution and canonicalized DeepONet predictions.

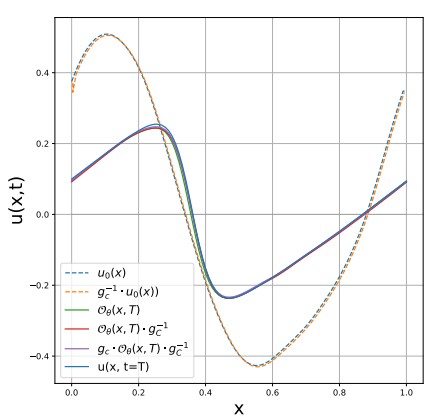
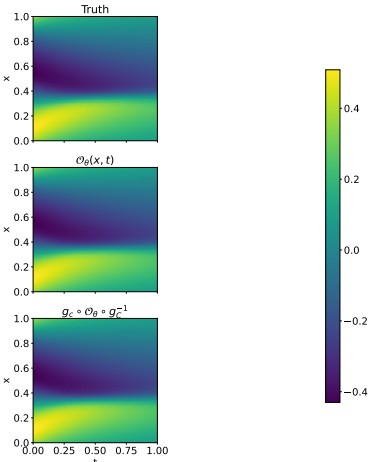

Figure 10: Illustration of Burgers' equation for in distribution GRF $u_0(x)$

Figure 11 shows canonicalicalisation of the initial conditions, out-of-distribution and canonicalized DeepONet predictions.

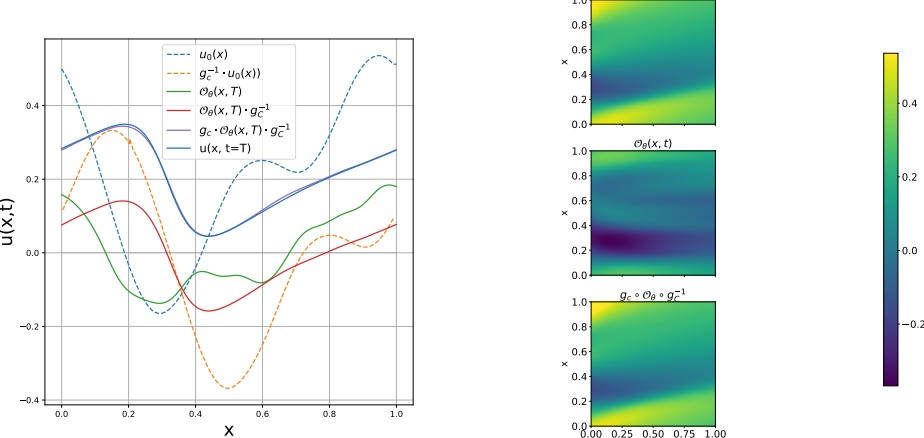

Figure 11: Canonicalization for Burgers' equation given out of distribution $u_0(x)$ and improved equivariant prediction

## D.5 ALLEN-CAHN EQUATION

The Lie point symmetries of Equation (4) can be found either by hand or using the computational software of Baumann (1998). Both yield a 4-dimensional Lie algebra with basis vectors:

$$v_1 = \partial_t, \quad v_2 = \partial_x, \quad v_3 = \partial_y, \quad v_4 = -y\partial_x + x\partial_y \tag{28}$$

Noting that there is no coupling between the space, time and the dependent components, we can identify this group as the connected component of the group of isometries of the underlying space. It is worth noting here that the full symmetry group can be observed to be $E(2)$, while the group generated by vectors in 28 is $SE(2)$, i.e. the identity connected component of $E(2)$.

**Training data** The POSEIDON (Herde et al., 2024) model is trained on random initial conditions[4] in the form of:

$$u_0(x, y) = \sum_{i,j=1}^{K} a_{ij} \cdot \left(i^2 + j^2\right)^{-r} \sin(\pi i x) \sin(\pi j y), \quad \forall x, y \in (0, 1), \tag{29}$$

where $K$ is an integer drawn uniformly at random from $[16, 32]$, and $r \sim U_{[0.7, 1.0]}$ and $a_{ij} \sim U_{[-1,1]}$. Based on this, one obvious limitation of the training data is that all sampled initial conditions are zero on the boundary. However, due to the semi-group property based training of Herde et al. (2024), as trajectories are sampled at different points, the values on the boundary are not guaranteed to remain zero over time. As a result, we expect the model to not be equivariant under the group of symmetries of ACE, as illustrated on Figure 12, but we do not have an obvious way to construct the energy. Based on the positive results in Section 4.2, we employ the same approach, by training a variational autoencoder and an adversarial regulariser. For the experiments, we considered the TINY POSEIDON model from Herde et al. (2024).

**Testing data** To test the equivariance, we use a dataset of modified initial conditions, achieved by transformations of Equation (29).

$$u_0(x, y) = \sum_{i,j=1}^{K} a_{ij} \cdot \left(i^2 + j^2\right)^{-r} \sin(\pi i \{x - x_0\}) \sin(\pi j \{y - y_0\}), \quad \forall x, y \in (0, 1), \tag{30}$$

where $\{\cdot\}$ denotes the sawtooth function, with parameters as in Equation (29), with $x_0, y_0 \sim U_{[0,1]}$.

---

[4]Note here that this is slightly different from the original paper. The form below was chosen, as the generated initial conditions provided by (Herde et al., 2024) either due to pre or post-processing turn out to not be periodic.

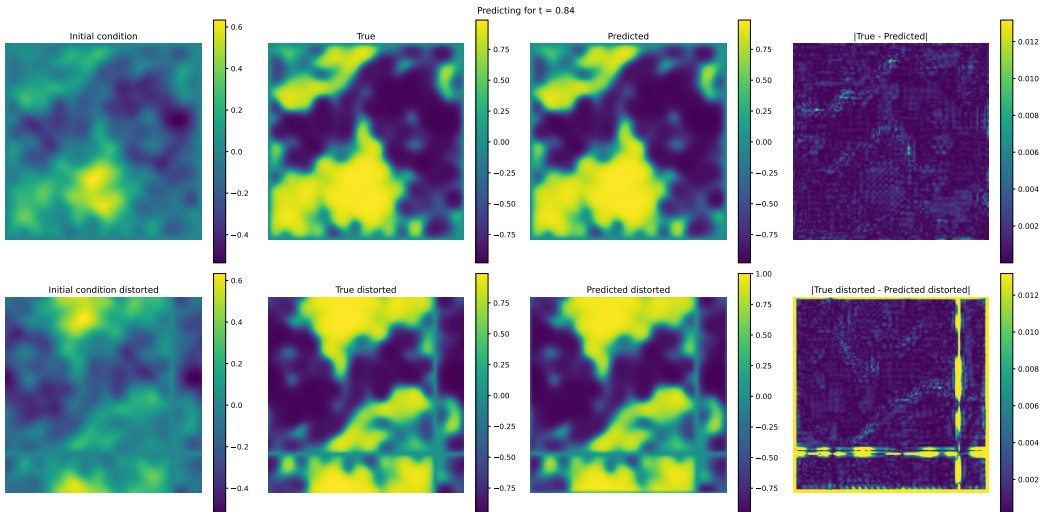

Figure 12: Example of failure of equivariance of the original POSEIDON model.

### D.5.1 FURTHER ACE ILLUSTRATIONS

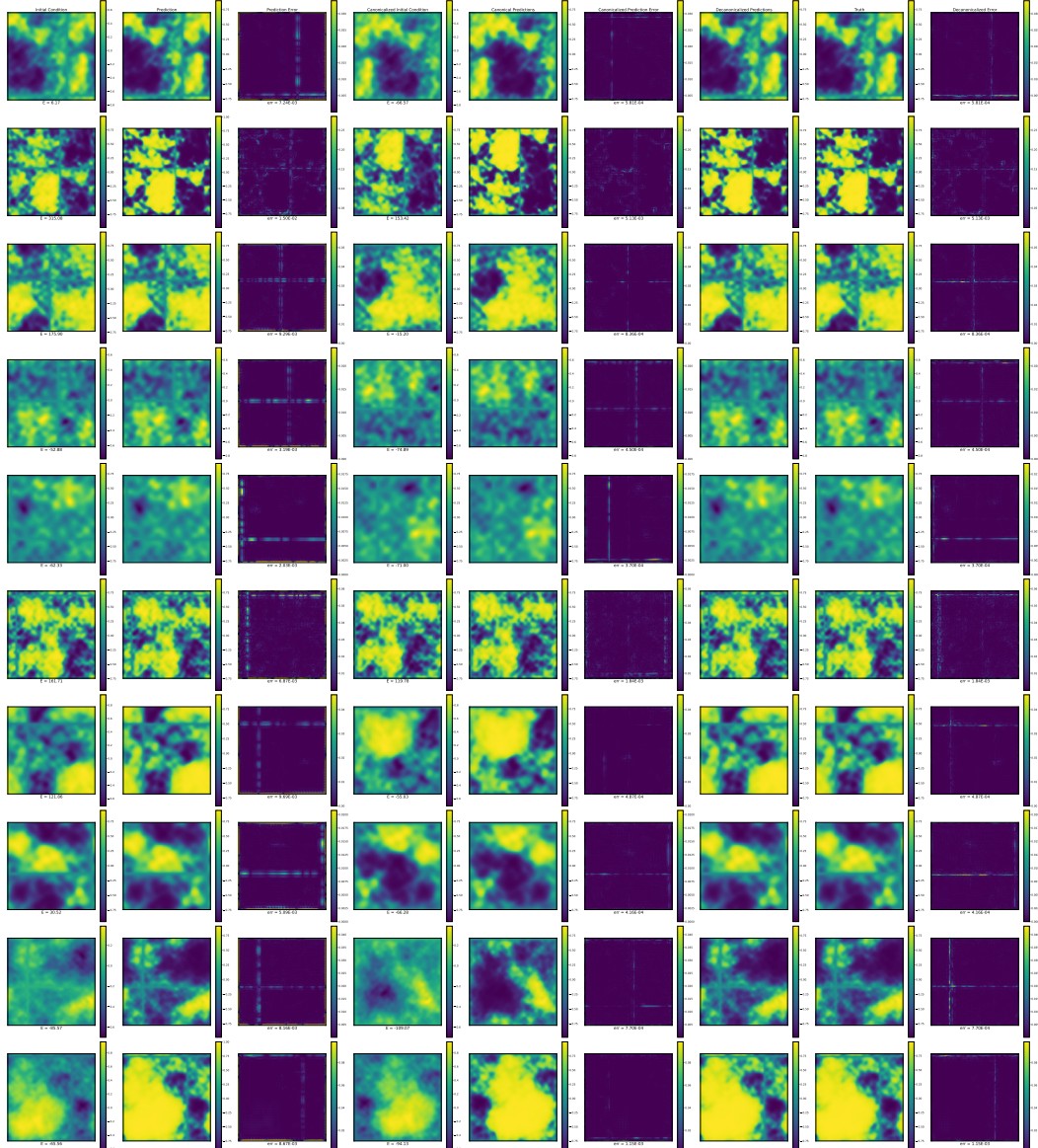

Figure 13: Further canonicalization illustration for the ACE.

# E  MEASURE THEORY BACKGROUND

Let $X$ be a set. Then a $\sigma$-algebra $\Sigma$ is a subset of $2^X$ such that

- $X \in \Sigma$
- If $A \in \Sigma$ then $X \backslash A \in \Sigma$
- For any countable collection $(A_n)_{n=1}^\infty$ of sets in $\Sigma$, the union $\bigcup_{n=1}^\infty A_n \in \Sigma$

A set $A \in \Sigma$ is called measurable. A pair $(X, \Sigma)$ of a set and a $\sigma$-algebra is called a measure space. Suppose $(X, \Sigma), (Y, \Sigma')$ are measure spaces. Then a function $f : X \to Y$ is measurable if for every measurable set $A \in \Sigma'$ the set $f^{-1}(A)$ is measurable.

Given a measure space, a measure on it is a set function $\mu : \Sigma \to [0, \infty]$ such that

- $\mu(\varnothing) = 0$
- For all countable collections $(A_n)_{n=1}^\infty$ of pairwise disjoint sets in $\Sigma$

$$\mu(\bigcup_{n=1}^\infty A_n) = \sum_{i=1}^\infty \mu(A_n) \tag{31}$$

We may also define signed and complex measures, which simply replace the codomain of a measure with either $\mathbb{R} \cup \{-\infty, +\infty\}$ or $\mathbb{C}$ respectively.

If $f : (X, \Sigma) \to (Y, \Sigma')$ is measurable, and $\mu$ is a measure on $(X, \Sigma)$, we define the pushforward measure $f_*\mu$ as follows

$$f_*\mu(A) = \mu(f^{-1}(A)) \tag{32}$$

There is always a measure on $(X, \Sigma)$ which simply sends every measurable set to $0$, we call this the zero measure. A finite measure is a measure $\mu$ such that $\mu(X) < \infty$. A probability measure is a measure $\mu$ such that $\mu(X) = 1$. Given a finite non-zero measure $\mu$ on $(X, \Sigma)$ we may always define a probability measure $\mu'$ on $(X, \Sigma)$ as follows

$$\mu'(A) = \frac{\mu(A)}{\mu(X)} \tag{33}$$

We will often call this the normalized measure associated to $\mu$.

If $X$ is a topological space there is a canonical $\sigma$-algebra associated to it, known as the Borel $\sigma$-algebra $\mathcal{B}(X)$. It is the smallest $\sigma$-algebra such that all open sets are measurable. Then any continuous function between $X$ and $Y$ is measurable when $X$ and $Y$ are equipped with their respective Borel $\sigma$-algebras. We often call the pair $(X, \mathcal{B}(X))$ a Borel space and when clear we will simply refer to this pair by $X$.

For a topological space $S$ and a real number $s \geq 0$ there is an associated Hausdorff measure of dimension $s$ which we shall denote $\mathcal{H}_S^s$. To $S$ we can also define what is known as the Hausdorff dimension, $\dim_{\mathcal{H}}(S)$, which is always non-negative. There is then a Hausdorff measure of that dimension, which we write as $\mathcal{H}_S$. All Borel sets are measurable for this measure. If $S$ is compact then $\mathcal{H}_S(S) < \infty$, but it may still be the zero measure. If it is not zero then we may as above construct the normalized Hausdorff measure. In this case this will be a Radon measure. For $S \cong [0, 1]^n$ the Hausdorff dimension is $n$ and the normalized Hausdorff measure is exactly the Lebesgue measure. Similarly for $S$ finite and discrete, the Hausdorff dimension is $0$ and the normalized Hausdorff measure is the normalized counting measure.

The Hausdorff measure is important to consider beacuse of Frostman's Lemma Mattila (1995).

**Lemma E.1** (Frostman's Lemma). *Let $A$ be a Borel subset of $\mathbb{R}^n$ and let $s > 0$. Then the following are equivalent.*

- *$\mathcal{H}^s(A) > 0$*

- *There is an unsigned Borel measure $\mu$ on $\mathbb{R}^n$ satisfying $\mu(A) > 0$ and such that*

$$\mu(B(x, r)) \leq r^s \tag{34}$$

  *for all $x \in \mathbb{R}^n, r > 0$.*

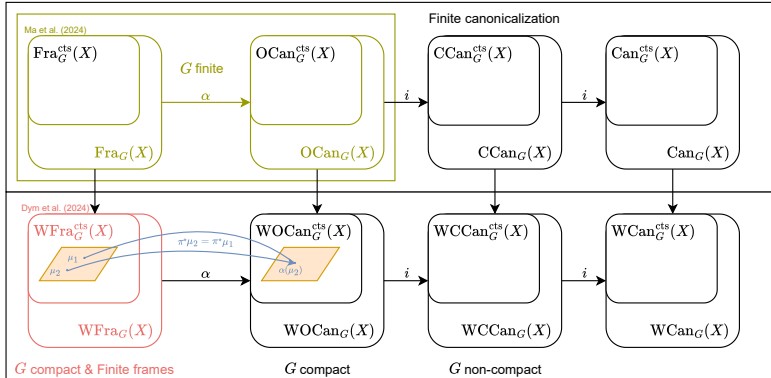

Figure 14: This diagram illustrates connections between the various notions of frames and canonicalizations introduced previously and in this work. The top row are the finite frames and canonicalizations, which all map vertically into their weighted versions by taking normalized counting measures. Inside of each we have the sequentially closed subspaces of those that perserve continuity.

Let $A$ be compact with Hausdorff dimension greater than zero. Then we know that the Hausdorff measure at the Hausdorff dimension is finite. By Frostman's Lemma, the Hausdorff measure of $A$ being zero means that we cannot find any probability measure on $A$ that interacts nicely with the metric structure. This shows that our assumption of not allowing sets whose Hausdorff measure is the zero measure is reasonable.

## F    MORE ON FRAMES AND CANONICALIZATION

Recall that a frame is defined as a function $\mathcal{F} : X \to 2^G \backslash \varnothing$ such that for $x \in X$ and $g \in G$

$$\mathcal{F}(gx) = g\mathcal{F}(x) \tag{35}$$

**Lemma F.1.** *Let $\mathcal{F}$ be frame for the action of a group $G$ on a set $X$. Then $\forall x \in X$, $\mathcal{F}(x)$ is a union of left cosets of $G_x \subseteq G$.*

*Proof.* Indeed, let $x \in X$ and let $g \in G_x$. Then

$$\mathcal{F}(x) = \mathcal{F}(gx) = g\mathcal{F}(x) \tag{36}$$

Hence we see that

$$\mathcal{F}(x) = G_x\mathcal{F}(x) \tag{37}$$

Therefore we see that $\mathcal{F}(x)$ is a union of left cosets of $G_x$. $\qquad\square$

### F.1    WEAKLY-EQUIVARIANT FRAMES

In Dym et al. (2024) weakly equivariant frames are introduced. These are maps $\mu : X \to \mathrm{PMeas}(G)$ satisfying equivariance up to averaging over stabilizers.

In the review process it was noted that this bears similarity to Nguyen et al., and indeed, in our framework their method is exactly the weighted frame defined by minimizing the energy given by minimizing orbit distance. Because the symmetry groups they consider are compact and act algebraically, they are able to obtain more concrete descriptions of these frames than we are able to in the general case. We do expect that in the compact case with algebraic actions, similar results should hold for energy minimizing canonicalizations.

First, for any $x \in X$ and Radon probability measure $\tau$ Dym et al. (2024) define for every measurable $A \subseteq G$,

$$\langle \tau \rangle_x(A) = \int_{s \in G_x} s_* \tau(A) ds \tag{38}$$

where $ds$ is the normalized Haar measure on $G_x$. For a frame $\mu$ we let $\overline{\mu}_x = \langle \mu_x \rangle_x$. Then $\mu$ is weakly equivariant if for all $x \in X$ and $g \in G$,

$$\overline{\mu}_{gx} = g_* \overline{\mu}_x \tag{39}$$

This definition does not extend to the case where the action has non-compact stabilizers.

To deal with this we propose imposing what we call $\pi_*$-equivariance. Again we consider simply a map $\mu : X \to \mathrm{PMeas}(G)$. For a point $x \in X$, let $\pi_x : G \to G/G_x$ be the quotient. We require that

$$(\pi_{gx})_* \mu_{gx} = (\pi_{gx})_* g_* \mu_x \tag{40}$$

**Lemma F.2.** *If the stabilizer $G_x$ is compact then $\langle \mu \rangle_x = \langle \mu' \rangle_x$ if and only if $(\pi_x)_* \mu = (\pi_x)_* \mu'$.*

*Proof.* For ease of notation let $\pi = \pi_x$. First suppose that $\langle \mu \rangle_x = \langle \mu \rangle_x$. Let $A \subseteq G/G_x$ be a measurable set. Note that $\pi^{-1}(A)$ is a union of cosets of $G_x$. Hence it is invariant under multiplication by elements in $G_x$. Therefore for all $s \in G_x$ we have $s^{-1}\pi^{-1}(A) = \pi^{-1}(A)$. Thus

$$\langle \mu \rangle_x(A) = \int_{s \in G_x} s_* \mu(\pi^{-1}(A)) ds = \int_{s \in G_x} \mu(\pi^{-1}(A)) ds = \mu(\pi^{-1}(A)) \tag{41}$$

and similarly, $\langle \mu' \rangle_x(\pi^{-1}(A)) = \mu'(\pi^{-1}(A))$. Hence,

$$\mu(\pi^{-1}(A)) = \mu'(\pi^{-1}(A)) \tag{42}$$

Therefore, $\pi_* \mu = \pi_* \mu'$.

Now suppose that $\pi_* \mu = \pi_* \mu'$. Then for any measurable set $A \subseteq G$, we have that

$$\mu(\pi^{-1}(\pi(A))) = \pi_* \mu(\pi(A)) = \pi_* \mu'(\pi(A)) = \mu'(\pi^{-1}(\pi(A)) \tag{43}$$

We have

$$\pi^{-1}(\pi(A)) = \bigcup_{s \in G_x} sA \tag{44}$$

Using Fubini's theorem and the fact that both $s$ and $\mu$ are probability measure, we may write Equation (38) as follows

$$\langle \mu \rangle_x = \int_{s \in G_x} s_* \mu(A) ds \tag{45}$$

$$= \int_{s \in G_x} \int_G \chi_A(sg) \, d\mu(g) ds \tag{46}$$

$$= \int_G \int_{s \in G_x} \chi_A(sg) \, ds d\mu(g) \tag{47}$$

$$\tag{48}$$

Consider the inner integral as a function $\phi$ on $G$. We claim that $\phi$ is $G_x$-invariant. Indeed, let $g' = s'g$ for some $s' \in G_x$. Then

$$\phi(g') = s(Ag^{-1}(s')^{-1}) = s(Ag^{-1}) = \phi(g) \tag{49}$$

as $s$ is the normalized Haar measure, hence is right-invariant. Therefore $\phi$ factors through the quotient map $\pi$. Let $\phi'$ be the corresponding function on $G/G_x$, then $\phi = \phi' \circ \pi$. Hence

$$\langle \mu \rangle_x = \int_G \phi(g) d\mu(g) \tag{50}$$

$$= \int_{G/G_x} \phi'([g]) d(\pi_* \mu)([g]) \tag{51}$$

$$= \int_{G/G_x} \phi'([g]) d(\pi_* \mu')([g]) \tag{52}$$

$$= \int_G \phi(g) d\mu'(g) = \langle \mu' \rangle_x \tag{53}$$

$\square$

**Corollary F.2.1.** *If the $G$-action on $X$ has compact stabilizers at each point, then $\pi_*$-equivariance is equivalent to weak equivariance.*

## F.2 ON PROJECTIONS

For a weighted orbit canonicalization $\mu$, $\mathcal{P}_\mu : B(X, \mathbb{R}) \to B(X, \mathbb{R})^G$ turns out to be a projection, as

$$
\begin{aligned}
\mathcal{P}_\mu \mathcal{P}_\mu(\phi)(x) &= \int_X \mathcal{P}_\mu(\phi)(s) \, d\mu_x(s) \\
&= \int_{Gx} \underbrace{\mathcal{P}_\mu(\phi)(s)}_{\text{G-invariant}} \, d\mu_x(s) = \mathcal{P}_\mu(\phi)(x)
\end{aligned}
\tag{54}
$$

## F.3 ENERGIES AND CLOSED CANONICALIZATIONS

By the construction of energy minimizing frames, we have a function from energy functions to weighted closed canonicalizations. We now describe the image of this.

**Theorem F.3.** *Let $\kappa \in \mathrm{WCCan}_G(X)$ satisfying the following:*

- *$\kappa_x$ is the normalized Hausdorff measure on its support.*

- *For all $y \in \overline{Gy}$, $\mathrm{supp}\, \kappa_y = \mathrm{supp}\, \kappa_x \cap \overline{Gx}$.*

*Then there is an energy function $E$ such that $\kappa_E = \kappa$.*

*Proof.* Suppose $\kappa$ is a closed canonicalization satisfying the requirements of the theorem. Let $E_\kappa(x) = 1 - \chi_{\mathrm{supp}\, \kappa_x}(x)$.

Now we consider the minimum sets

$$
\mathcal{M}_{E_\kappa}(x) = \min_{y \in \overline{Gx}} E_\kappa(y)
\tag{55}
$$

We have that $\mathrm{supp}\, \kappa_x$ is a non-empty closed subset of $\overline{Gx}$, hence the minimum of $E_\kappa$ over $\overline{Gx}$ is zero. Therefore $\mathcal{M}_{E_\kappa}(x) = E_\kappa^{-1}(0) \cap \overline{Gx}$. In other words it is those $y \in \overline{Gx}$ such that $y \in \mathrm{supp}\, \kappa_y$. Further, by assumption, for any $y \in \overline{Gx}$,

$$
\mathrm{supp}\, \mu_y = \mathrm{supp}\, \kappa_x \cap \overline{Gy}
\tag{56}
$$

Therefore $y \in \mathrm{supp}\, \kappa_y$ if and only if $y \in \mathrm{supp}\, \kappa_x$. Hence we see that

$$
\mathcal{M}_{E_\kappa}(x) = \mathrm{supp}\, \kappa_x
\tag{57}
$$

But the by the assumption that $\kappa_x$ is simply the normalized Hausdorff measure on $\mathrm{supp}\, \kappa_x$, we get that the $\kappa_{E_\kappa} = \kappa$. $\qquad\square$

# G PROOFS

**Proposition G.1.** *$\mathcal{F}_E$ defines an* equivariant frame, *i.e. $\mathcal{F}_E(gx) = g\mathcal{F}_E(x)$ for any $g \in G$, $x \in X$.*

*Proof.* Suppose that $x \in \mathcal{X}$ and $g \in G$. Then

$$
\begin{aligned}
\mathcal{F}_E(gx) &= \arg\min_{h \in G} E(h^{-1}gx) \\
&= \arg\min_{gh \text{ s.t. } h \in G} E((gh)^{-1}gx) \\
&= \arg\min_{gh \text{ s.t. } h \in G} E(h^{-1}x) \\
&= g[\arg\min_{h \in G} E(h^{-1}x)] = g\mathcal{F}(x).
\end{aligned}
$$

Here, we have simply used that group multiplication is bijective when either of the arguments is fixed. $\qquad\square$

**Proposition G.2.** *$\mathcal{M}_E(x)$ is G-invariant and non-empty.*

*Proof.* $\mathcal{M}_E(x)$ is $G$-invariant, as the infimum is taken over the orbit $\overline{Gx}$, and this orbit is $G$-invariant. By the coercivity assumption on $E$, this is non-empty. Indeed, let $x \in X$ and let $N$ be large. Let $K$ be a compact set of $X$ such that $E(y) > N$ for $y \in X \backslash K$. Taking $N$ large enough, we can assume that $K \cap \overline{Gx}$ is non-empty. Then we see that $\mathcal{M}_E(x) \subseteq K \cap \overline{Gx}$. $K \cap \overline{Gx}$ is a closed subset of the compact set $K$, hence is compact. Therefore $E$ attains its minimum on this set, hence $\mathcal{M}_E(x)$ is non-empty. $\square$

**Theorem G.3.** *Let $G$ be a Lie group acting smoothly on $X$ a manifold. Let $j_x : G \to X$ be defined as $g \mapsto g^{-1} \cdot x$. Then we have a map $\alpha : \mathrm{WFra}_G(X) \to \mathrm{WOCan}_G(X)$ defined by*

$$\alpha(\mathcal{F})(x) = (j_x)_*(\mathcal{F}(x)) \tag{58}$$

*Then for any $\mathcal{F}, \mathcal{F}' \in \mathrm{WFra}_G(X)$ such that $\alpha(\mathcal{F}) = \alpha(\mathcal{F}')$ we have that $(\pi_x)_*\mathcal{F} = (\pi_x)_*\mathcal{F}'$ for every $x \in X$ where $\pi_x : G \to G/G_x$. If each $G_x$ has a probability measure $P_x$, then $\alpha$ has image exactly the set of weighted orbit canonicalizations with compact support.*

*Proof.* First we check that $\alpha$ as defined above lands in $\mathrm{WOCan}_G(X)$. Let $\mathcal{F} \in \mathrm{WFra}_G(X)$. Then $\alpha(\mathcal{F})(x)$ is $G$-invariant as for any measurable $A \subseteq X$.

$$\begin{aligned}
\alpha(\mathcal{F})(gx)(A) &= (j_{gx})_*(\mathcal{F}(gx))(A) \\
&= \mathcal{F}(gx)(\{h \in G \mid h^{-1} \cdot (gx) \in A\} \\
&= \mathcal{F}(gx)(\{h \in G \mid h^{-1} \cdot (gx) \in A\}) \\
&= \pi_*(\mathcal{F}(gx))(\pi(\{h \in G \mid h^{-1} \cdot (gx) \in A\})) \\
&= \pi_* g_*(\mathcal{F}(x))(\pi(\{h \in G \mid h^{-1} \cdot (gx) \in A\})) \\
&= g_*(\mathcal{F}(x))(\{h \in G \mid h^{-1} \cdot (gx) \in A\}) \\
&= \mathcal{F}(x)(\{h \in G \mid (gh)^{-1} \cdot (gx) \in A\}) \\
&= \mathcal{F}(x)(\{h \in G \mid h^{-1} \cdot x \in A\}) = \alpha(\mathcal{F})(x)(A)
\end{aligned}$$

Thus we see that $\alpha(\mathcal{F}) \in \mathrm{WCan}_G(X)$, as $\alpha(\mathcal{F})(x)(X) = \mathcal{F}(x)(G) = 1$, as $\mathcal{F}(x)$ is a probability measure. To check that it is a weighted orbit canonicalization, first note that it is clear that $\mathrm{supp}\, \alpha(\mathcal{F})(x) \subseteq \overline{Gx}$, as for any $y \notin \overline{Gx}$ we can find an open neighborhood $U$ of $y$ such that $j_x^{-1}(U) = \varnothing$ which is always $\mathcal{F}(x)$-measure zero. Suppose that $y \in \overline{Gx} \backslash Gx$. $\mathcal{F}(x)$ is a weighted frame, hence it is supported on some compact set $K \subseteq G$. We claim that there is an open set $U$ containing $y$ such that $j_x^{-1}(U) \subseteq G \backslash K$. Suppose not, then for every open neighborhood $U$ of $y$ there is an $g_U \in K$ such that $j_x(g) \in U$. Then the collection $(g_U)$ over the neighborhoods of $y$ forms a net in $K$. As $K$ is compact, this net must have a convergent subsequence. Call this subsequence $(g_n)$ and let $g$ be the limit. As the action of $G$ on $X$ is continuous we have

$$g \cdot x = \lim_{n \to \infty} g_n \cdot x \tag{59}$$

For any $U$ an open neighborhood of $y$ we may, by definition of a net, find an $N$ such that for any $n \geq N$, $g_n \cdot x \in U$. As $X$ is Hausdorff, we then must have that the limit $\lim g_n \cdot x$ is simply $y$. But this contradicts $y \notin Gx$. Hence there is some open neighborhood $U$ of $y$ such that $j_x^{-1}(U) \subseteq G \backslash K$. Then $j_x^{-1}(U)$ is measure zero with respect to $\mathcal{F}(x)$. Hence $\alpha(\mathcal{F})(x)(U) = 0$, so $y \notin \mathrm{supp}\, \alpha(\mathcal{F})$. Therefore we see that $\alpha(\mathcal{F}) \in \mathrm{WOCan}_G(X)$.

Suppose that $\mathcal{F}, \mathcal{F}'$ are weighted frames such that $\alpha(\mathcal{F}) = \alpha(\mathcal{F}')$. Let $x \in X$. Then for any measurable $A \subseteq X$,

$$\mathcal{F}(x)(j_x^{-1}(A)) = \alpha(\mathcal{F})(x)(A) = \alpha(\mathcal{F}')(x)(A) = \mathcal{F}'(x)(j_x^{-1}(A)) \tag{60}$$

Both $\mathcal{F}$ and $\mathcal{F}'$ are compactly supported, say on compact subsets $K, K'$ respectively. Consider the natural map $\overline{j_x} : G/G_x \to X$. This is injective for group-theoretic reasons. Let $\pi : G \to G/G_x$ be the quotient map. Then $\pi(K)$ and $\pi(K')$ are compact subsets of $G/G_x$, as they are continuous images of compacts. Now as $\pi(K)$ and $\pi(K)'$ are compacts which inject into a Hausdorff space $X$, $\overline{j_x}$ restricts to homeomorphisms $\pi(K) \cong \overline{j_x}(\pi(K))$ and similar for $K'$. Using Equation (60) we see that for any $U \subseteq G/G_x$ open we have

$$\mathcal{F}(x)(\pi^{-1}(U)) = \mathcal{F}(x)(\pi^{-1}(U)) \tag{61}$$

Therefore $\pi_*\mathcal{F}$ and $\pi_*\mathcal{F}'$ agree on the open sets of $G/G_x$, but then that implies that they agree on the entire Borel $\sigma$-algebra of $G/G_x$. Hence $\pi_*\mathcal{F} = \pi_*\mathcal{F}'$ as desired.

Finally, let $v \in \mathrm{WOCan}_G(X)$ have compact support. Let $x \in X$. Then $v_x$ is some probability measure on $X$ with $\mathrm{supp}\, v_x \subseteq K \subseteq Gx$ for some compact $K \subseteq Gx$. By Theorem A.2 we have that $\overline{j_x} : G/G_x \to X$ is a immersion with image $Gx$. $\overline{j_x}$ restricts to a homeomorphism on $K$, and using this we can define a measure $v'_x$ on $G/G_x$ which agrees with $v_x$ under this homeomorphism. It is supported on the preimage $K'$ of $K$ which is also compact. $\pi : G \to G/G_x$ is a principal bundle with fiber $G_x$. Therefore, locally on $G/G_x$, $\pi$ is the trivial fibration. We may cover $K'$ by finitely many open sets over which $\pi$ is trivial, call them $U_1, \ldots, U_n$. Let $V_i = \pi^{-1}(U_i) \cong U_i \times G_x$. Now define

$$V_i' = V_i \backslash (\bigcup_{j<i} V_j) \tag{62}$$

Then these are disjoint measurable sets which cover $\pi^{-1}(K')$. They are all trivially fibered over the base. Now we define a measure $\nu$ using these, $\nu$ will be supported on $\pi^{-1}(K')$. Let $A$ be a measurable set in $G$. Firstly we may assume that $A \subseteq \pi^{-1}(K')$. Let $A_i = A \cap V_i'$. Then the $A_i$ are disjoint measurable sets which cover $A$. So the value of $\nu$ on $A$ is determined by the value of $\nu$ on each $A_i$. We define this value to be

$$\nu(A_i) = (v'_x \times P_x)(A_i) \tag{63}$$

where here we use the homeomorphism $V_i' \cong \pi(V_i') \times G_x$ to put the product measure on this set. $\pi_*\nu = v'_x$, and this is equivariant as $G_{gx} = g^{-1}G_x$. Hence $\nu$ is $\pi_*$-equivariant. $\qquad\square$

Note that for the above theorem to work, we need to assume that each $G_x$ has a probability measure. This is automatic in the case that each $G_x$ is compact, as in that case we may take that normalized Haar measure on it. It will also hold in the case that each $G_x$ has non-zero Hausdorff measure.

**Theorem G.4.** *The sequential closure of* $\mathrm{WOCan}_G(X)$ *is* $\mathrm{WCCan}_G(X)$

*Proof.* First we show $\mathrm{WCCan}_G(X)$ is sequentially closed. Suppose that $(f^n)$ is a sequence in $\mathrm{WCCan}_G(X)$ converging to $f \in \mathrm{WCan}_G(X)$. Fix $x \in X$ and $y \in X \backslash \overline{Gx}$. Then it is sufficient to show that $y \notin \mathrm{supp}\, f_x$. As $X$ is normal Hausdorff, by Urysohn's lemma there exists continuous $\phi : X \to [0,1]$ such that $\phi(\overline{Gx}) = 1$ and $\phi(\{y\}) = 0$.

As $f^n \to f$, for any $\varepsilon > 0$, we can find $N$ such that for any $n \geq N$ we have the following

$$\|f_n(\phi) - f(\phi)\| < \varepsilon \tag{64}$$

We may easily compute $f_n(\phi)(x)$ by the fact that $\mathrm{supp}\, f_n(x) \subseteq \overline{Gx}$.

$$f_n(\phi)(x) = \int_{\overline{Gx}} \phi\, df_x^n = 1 \tag{65}$$

Similarly,

$$f_n(\phi)(y) = \int_{\overline{Gy}} \phi\, df_y^n = 0 \tag{66}$$

Hence, by taking the limit we compute that $f(\phi)(x) = 1$ and $f(\phi)(y) = 0$. Let $U = \phi^{-1}([0, 1/2])$, this is an open neighborhood of $y$.

$$1 = f(\phi)(x) = \int_X \phi\, df_x \tag{67}$$

$$= \int_{X \backslash U} \phi\, df_x + \int_U \phi\, df_x \tag{68}$$

$$\leq f_x(X \backslash U) + \frac{1}{2} f_x(U) \tag{69}$$

$$= 1 - f_x(U) + \frac{1}{2} f_x(U) \tag{70}$$

$$= 1 - \frac{1}{2} f_x(U) \tag{71}$$

Hence, $f_x(U) = 0$. Therefore $y \notin \mathrm{supp} f_x$.

Clearly, $\mathrm{WOCan}_G(X) \subseteq \mathrm{WCCan}_G(X)$, hence the sequential closure of $\mathrm{WOCan}_G(X)$ is contained inside $\mathrm{WCCan}_G(X)$. We now show the reverse inclusion. Fix an arbitrary metric on $X$, this is possible as $X$ is assumed metrizable.

Suppose we have a weighted closed canonicalization $f$. Let $h$ be the left invariant Haar measure on $G$. Let $\mu_x = (i_x)_* h$, this is a measure supported on $\overline{Gx}$. Further it is a $\sigma$-finite measure as $G$ is locally compact. Therefore we can use the Lebesgue decomposition theorem to decompose $f(x)$ as

$$f_x = f_x^{ac} + f_x^{sc} + f_x^{sd} \tag{72}$$

where $f_x^{ac}$ is absolutely continuous with respect to $\mu_x$, $f_x^{sc}$ is singular with respect to $\mu_x$ and continuous, and $f_X^{sd}$ is singular with respect to $\mu_x$ and discrete.

First we construct a sequence of weighted closed canonicalizations $f^n$ converging to $f$ such that upon decomposing each $f^n$ as above, there is no singular continuous factor.

$f^{sc}$ is singular continuous with respect to $\mu_x$, hence there is a measurable set $S \subseteq \overline{Gx}$ such that $\mu_x(S) = 0$ and $f_x^{sc}(S^c) = 0$. Fix a point $y \in S$. For $l > 0$, let $K_l = \overline{B(y,l)} \cap \overline{S}$. Then we have

$$\bigcup_l K_l = \overline{S} \tag{73}$$

By continuity of measure

$$\lim_{l \to \infty} f_x^{sc}(K_l) = f_x^{sc}(\overline{S}) = f_x^{sc}(S) + f_x^{sc}(\overline{S} \cap S^c) = f_x^{sc}(S) \tag{74}$$

Fix $l$ large enough such that $f_x^{sc}(K_l) > f_x^{sc} - \frac{1}{n}$. $K_l$ is compact so we may find finitely many points $x_1, \ldots, x_{p(n)}$ such that the $\frac{1}{n}$ balls around these points cover $K_l$. Call these open balls $U_i$. Let $U_i' = U_i \backslash \bigcup_{j < i} U_j$. Then each $U_i'$ contains only $x_i$ and none of the other $x_j$. Define a discrete measure $f_n'$ which is non-zero only on the $x_i$ and taking the following values at those points

$$\hat{f}^n(\{x_i\}) = f_x^{sc}(U_i') \tag{75}$$

Then one easily sees that

$$\hat{f}^n(S) = f_x^{sc}(U_1') + f_x^{sc}(U_2') + \cdots + f_x^{sc}(U_{p(n)}') = f_x^{sc}(K_l) \tag{76}$$

Now let $f^n = f_x^{ac} + \hat{f}^n + f_x^{sd}$. This is then a measure with no singular discrete part with respect to $\mu_x$. We now check that $f^n \to f$. Let $\phi \in C_b^0(X)$. We may assume that $\phi$ is non-negative on $X$ by splitting it up as $\phi = \phi^+ + \phi^-$ for $\phi^+$ non-negative and $\phi^-$ non-positive.

$$f_n(\phi)(x) - f(\phi)(x) = \int \phi \, d(f_n(x) - f(x))$$

$$= \int \phi \, d(f_{sc}(x) - f_n'(x))$$

$$= \sum_{i=1}^p \left( \int_{U_i'} \phi \, df_{sc}(x) - \phi(x_i) f_n'(U_i') \right) + \int_{S \backslash K_n} \phi \, df_{sc}(x)$$

Therefore we see that, with $C = \max \phi$,

$$|f_n(\phi)(x) - f(\phi)(x)| \leq \max\left( \sum_{i=1}^{p(n)} f_{sc}(U_i')(\max_{U_i'} \phi - \phi(x_i)), \sum_{i=1}^p f_{sc}(U_i')(\phi(x_i) - \min_{U_i'} \phi) \right) + \frac{C}{n} \tag{77}$$

Let $\epsilon > 0$. Take $N$ large enough such that for all $n > N$, and $i = 1, \ldots p$

$$|\max_{U_i'} \phi - \min_{U_i'} \phi| < \epsilon \tag{78}$$

Recall that we can do this as there are finitely many $U_i'$, each of which is contained in the $\frac{1}{n}$ ball around $x_i$. Plugging this into Equation (77) we get

$$|f_n(\phi)(x) - f(\phi)(x)| \leq \frac{C}{n} + \sum_{i=1}^{p(n)} f_{sc}(U_i')\epsilon = \frac{C}{n} + \epsilon f_{sc}(K_n) \leq \frac{C}{n} + \epsilon f_{sc}(S) \tag{79}$$

Taking $n$ large and $\epsilon$ small gives the convergence result.

Now we show that we can approximate a weighted closed canonicalization with no singular continuous part by weighted orbit canonicalizations. As before, decompose $f_x = f_x^{ac} + f_x^{sd}$ with respect to $\mu_x$. For any $n$, choose $V_n(x)$ to be the $\frac{1}{n}$ open ball around $\overline{Gx} \backslash Gx$. $f^{sd}$ is singular and discrete, hence it is supported on a (possibly infinite) set of points $x_1, x_2, \ldots$ in $\overline{Gx}$. For each $x_i$ choose a sequence of points $(x_{in})$ in $Gx$ converging to $x_i$. We may choose these such that for a fixed $n$, each of the $x_{in}$ are different for every $i$. Further we assume that these are chosen such that for every $i$

$$d(x_i, x_{in}) < \frac{1}{n} \tag{80}$$

Then define the singular measure $\hat{f}^n$ which is non-zero only on the $x_{in}$ for every $i$, taking value

$$\hat{f}^n(\{x_{in}\}) = f_x^{sd}(\{x_i\}) \tag{81}$$

Define the absolutely continuous measure $\tilde{f}^n$ by

$$\tilde{f}^n(U) = f^{ac}(U \backslash V_n(x)). \tag{82}$$

Then let

$$f^n(x) = \hat{f}^n + \tilde{f}_n \tag{83}$$

This is then a measure supported on $Gx$ and by construction it is $G$-invariant. Hence this is a weighted orbit canonicalization. Now we show that $f^n \to f$. Let $\phi \in C_b^0(X)$ and $x \in X$. Then

$$|f^n(\phi)(x) - f(\phi)(x)| = \left| \int_{\overline{Gx}} \phi \, df_x - \int_{Gx} \phi \, df_x^n \right| \tag{84}$$

$$= \left| \int_{\overline{Gx}} \phi \, df_x^{ac} - \int_{Gx} \phi \, d\tilde{f}_x^n \right| + \left| \int \phi \, df_x^{sd} - \int \phi \, d\hat{f}_x^n \right| \tag{85}$$

$$= \left| \int_{V_n(x)} \phi \, df_x^{ac} \right| + \left| \sum_i f_x^{sd}(\{x_i\})(\phi(x_i) - \phi(x_{in})) \right| \tag{86}$$

$$\leq C f_x^{ac}(V_n(x)) + \sum_i f_x^{sd}(\{x_i\}) \, |\phi(x_i) - \phi(x_{in})| \tag{87}$$

Now fix $\epsilon > 0$. As $f_x^{sd}$ is a finite measure, there is some $I$ such that

$$\sum_{i=I}^{\infty} f_x^{sd}(\{x_i\}) < \epsilon \tag{88}$$

Thus we get

$$|f^n(\phi)(x) - f(\phi)(x)| \leq C f_x^{ac}(V_n(x)) + \left( \sum_{i=1}^{I-1} + \sum_{i=I}^{\infty} \right) f_x^{sd}(\{x_i\}) |\phi(x_i) - \phi(x_{in})| \tag{89}$$

$$\leq C f_x^{ac}(V_n(x)) + \sum_{i=1}^{I-1} f_x^{sd}(\{x_i\}) |\phi(x_i) - \phi(x_{in})| + 2C\epsilon \tag{90}$$

Now choose $n$ large enough such that $f^{ac}(V_n(x)) < \epsilon$ and for each $i = 1, \ldots, I-1$, $|\phi(x_i) - \phi(x_{in})| < \epsilon$. Then we get a bound

$$|f^n(\phi)(x) - f(\phi)(x)| \leq 3C\epsilon + D\epsilon \tag{91}$$

Hence, $f^n \to f$. □

**Theorem G.5.** $\mathrm{WCan}_G(X)^{\mathrm{cont}}$ *is sequentially closed.*

*Proof.* Suppose that $(\mu^n)_n$ is a convergent sequence of continuous weighted canonicalizations, converging to a weighted canonicalization $\mu$. We will show that $\mu$ is continuous. To do this, fix a

convergent sequence $(x_m)_m$ in $X$ converging to $x$. Let $\phi \in C_b^0(X)$. Then we have the following bound

$$|\mu(\phi)(x)-\mu(\phi)(x_m)| \leq |\mu(\phi)(x)-\mu_n(\phi)(x)|+|\mu_n(\phi)(x)-\mu_n(\phi)(x_m)|+|\mu_n(\phi)(x_m)-\mu(\phi)(x_m)| \tag{92}$$

Fix $\epsilon > 0$. As $\mu_n$ converges to $\mu$ we may find a large enough $n$ such that both the first and third terms are less than $\epsilon$. For this $n$, as $\mu_n$ is continuous, we may find $m$ large enough such that the middle term is less than $\epsilon$ as well. Hence the right hand side is bounded by $3\epsilon$, so we see that $\mu$ is continuous. $\square$

**Proposition G.6.** *Let $Y$ be a topological space. Let $f : Y \to [0, +\infty]$ be lower semi-continuous. Then the set*

$$M = \{y \in Y | f(y) = \inf_Y f\} \tag{93}$$

*is closed.*

*Proof.* Let $\alpha$ be the infimum of $f$. We have that $M = f^{-1}(\{\alpha\})$. As $\alpha$ is the infimum over $Y$, we see that in fact $M = f^{-1}([0, \alpha])$. Then as $f$ is lower semi-continuous, this set is closed in $Y$. $\square$

**Lemma G.7.** *Let $E$ be a lower semi-continuous energy function. Then $\kappa_E$ is continuous if and only if for every convergent sequence $x_n \to x$ we have $\mathrm{Li}\, \mathcal{M}_E(x_n) = \mathcal{M}_E(x)$.*

*Proof.* Let $E : X \to [0, +\infty]$ be an energy function such that the corresponding energy minimizing canonicalization $\kappa_E$ is continuous. Consider a convergent sequence $(x_n)$ in $X$, converging say to $x$. $\mathcal{M}_E(x)$ is closed inside $\overline{Gx}$, being the set of minimizers of a lower semi-continuous function. As $\overline{Gx}$ is closed in $X$, $\mathcal{M}_E$ is closed in $X$ as well. As $X$ is metrizable, we may choose a countable neighborhood basis of $\mathcal{M}_E(x)$, which we shall write as a decreasing sequence $(U_m)$ of open neighborhoods such that

$$\bigcap U_m = \mathcal{M}_E(x) \tag{94}$$

Let $C_m = X \backslash U_m$, this is closed. As $X$ is normal Hausdorff, we may find $\phi_m : X \to [0, 1]$ continuous such that $\phi_m^{-1}(\{0\}) = C_n$ and $\phi_m^{-1}(\{1\}) = \mathcal{M}_E(x)$.

Now as $\kappa_E$ is continuous we have that the measures $\kappa_E(x_n)$ weakly converge to $\kappa_E(x)$. Hence for every $m$ we have that

$$\lim_{n \to \infty} \int_{\mathcal{M}_E(x_n)} \phi_m \, d\mu_{x_n} \to \int_{\mathcal{M}_E(x)} \phi_m \, d\mu_x = 1 \tag{95}$$

Hence

$$\lim_{n \to \infty} \int_{\mathcal{M}_E(x_n)} \phi_n \, d\mu_{x_n} = 1 \tag{96}$$

Let $\epsilon > 0$. We may find $N$ such that for all $n > N$, the left hand side is greater than $1 - \epsilon$. As $\mu_{x_n}(\mathcal{M}_E(x_n)) = 1$, we must have that the minimum of $\phi_n$ over $\mathcal{M}_E(x_n)$ is greater than $1 - \epsilon$. As $C_n = \phi_n^{-1}(\{0\})$, we then must have that $\mathcal{M}_E(x_n) \subseteq U_n$. Hence by the definition of Kuratowski convergence we have $\mathrm{Li}\, \mathcal{M}_E(x_n) \subseteq \mathcal{M}_E(x)$.

Now suppose that $y \in \mathcal{M}_E(x)$. Then we may as above choose a decreasing sequence $(U_m)$ of open neighborhoods of $y$ which form a neighborhood basis. Then fix for each $m$, $\phi_m : X \to [0, 1]$ continuous such that $\phi_m^{-1}(\{0\}) = X \backslash U_m$ and $\phi_m^{-1}(\{1\}) = \{y\}$. By the same estimates as before we must have that for large enough $n$, each $\mathcal{M}_E(x_n)$ intersects $U_m$ non-trivially. Hence $y \in \mathrm{Li}\, \mathcal{M}_E(x_n)$. So $\mathrm{Li}\, \mathcal{M}_E(x_n) = \mathcal{M}_E(x)$.

Conversely suppose that we are given that for every convergent sequence $x_n \to x$, $\mathrm{Li}\, \mathcal{M}_E(x_n) = \mathcal{M}_E(x)$. Let $\phi \in C_b^0(X)$. We may assume that $\phi$ is non-negative. Fix $\epsilon > 0$. As $\mathcal{M}_E(x)$ is compact, we may find an open set $U$ containing $\mathcal{M}_E(x)$ such that

$$\max_{y \in U} \phi(y) - \min_{y \in U} \phi(y) < \epsilon \tag{97}$$

As $\mathrm{Li}\, \mathcal{M}_E(x_n) \subseteq \mathcal{M}_E(x)$, we may find $N$ large enough such that for all $n \geq N$, $\mathcal{M}_E(x_n) \subseteq U$. Then

$$\min_{y \in \mathcal{M}_E(x)} \phi(y) \leq \int_{\mathcal{M}_E(x)} \phi \, d\mu_x \leq \max_{y \in \mathcal{M}_E(x)} \phi(y) \tag{98}$$

and similarly

$$\min_{y \in \mathcal{M}_E(x_n)} \phi(y) \leq \int_{\mathcal{M}_E(x_n)} \phi \, d\mu_{x_n} \leq \max_{y \in \mathcal{M}_E(x_n)} \phi(y) \tag{99}$$

Hence we see both of these integrals are bounded between the min and max of $\phi$ on $U$. Hence

$$\left| \int_{\mathcal{M}_E(x)} \phi \, d\mu_x - \int_{\mathcal{M}_E(x_n)} \phi \, d\mu_{x_n} \right| \leq 2\epsilon \tag{100}$$

Therefore this converges and we have weak convergence. □

## H    OVERVIEW OF PREVIOUS CONCEPTS

Frames and canonicalization both represent different ways to impose group equivariance or invariance on arbitrary functions. The most basic approach is simply to average over all group elements, sometimes refered to as the Reynolds operator. For a finite group $G$ and a continuous bounded function $\phi$ it is given by $\mathcal{R}_G(\phi)(x) = \frac{1}{|G|} \sum_{g \in G} \phi(g^{-1} \cdot x)$. This can also be modified to impose $G$-equivariance by changing the integrand to $g \cdot \phi(g^{-1} \cdot x)$. Frames allow one to not have to integrate over the full group $G$ while still giving a invariant or equivariant function as output. Specifically, a frame is a function $\mathcal{F} : X \to 2^G \backslash \varnothing$ such that for any $x \in X$ and $g \in G$: $\mathcal{F}(gx) = g\mathcal{F}(x)$. In order to define an action on continuous groups, one must assume that the frame is finite, i.e. that for every $x \in X$ the set $\mathcal{F}(x)$ is finite. This is immediate if the group is finite, but if the group is non-finite this is a very strong restriction, as it implies that the stabilizer groups of the action are finite, as $\mathcal{F}(x) = G_x \mathcal{F}(x)$. This is detailed further in Appendix F.

In order to deal with this restriction, Dym et al. (2024), introduced the notion of *weighted weakly equivariant frames*. These generalize frames to be $G$-equivariant maps from $X$ to probability measures on $G$. Then for any continuous bounded function $\phi$ we may apply a weighted frame $\mu_x$ by $\mu(\phi)(x) = \int_G \phi(g^{-1} \cdot x) \, d\mu_x(g)$. This is still not enough in the setting of non-compact groups as discussed in the subsequent section[5]. Even worse, there is an inherent limitation to using frames: the input to $\phi$ may end up outside the training distribution, resulting in worsened performance. The overall network would therefore need to be retrained, which may be exponentially hard (Kiani et al., 2024a).

To deal with this, Kaba et al. (2023) proposed canonicalization as simply a $G$-invariant function $f : X \to X$ such that for each $x \in X$, $f(x) \in Gx$, which upon pre-composing with a function would make it invariant. In order to preserve continuity of functions, one needs to impose that $f$ itself is continuous. In Dym et al. (2024) it is shown that there are many topological obstructions to constructing such continuous canonicalization functions.

To get around these obstructions Ma et al. (2024) proposes canonicalization to now be set-valued $G$-invariant function $\mathcal{C} : X \to 2^X \backslash \varnothing$ for $\mathcal{C}(x)$ finite, acting on continuous functions $\phi$ by $\mathcal{C}(\phi)(x) = \frac{1}{|\mathcal{C}(x)|} \sum_{y \in \mathcal{C}(x)} \phi(y)$. Contractive canonicalizations, which we refer to instead as *orbit canonicalizations*, are a specific kind of canonicalization where for every $x \in X$, $\mathcal{C}(x) \subseteq Gx$. The orbit-stabilizer theorem applies to show these are equivalent to frames. Namely, there is a map $\alpha$ which sends equivariant frames to orbit canonicalizations. For an equivariant frame $\mathcal{F}$ the corresponding orbit canonicalization $\alpha(\mathcal{F})$ is defined as $\alpha(\mathcal{F})(x) = \{g^{-1} \cdot x | g \in \mathcal{F}(x)\}$. Ma et al. (2024, Theorem 3.1) show that for $G$ finite, $\alpha$ is an isomorphism between the set of equivariant frames and orbit canonicalizations. Energy-based canonicalization was first introduced in Kaba et al. (2023). In energy-based canonicalization, given an energy function $E$, the canonicalizing element $c : X \to G$ is defined as $c(x) \in \arg \min_{g \in G} E\left(g^{-1}x\right)$. As shown in the subsequent section, considering the full set of minimizers, instead of a singleton, naturally gives rise to a frame, and naturally results in set-valued canonicalizations. Extending these to non-compact groups naturally gives rise to weighted closed canonicalizations.

---

[5]*Weakly* equivariant weighted frames also require compact stabiliser subgroups, while the equivalent $\pi_*$ equivariance of F.1 can be defined for non-compact groups.

### H.1 Weighted Canonicalization

For this section let $G$ be a Lie group acting smoothly on a manifold $X$. Let $C_b^0(X)$ be the bounded continuous $\mathbb{R}$-valued functions on $X$. Let $\mathrm{PMeas}(X)$ be the set of Radon probability measures on $X$, with respect to the Borel $\sigma$-algebra on $X$. Let $B(X, \mathbb{R})$ be the space of bounded functions on $X$ and $B(X, \mathbb{R})^G$ is the subspace of $G$-invariant such functions.

**Definition H.1.** A *weighted canonicalization* is a $G$-invariant function $\kappa_{[\cdot]} : X \to \mathrm{PMeas}(X)$. We call the space of weighted canonicalizations $\mathrm{WCan}_G(X)$.

Given a weighted canonicalization $\kappa$ we may define an operator $\mathcal{P}_\kappa : B(X, \mathbb{R}) \to B(X, \mathbb{R})^G$ via

$$(\mathcal{P}_\kappa(\phi))(x) = \int_X \phi \, d\kappa_x. \tag{101}$$

We may write the function $\mathcal{P}_\kappa(\phi)$ simply as $\kappa(\phi)$. As $\kappa_x \in \mathrm{PMeas}(X)$ for each $x \in X$, we have $\|\mathcal{P}_\kappa(\phi)\|_\infty \le \|\phi\|_\infty$. $\kappa(\phi)$ is $G$-invariant as $\kappa$ is. We also define the function $P_\phi(\kappa) = \mathcal{P}_\kappa(\phi)$.

The entire space $\mathrm{WCan}_G(X)$ is much too general to be practical. We would like to consider a much more natural subset from the perspective of canonicalization, namely orbit canonicalizations. We now define a corresponding weighted notion.

**Definition H.2.** A *weighted orbit canonicalization* is a weighted canonicalization $\kappa : X \to \mathrm{PMeas}(X)$ such that for every $x \in X$, $\mathrm{supp}\,\kappa(x) \subseteq Gx$. We denote the set of weighted orbit canonicalizations as $\mathrm{WOCan}_G(X)$.

In particular, for weighted orbit canonicalizations both operators become projections, see Appendix F.2. As in the finite case, there is a function $\alpha : \mathrm{WFra}_G(X) \to \mathrm{WOCan}_G(X)$, which is surjective and injective up to weak-equivalence, see Appendix F.1.

Given the action on continuous functions, we may equip $\mathrm{WCan}_G(X)$ with a certain weak topology, namely the coarsest topology such that all the $P_\phi$ for $\phi$ continuous and bounded become continuous functions. Specifically this means that a sequence of weighted canonicalizations $\kappa_n$ converges to $\kappa$ if and only if for every $\phi \in C_b^0(X)$ the sequence $\kappa_n(\phi)$ converges to $\kappa(\phi)$ pointwise. For the canonicalization problem this is a very natural topology to consider, as we only care about canonicalization through their respective actions on continuous functions.

So far, we have presented a generalization of canonicalization, which is readily applicable to finite or compact Lie groups. However, non-compact Lie groups may have non-closed orbits, and the space of weighted orbit canonicalizations is not closed under taking limits in the weak topology. To understand the closure under taking limits, we introduce the concept of a weighted closed canonicalization.

**Definition H.3.** A *weighted closed canonicalization* is a weighted canonicalization $\kappa : X \to \mathrm{PMeas}(X)$ such that for every $x \in X$, $\mathrm{supp}\,\kappa(x) \subseteq \overline{Gx}$. Call the set of weighted closed canonicalizations $\mathrm{WCCan}_G(X)$.

The following theorem shows that these allow us to fully characterize the behaviour of weighted orbit canonicalizations under taking limits.

**Theorem H.1.** *(Theorem G.4) The sequential closure of* $\mathrm{WOCan}_G(X)$ *is* $\mathrm{WCCan}_G(X)$

Using this theorem and the setup above we may now write down our main diagram connecting all the different notions of canonicalization and framing together, shown in Figure 3. In analogy with Dym et al. (2024), we may now ask whether such canonicalizations take continuous functions to continuous functions. In the presented framework this is a simple condition to add.

**Definition H.4.** A weighted canonicalization $\kappa : X \to \mathrm{PMeas}(X)$ is continuous if for every $\phi \in C_b^0(X)$, $\kappa(\phi)$ is continuous. Call the space of continuous weighted canonicalizations $\mathrm{WCan}_G^{cts}(X)$. We similarly define continuous weighted orbit canonicalizations and continuous weighted canonicalizations.

**Proposition H.2.** *A weighted canonicalization $\kappa$ is continuous if and only if for every convergent sequence $x_n \to x$ in $X$, the sequence of measures $\kappa_{x_n}$ converges weakly to $\kappa_x$.*

**Theorem H.3.** *(Theorem G.5)* $\mathrm{WCan}_G^{cts}(X)$ *is sequentially closed inside* $\mathrm{WCan}_G(X)$.

## H.2 Energy Minimizing Canonicalization

In Section 2 so far, we have provided a generalized theoretical understanding of frames and canonicalization, encompassing various scenarios with compact/non-compact and discrete/continuous groups. However, their relevance and how these can be used in practice remains unaddressed. We now move on to the concept of energy frames and canonicalizations, for general Lie groups. We define an energy function on $X$ to be a non-constant function $E : X \to [0, +\infty]$. We further assume that $E$ is topologically coercive: for any $N \in \mathbb{R}^+$ there is a compact $K \subseteq X$ such that $E|_{X \setminus K} > N$.

In the case where $G$ is finite, every orbit in $X$ necessarily has a minimum of $E$, and we note that energy minimization naturally results in a frame

$$\mathcal{F}_E(x) = \underset{g \in G}{\arg\min}\, E(g^{-1}x) \tag{102}$$

**Proposition H.4.** $\mathcal{F}_E$ defines an equivariant frame, i.e. $\mathcal{F}_E(gx) = g\mathcal{F}_E(x)$ for any $g \in G$, $x \in X$.

However when $G$ is not finite this set is not guaranteed to be finite. If $G$ is additionally non-compact, then $E$ does not necessarily achieve its minimum on every orbit in $X$, making this optimization problem not well-defined. For this reason, we instead turn to weighted canonicalizations. By taking the closures of orbits, we solve the problem of $E$ not having minima, so we are naturally led to constructing weighted closed canonicalizations. First, we define the energy-minimizing set, $\mathcal{M}_E(x) = \{x' \in \overline{Gx} \mid E(x') = \inf_{y \in \overline{Gx}} E(y)\}$, set of minima of $E$ on the closure of the orbit of $x$.

**Proposition H.5.** *(Proposition G.2)* $\mathcal{M}_E(x)$ is G-invariant and non-empty.

If $\mathcal{M}_E(x)$ is finite for each $x \in X$, then we define the (non-weighted) closed canonicalization $\kappa_E \in \text{WCCan}_G(X)$ by taking $\kappa_E(x)$ to be the normalized counting measure on $\mathcal{M}_E(x)$ as usual.

For a general $E$, we may not be able to put any reasonable probability measure on the set $\mathcal{M}_E(x)$, for example if $\mathcal{M}_E(x)$ has highly fractal geometry. We therefore must assume that $E$ is such that for all $x$, the Hausdorff measure of $\mathcal{M}_E(x)$ is non-zero. As each $\mathcal{M}_E(x)$ is compact, the Hausdorff measure is finite, hence combined with this assumption we may normalize it to obtain a probability measure on $\mathcal{M}_E(x)$ which is a weighted canonicalization $\kappa_E(x) \in \text{WCCan}_G(X)$. Refer to Appendix E for more discussion on the Hausdorff measure and on why this assumption is reasonable. Note that if $\mathcal{M}_E(x)$ is finite, then the normalized Hausdorff measure constructed agrees with the normalized counting measure. Hence this construction is a generalization of the purely finite case.

For lower semi-continuous energy functions, we have the following characterization of which energies give rise to continuous weighted canonicalizations.

**Lemma H.6.** *(Lemma G.7) Let $E$ be a lower semi-continuous energy function. Then $\kappa_E$ is continuous if and only if for every convergent sequence $x_n \to x$ we have $\text{Li}\,\mathcal{M}_E(x_n) = \mathcal{M}_E(x)$.*

We have the following theorem telling us which weighted closed canonicalizations arise from energy minimizations.

**Theorem H.7.** *(Theorem F.3) Let $\kappa \in \text{WCCan}_G(X)$ be such that $\kappa_x$ is the normalized Hausdorff measure on its support and for all $y \in \overline{Gy}$, $\text{supp}\,\kappa_y = \text{supp}\,\kappa_x \cap \overline{Gx}$. Then there is an energy function $E$ such that $\kappa_E = \kappa$.*

## H.3 Almost everywhere continuity is not as impossible

**Canonicalization** Let $X$ be a space with a group $G$ acting on it by homeomorphisms. Then a continuous canonicalization function is a continuous $G$-equivariant map $v : X \to X$ which sends any element $x$ to an element of the orbit $Gx$. In Dym et al. (2024) it is shown that in general these do not exist, as it is much too strong of a condition. In fact, because a continuous canonicalization function will be a right-inverse to the quotient map $X \to X/G$, a necessary condition to the existence of $v$ is that the map $\pi_1(X) \to \pi_1(X/G)$ induced by the quotient must be a surjection. If the quotient $X \to X/G$ is a covering map then this only holds if $G$ is trivial. For a concrete example of this, we can consider quotienting $\mathbb{R}^2$ by the translation action of $\mathbb{Z}^2$. Then $\pi_1(\mathbb{R}^2) = 0$ while $\pi_1(\mathbb{R}^2/\mathbb{Z}^2) = \mathbb{Z}^2$, showing that a continuous canonicalization function can't exist.

On the other hand, global existence of a continuous canonicalization function is unnecessary for machine learning applications. As long as we can construct a canonicalization function which is

continuous away from a set of measure zero, this will suffice for our applications. Specifically we introduce the notion of an almost everywhere (a.e.) continuous canonicalization function.

**Definition H.5.** An a.e. continuous canonicalization function is a function $v : X \rightarrow X$ such that there exists a set $S \subset X$ of measure-zero such that

   (i) $G(S) \subseteq S$

  (ii) $v$ is continuous away from $S$

 (iii) $\forall x \in S, \exists g \in G$ st. $v(x) = gx$

 (iv) $v$ is $G$-invariant

As with the regular canonicalization functions, we can use such a function $v$ to construct equivariant and invariant functions. Given an arbitrary function $f : X \rightarrow \mathbb{R}$ it is easy to see that

$$\mathcal{I}_v(f)(x) = f(v(x)) \tag{103}$$

is $G$-invariant. Further it is easy to check that this sends a.e. continuous functions to a.e. continuous functions.

The added freedom of the set $S$ where $v$ can take arbitrary values allows us to avoid the topological restrictions that prevent the existence of globally continuous canonicalization functions. For a simple example consider again the presentation of a torus as the quotient of $\mathbb{R}^2$ by the translation action. Then an easy example of an a.e. continuous canonicalization function is the function

$$v(x, y) = ([x], [y])$$

where $[a]$ is the decimal part of a real number $a \in \mathbb{R}$. This is discontinuous whenever either $x$ or $y$ are integers, but this set has measure zero.

For another example, consider the action of the orthogonal group $O(d)$ on the space of matrices $\mathbb{R}^{d \times n}$. The action preserves the rank of a matrix $A \in \mathbb{R}^{d \times n}$, so the subset of full rank matrices is preserved by $O(d)$. This is an open dense set, and if we take $\mu$ to just be the Lebesgue measure, its complement has measure zero. So take $S$ to be the set of matrices with non-full rank. Let $A$ be a matrix of full rank. Then one of the $d \times d$ minors of $A$ is invertible, we may assume that it is the first one. Thus we can write $A = (A_0 A_1)$ with $A_0$ an invertible $d \times d$ matrix. Applying the Gram-Schmidt procedure to $A_0$, we obtain $A_0 = QR$ with $Q$ orthogonal and $R$ upper-triangular. Now we take $v(A) = Q^T A$. $Q$ is given by a polynomial in the elements of $A$, hence $v$ is smooth. Further, if $O \in O(d)$, then we can decompose $OA$ as $OQR$ with $OQ$ orthogonal. Hence,

$$v(OA) = (OQ)^T OA = Q^T O^T OA = Q^T A = v(A)$$

Therefore $v$ is an a.e. continuous canonicalization function, with $S$ the set of non-full rank matrices in $\mathbb{R}^{d \times n}$.

**Invariant Theory**    Finding and computing $G$-invariant functions is a rich topic which comes up in many different situations. In algebraic geometry, one wants to find generators for $G$-invariant polynomials, i.e. a set of $G$-invariant polynomials $\sigma_1, \ldots, \sigma_n$ such that any other $G$-invariant polynomial $f$ can be written as

$$f = g(\sigma_1, \ldots, \sigma_n)$$

for some polynomial $g$. For linearly reductive groups acting on $\mathbb{C}^n$, there is an algorithm for computing these generators, which is described in Derksen & Kemper (2015).

In the smooth case, there is a classic result of Schwarz (Schwarz, 1975), telling us that if $G$ is a compact Lie group acting on a smooth manifold $X$, with generators for the space of $G$-invariant polynomials being $\sigma_1, \ldots, \sigma_n$, then any $G$-invariant smooth functions $f$ can be written as

$$f = g(\sigma_1, \ldots, \sigma_n)$$

where $g$ is an some smooth function on $\mathbb{R}^n$.

---

**Algorithm 2:** Canonicalization function via Lie algebra descent

---

**Data:** Non-canonical input $x$
**parameters :** $N_{\texttt{outer}}$, $N_{\texttt{inner}}$, step-sizes $\eta_i$, Energy $E : \mathcal{X} \to \mathbb{R}$
**Result:** Canonicalized input $\hat{x} = g^{-1} \cdot x$; inverse of canonicalizing group element $g^{-1}$;
         canonicalizing group element $g$.
$g^{-1} \leftarrow \text{id}$;
steps = {};
**for** $k = 0 \ldots N_{outer}$ **do**
   # Do gradient descent on $\xi \in \mathfrak{g}$
   $\xi \leftarrow 0$;
   **for** $i = 0 \ldots N_{inner}$ **do**
      $\xi \leftarrow \xi - \eta_i \nabla_\xi E(g^{-1} \exp(-\xi) \cdot x)$
   **end**
   steps.append($\xi$);
   $g^{-1} \leftarrow g^{-1} \exp(-\xi)$  #Recalibrate to be in the right tangent plane
**end**
$g \leftarrow \text{id}$  #Reconstruct the element $g$
**for** $k = N_{outer} \ldots 0$ **do**
   $g \leftarrow \exp(\text{steps}[k])g$
**end**
**return** $g^{-1} \cdot x$; $g^{-1}$; $g$

---

## I    ALGORITHMS FOR SOLVABLE AND NON-SOLVABLE LIE ALGEBRAS

As discussed in the main body, in certain cases it is possible to significantly simplify the algorithmic approach for minimizing Equation (2). We will distinguish to specific cases:

**If the group is compact or the algebra is solvable**    In either of these cases, we are able to parameterise the group globally using the Liealgebra. Denoting the basis as $v_i \in \mathfrak{g}$, and the coefficients as $\alpha_i \in \mathbb{R}$, we have that we can represent any $g \in G$ as

$$g = \exp \sum_i \alpha_i v_i,$$

if $G$ is compact, or as

$$g = \prod_i \exp \alpha_i v_i,$$

if the Lie algebra is solvable with $v_i$ ordered as to represent the nested subalgebras. In both of these cases we can reduce the optimization to be done over the vector of coefficients $\alpha \in \mathbb{R}^{\dim \mathfrak{g}}$. Both of these can be viewed as specific types of global retraction mappings $\tau : \mathfrak{g} \to G$. For this case we use Algorithm 2

**If the group is non-compact or the algebra is not solvable**    Such a setting is particularly important for PDE-based groups, as these naturally result in groups that are non-compact, see section 4.3. In this case it is no longer possible to find a global retraction, and a natural question arises - can the infinitesimal actions ever be enough? It turns out that the answer is yes, quite generically, resulting in Algorithm 1:

**Proposition I.1** (Proposition 1.24 from Olver (1993))**.** *Let $G$ be a connected Lie group and $U \subset G$ a neighbourhood of the identity. Also, let $U^k \equiv \{g_1 \cdot g_2 \cdot \ldots \cdot g_k : g_i \in U\}$ be the set of $k$-fold products of elements of $U$. Then*

$$G = \bigcup_{k=1}^{\infty} U^k.$$

*In other words, every group element $g \in G$ can be written as a finite product of elements of $U$.*

---

**Algorithm 3:** Canonicalization function via coordinate Lie algebra descent

---

**Data:** Non-canonical input $x$
**parameters:** N, step-sizes $\eta_i$, Energy $E : \mathcal{X} \to \mathbb{R}$, $\tau_k$ if using proximal
**Result:** Canonicalized input $\hat{x} = g^{-1} \cdot x$; inverse of canonicalizing group element $g^{-1}$;
       canonicalizing group element $g$.
$g^{-1} \leftarrow \text{id}$;
steps = {};
**for** $k = 0 \ldots N$ **do**
      # Do coordinate descent on $\xi \in \mathfrak{g}$, by picking index $j$
      $j \leftarrow \text{pick}[0, \ldots, \dim \mathfrak{g}]$
      $\lambda_k \leftarrow \arg\min_\lambda E(g^{-1} \exp(-\lambda v_j) \cdot x)$ ;
      # May be desirable to use proximal descent
      # $\lambda_k \leftarrow \arg\min_\lambda E(g^{-1} \exp(-\lambda v_j) \cdot x) + \frac{1}{2\tau_k}\|\lambda\|^2$ ;
      $\xi \leftarrow \lambda_k v_j$
      steps.append($\xi$);
      $g^{-1} \leftarrow g^{-1} \exp(-\xi)$   #Recalibrate to be in the right tangent plane
**end**
$g \leftarrow \text{id}$   #Reconstruct the element $g$
**for** $k = N \ldots 0$ **do**
      $g \leftarrow \exp(\text{steps}[k])g$
**end**
**return** $g^{-1} \cdot x$; $g^{-1}$; $g$

---

**If the group only provides information about basis vector action** In most examples of PDE symmetries, integrating the action for a generic $v \in \mathfrak{g}$ is impossible, and instead only the action with respect to certain basis $\{v_i\}$ is known. In this case coordinate descent schemes must be used. For this we use Algorithm 3.

