# OpenReview forum: "Lie Algebra Canonicalization: Equivariant Neural Operators under arbitrary Lie Groups"
_ICLR.cc/2025/Conference — ICLR 2025 Poster_

### Official Review · Reviewer_CTGT · 2024-11-03

**Soundness:** 3
**Presentation:** 2
**Contribution:** 3
**Rating:** 6
**Confidence:** 2

**Summary:**

The paper proposes LieLAC, a method to make pre-trained models equivariant to Lie group symmetries through input canonicalization via energy minimization. The key strength of the work is its well-grounded theoretical foundation and potential applicability to a wide range of symmetry groups and tasks. However, the practical challenges of finding an appropriate energy function, finding its minimizer, the requirement of finetuning canonicalized network and limited experimental evaluation raise concerns about broader applicability beyond simple demonstrated cases.

**Strengths:**

1. The proposed canonicalization approach allows employing pre-trained non-equivariant models while making them equivariant with canonicalization.
2. The paper provides a good overview of the related work on canonicalization and frame-averaging methods.
2. Figure 1 provides an intuitive example of the effect of canonicalization on the decision boundary.
3. The proposed approach is generic and is potentially applicable to a wide range of models and learning tasks.
4. The paper provides strong theoretical support for the proposed method.
5. Canonicalization and frame averaging methods have been demonstrated to be successful in various domains, including geometric and imaging modalities. The paper further extends the canonicalization framework to PDE/ODE modeling with more complex symmetry groups.
6. The paper nicely elaborates on the challenges of choosing the appropriate energy function and it outlines key considerations in doing so.

**Weaknesses:**

1. The paper states "Prior work on equivariant neural networks focuses on simple groups ... that are not reach enough to encode specific structure found in scientific applications". Does it apply to the recently introduced Clifford algebra and geometric algebra networks which aim to provide a more general and flexible framework for equivariant NNs?
2. Paper is uneasy to digest. For example, 4 definitions, 3  theorems, 3 propositions, and one lemma are stated on just two pages.  Authors should think about how to make the flow of the paper more intuitive by either providing a more comprehensive background or by presenting a simplified formulation in the main text of the paper while providing the complete theoretical details in the appendix.
3. The distance-based relaxation in 331-333 is not clear. Can authors elaborate more on what is meant here?
4. While the method initially appears generic, finding the minimizer of the energy function is challenging and it is not clear how well it can be done besides the cases presented in the paper. The optimization problem of energy minimization in Eq.2 requires a lot of design choices and heuristics such as adversarial regularization.
5. Experimental evaluation is rather limited and is focused on toy or simple tasks. Also, the reported PDE evolution errors are reported only on one instance of initial conditions.
6. Table 2 suggests the misalignment of canonicalized samples with the original data indeed takes place (LieLAC+Poseidon vs Poseidon+ft) which again outlines the challenge of minimizing the energy function. This requires finetuning the baseline neural solver to adapt to new OOD  samples which can be a serious practical bottleneck.

Some important missed related work on equivariance:
- M. Zhdanov et al. Clifford-steerable convolutional neural networks NeurIPS 2023.
- D. Ruhe et al. Clifford group equivariant neural networks NeurIPS 2023.

Some important missed related work on Lie groups in NNs:
- A. Moskalev et al. Liegg: Studying learned lie group generators. NeurIPS 2022.
- N. Gruver et al. The Lie Derivative for Measuring Learned Equivariance. ICLR 2022.

**Questions:**

I suggest authors address and elaborate on weaknesses.

---

> ### Author Response · Authors · 2024-11-19
> **[1/2]**
>
> We thank the reviewer for their thoughtful feedback and appreciate their recognition of our work's strengths, particularly its theoretical grounding and potential for broad applicability. We would also like to thank the reviewer for the references provided, which have now been included in the paper. We address the specific concerns raised below:
>
> * “ The paper states "Prior work on equivariant neural networks focuses on simple groups ... that are not rich enough to encode specific structure found in scientific applications". Does it apply to the recently introduced Clifford algebra and geometric algebra networks which aim to provide a more general and flexible framework for equivariant NNs? “
>   * These statements do indeed apply to geometric algebra - Geometrical Algebras implicitly assume that the objects considered transform in a linear representation of isometries of the underlying space. In this case one can decompose all possible fields in an equivariant representation. A similar thing follows when we have an algebraic group acting algebraically on a vector space, because then there is a very rich theory of invariant polynomials - both theoretically and computationally. Unfortunately, the groups necessary for working with PDEs, while they seem to generally be algebraic, do not act algebraically (for instance, the heat equation has a square root and an exponential which are not algebraic). This is precisely why we needed to introduce this more general framework - this has now also been clarified further in the introduction.
> * “Paper is uneasy to digest. For example, 4 definitions, 3 theorems, 3 propositions, and one lemma are stated on just two pages. Authors should think about how to make the flow of the paper more intuitive by either providing a more comprehensive background or by presenting a simplified formulation in the main text of the paper while providing the complete theoretical details in the appendix.”
>   * We completely agree with the reviewers comment and have revised our manuscript accordingly to improve clarity and focus. We would appreciate any further feedback on this.
> * “ The distance-based relaxation in 331-333 is not clear. Can authors elaborate more on what is meant here? “
>   * Taking the heat equation as an example, we are interested in transformations that map $x$ into the interval $[0,2\pi]$. Thus, to enforce this, we can simply add an extra term $(\max(x_{min}-0,0))^2 + (\max(x_{max}-2\pi,0))^2$ to the energy.
> * “ While the method initially appears generic, finding the minimizer of the energy function is challenging and it is not clear how well it can be done besides the cases presented in the paper. The optimization problem of energy minimization in Eq.2 requires a lot of design choices and heuristics such as adversarial regularization. “
>   * We agree with the reviewer - a price has to be paid somewhere, when imposing equivariance. For simple groups many equivariant architectures have been proposed, making the our approach rather clunky. However, there exists a much larger class of problems (of which PDE symmetries is one) where no such architectures exist. This work enables achieving equivariance in such problems. While we do pay a price in the complexity of trying to find minimizers of an energy, this allows us to construct a method that works for many difficult cases of group transformations - something that is not present in the field at all.
>   * It is also worth emphasising that the goal of this work was not to propose the best possible energy choice, which may turn out to be much simpler than the proposed constructions, but instead showcase that this is possible.

---

> ### Author Response · Authors · 2024-11-19
> **[2/2]**
>
> * “Experimental evaluation is rather limited and is focused on toy or simple tasks. Also, the reported PDE evolution errors are reported only on one instance of initial conditions.”
>   * The reported metrics for PDE evolution are taken as an average over a number of different initial conditions. This has now been clarified in the paper. We agree that some of the experiments are toy example (particularly the 2D setting or Heat and Burgers evolution). However, the MNIST experiments consider a relatively complex group as an example, while the ACE Poseidon example considers a large scale network, with large images, illustrating the scaling of the method with respect to the problem size.
>   * Could you elaborate on what kinds of tasks you would consider to be non-simple in this context? This would be very helpful in guiding our future research and ensuring the method's applicability to a wider range of problems.
> * “Table 2 suggests the misalignment of canonicalized samples with the original data indeed takes place (LieLAC+Poseidon vs Poseidon+ft) which again outlines the challenge of minimizing the energy function. This requires finetuning the baseline neural solver to adapt to new OOD samples which can be a serious practical bottleneck.”
>   * We would like to emphasise that kNN classification, MNIST, Heat and Burger experiments *did not* require any finetuning, as the tasks considered are not as complex and do not require the same level of precision. Particularly, even without finetuning in Table 2 we can see significant improvement on out of domain data. The base solver needs to be tuned in this case, as this is a regression task. What is more - the Poseidon model is large and pre-trained on a lot of data - making it already very good to begin with - finetuning was only necessary to show that the model is able to reach the same level of accuracy on both in and out of domain data. Table 2 does indeed suggest a slight misalignment - but it is important to note the emphasis on ‘slight’.

---

> ### Author Response · Authors · 2024-11-25
>
> As the discussion period draws to a close, we'd like to check in on whether you've had a chance to review our responses and have any follow up questions?
>
> We hope that our reply clarifies and alleviates the reviewer’s concerns. If this is the case, we kindly ask the reviewer to consider raising their rating, given that they are acknowledging the novelty, the strengths and the contributions of our paper.

---

> ### Author Response · Authors · 2024-12-02
>
> Dear Reviewer, with the deadline so close, we'd like to check whether you've had a chance to review our responses and have any follow up questions?
>
> We believe our clarifications address your initial concerns, and if you agree, we would appreciate it if you would consider raising your rating.  We're grateful that you recognize the novelty and contributions of our work.

---

> > ### Comment · Reviewer_CTGT · 2024-12-03
> > **Reviewer's response**
> >
> > I appreciate the author's response which has addressed my concerns. Based on the feedback from other reviewers and the revised version of the paper, I see that the work has already improved significantly. I am not sure the paper is 8/10 due to the limited (although informative) experiments. Since there is no 7/10 option, I will retain my 6/10 rating which is already a positive assessment.

---

> > > ### Author Response · Authors · 2024-12-03
> > >
> > > We would like to thank the reviewer for their quick response, and for positive assesment of our work. For calibration purposes, we’d like to note that the ICLR 2025 rubric differs slightly from previous similar conferences. For example:
> > > * To indicate "Accept", the NeurIPS 2024 rubric says to use 7 whereas the ICLR 2025 rubric says to use 8
> > > * To indicate "Strong Accept", the NeurIPS 2024 rubric says to use 9 whereas the ICLR 2025 rubric says to use 10

---

### Official Review · Reviewer_aU4A · 2024-11-03

**Soundness:** 3
**Presentation:** 1
**Contribution:** 2
**Rating:** 6
**Confidence:** 3

**Summary:**

This paper extends the energy-based canonicalization approach, introduced in [1] to settings where the group in non-discrete and non-compact, exploiting only the infinitesimal generators of Lie algebras. They provide a general framework of constructing energy functionals which can be optimized using standard Lie group descent schemes.'

[1] Sékou-Oumar Kaba, Arnab Kumar Mondal, Yan Zhang, Yoshua Bengio, and Siamak Ravanbakhsh.Equivariance with Learned Canonicalization Functions. In Proceedings of the 40th International Conference on Machine Learning, pp. 15546–15566. PMLR, July 2023.

**Strengths:**

The topic of constructing equivariant networks when one only has access to the infinitesimal generators is an extremely interesting and well-motivated direction of research. The approach this paper takes, which is adapting the canonicalization framework for these settings is novel and using the energy-based canonicalization seems like a promising direction for these cases.  The figures in the paper are also very useful and help provide an intuition for how canonicalization is helping.

**Weaknesses:**

- One of the main limitations of the paper is that how the theoretical results and understandings provided in sections 2 and 3 are used in practice to train a canonicalizer network is not well-explained. I found it difficult to follow how the energy functionals were constructed  in each of the cases and how much that approach is generalizable to more complex systems and PDEs.
- The general experimental setup was not well-explained in the paper. For example, it would be better to include a main algorithm, provide a more detailed explanation of how the energy functional was constructed in each of the experiments settings, and clearly explain the experimental setup and training of each of the datasets.
- The reported results do not include standard deviations over seeds (for example, in tables 1 and 2), and the results for Heat and Burgers’ are missing from the main paper (although included in the appendix). The paper also doesn’t include any comparisons with other baselines, such as data or loss augmentation.
- The paper could be strengthened by providing an understanding of how expensive these optimizations are and how generalizable the suggested approach is for more complex settings.

Overall, I think the topic and direction explored is very interesting and the experiments show promising results but the paper was extremely  difficult to follow (especially understanding how the theoretical discussions are mapped to practical algorithms) which limits its impact.

**Questions:**

See above.

---

> ### Author Response · Authors · 2024-11-19
> **[1/1]**
>
> We thank the reviewer for their thoughtful feedback and appreciate their acknowledgment of the novelty and potential of our approach. We have carefully considered the reviewer's comments and have made significant revisions to the manuscript to address the concerns raised.
>
> Regarding specific questions:
> * “How the theoretical results and understandings provided in sections 2 and 3 are used in practice to train a canonicalizer network is not well-explained.”
>   * We completely agree with the reviewers comment regarding clarity and we have revised the manuscript and more particularly sections 2 and 3 to improve clarity and focus.  To help with presentation of section 2, we clarify precisely the contribution of this paper compared to other neural network based approaches. Section 2 separately summarises existing limitations of previous works, illustrated through an explicit example, showing how our framework overcomes them. Section 3 now combines both generic explanations, as well as specific constructions for the examples provided.
>   * We would like to clarify that there is no canonicalization *network*. Constructing a canonicalization network necessarily requires one to use equivariant architectures, which may not exist (or may be too expensive) for the groups considered in this paper. Instead, the energy functional is parameterized as a network. We have added further detailed information about how the energy functionals were chosen for all problems considered.
>   * We would also like to clarify that sections 2 and 3 concern themselves with formalizing equivalence of frames and canonicalizations for the case of non-compact lie groups in the most general case. Theory provided in those sections does not provide any constructive information towards how to select the energy. In fact there is no unique good way to achieve this, as emphasised in section 4, and the best anyone may be able to do is provide *a* way to do this, which works in practice.
>   * The approach considered in this paper (for MNIST and ACE) has been to train a VAE on the training set of the operator and a (convex) adversarial regulariser based on infinitesimal generators of the group - we have shown that they are able to achieve equivariance and improved out of domain performance, emphasising that they do work. We add a note on all of the above to the paper.
> * “The general experimental setup was not well-explained in the paper ...”
>   * Thank you for the feedback - the training algorithm has now been explained in the paper in the section on energy selection and the canonicalization algorithm included in the main text. However, as mentioned above and in section 4, there is no unique algorithm. Different knowledge of the group requires one to utilise different algorithms 1,2,3. We have included information on this in the paper.
> * “The reported results do not include standard deviations over seeds (for example, in tables 1 and 2), and the results for Heat and Burgers’ are missing from the main paper (although included in the appendix). The paper also doesn’t include any comparisons with other baselines, such as data or loss augmentation.”
>   * Due to space restrictions, experimental results for heat and Burgers equations had to be moved to the appendix. In order to improve the presentation we have now also moved particular information about the heat and Burgers equation to the appendix also.
>   * We agree that it would be beneficial to include comparison with data augmentation and we are currently running the relevant experiments to be included in the revised version. However, due to the expensiveness of data augmentation in training - we are likely to be unable to provide these before the end of the rebuttal period. This particularly emphasises the benefit of using canonicalization - as no extra training (at most - minimal finetuning) of the operators is needed.
>   * In regards to standard deviations, we agree it is a good idea for these to be included. We are now re-running the same experiments with different seeds to be included in the revised version via standard deviations. In the same manner, we are likely to be unable to provide these before the end of the rebuttal period.
>   * For loss augmentation - unfortunately main approaches do not have public code available, and while it is possible to reproduce them, they *do not* result in equivariant models, and no out of distribution generalization occurs. They instead result in more data-efficient models - see eg Akhound-Sadegh et al. 2023.
> * “The paper could be strengthened by providing an understanding of how expensive these optimizations are and how generalizable the suggested approach is for more complex settings. “
>   * We agree with this, and have now provided an extra explanation of the time requirements of canonicalization. However, it is worth emphasising that most of these have not been optimized for, and can most definitely be sped up further.

---

> > ### Author Response · Authors · 2024-11-20
> > **Additional experiments error bars and data augmentation**
> >
> > To provide error bars and compare data augmentation and canonicalization we performed these additional experiments illustrated for the Heat equation and will include in the revised version of the paper. Similar experiments will be conducted for ACE with the Poseidon model, but this may not be completed before the end of the rebuttal period due to the expensiveness of retraining the large model.
> >
> > For the DeepONet applied to the Heat equation we trained the model in two regimes: (1) fixed amplitude  A_k=1 , analogous to PINN training, and (2) amplitude sampled from  A \sim \mathcal{U}[0.5, 5.0] , representing a broader operator training distribution. The model was trained using the physics loss with results averaged over 10 seeds.
> >
> > As shown in Table 1, DeepONet fails to generalize to out-of-distribution amplitudes ( A \sim \mathcal{U}[0.5, 5.0] ) when trained on the fixed  A_k=1  regime. Applying LieLAC restores test accuracy to in-distribution levels. Extending the training range (Table 2) improves generalization via data augmentation but is still outperformed by LieLAC, which uses a canonicalizing group action instead of relying on broader sampling.
> >
> > ### Table 1: L2 relative error for Heat equation (with fixed A_k^Train = 1)
> >
> > | Model               | A_k^Test in [0.95, 1.05] | A_k^Test in [0.5, 5.0]  |
> > |----------------------|--------------------------|--------------------------|
> > | DeepONet            | 0.0498 ± 0.0072         | 0.6572 ± 0.1235         |
> > | LieLAC [DeepONet]   | **0.0443 ± 0.0027**     | **0.0435 ± 0.0017**     |
> >
> > ### Table 2: L2 relative error for Heat equation (A_k^Train in [0.5, 5.0])
> >
> > | Model               | A_k^Test in [0.95, 1.05] | A_k^Test in [0.5, 5.0]  |
> > |----------------------|--------------------------|--------------------------|
> > | DeepONet            | 0.0504 ± 0.0014         | 0.0687 ± 0.0044         |
> > | LieLAC [DeepONet]   | **0.0500 ± 0.0003**     | **0.0500 ± 0.0003**     |

---

> ### Author Response · Authors · 2024-11-25
>
> As the discussion period draws to a close, we'd like to check in on whether you've had a chance to review our responses and have any follow up questions?
>
> We hope that our reply clarifies and alleviates the reviewer’s concerns. If this is the case, we kindly ask the reviewer to consider raising their rating, given that they are acknowledging the novelty, the strengths and the contributions of our paper.

---

> > ### Comment · Reviewer_aU4A · 2024-11-27
> > **Response to Authors' Rebuttal**
> >
> > Thank you for your response. I have read the authors' responses and the new revision. I believe that the updates have indeed made the paper and the contributions more clear and I am willing to increase my score from 5 to 6.

---

### Official Review · Reviewer_VdBH · 2024-11-04

**Soundness:** 3
**Presentation:** 3
**Contribution:** 3
**Rating:** 8
**Confidence:** 4

**Summary:**

The contribution of this paper can be summarized as follows:
- An extension of frames and canonicalization for neural network symmetrization [1-4] to non-compact Lie groups specified by their infinitesimal generators (Section 2.2 and 3),
- A class of optimization-based algorithms for the above energy-based canonicalization using (coordinate) Lie algebra descent (Appendix H),
- Applications of the proposed method to affine and homography group invariant MNIST classification (Section 4.2) and, importantly, neural operator modeling of three PDEs with known point symmetry groups by canonicalizing a pre-trained model (Section 4.3).

[1] Puny et al. Frame averaging for invariant and equivariant network design (2021)

[2] Kaba et al. Equivariance with learned canonicalization functions (2023)

[3] Dym et al. Equivariant frames and the impossibility of continuous canonicalization (2024)

[4] Ma et al. A canonicalization perspective on invariant and equivariant learning (2024)

**Strengths:**

- S1. The contributions of this paper can be understood from two perspectives. The first is extending frame/canonicalization approaches for neural network symmetrization to non-compact Lie groups specified by their infinitesimal generators. The second is improving the performance of pre-trained neural operators for PDEs by canonicalizing them in accordance to the point symmetry of a downstream PDE. Both are original and significant contributions as far as I am aware.
- S2. The extension of frames and canonicalization considered in prior work [3, 4] to non-compact Lie groups, and the proposed Lie algebra descent algorithms that implement the idea, are original and technically sound as far as I am aware.
- S3. Experimental results are shown for a comprehensive set of problems (synthetic, two computer vision problems, and three PDEs) with informative visualizations and support the validity of the approach.

[3] Dym et al. Equivariant frames and the impossibility of continuous canonicalization (2024)

[4] Ma et al. A canonicalization perspective on invariant and equivariant learning (2024)

**Weaknesses:**

- W1. In Section 2.2, the authors propose to treat non-weighted energy-minimizing closed canonicalization as weighted canonicalization by taking the normalized Hausdorff measure on energy minimizing set (Line 287-296). The resulting class of weighted closed canonicalization (Theorem 2.7) has a weakness that, with the energy function specified, it is not possible to adjust the weights of canonicalization from training data, unlike in related prior work [5-7]. This may have led to the reliance on carefully designed energy functions based on domain knowledge (Line 324-332 and Section 4.3) which leaves a room for improvement.
- W2. For the ACE experiment (Section 4.3.3), the current comparison is made only between Poseidon and its canonicalization. A comparison to existing intrinsically symmetric methods [8, 9] would be informative and show the usefulness of canonicalization, since Poseidon can benefit from pre-training while intrinsically symmetric approaches cannot.

Minor comments and suggestions

- In Line 249, the notation for closure $\bar{X}$ of a set $X$ is used without definition.
- In Line 260, $\mathrm{PMeas}$ -> $\mathrm{PMeas}(X)$
- In Line 976, euclidean -> Euclidean
- In Line 1758, the union of -> a union of?
- On the orbit distance constraint (Line 357), the authors may find [10] relevant, as the approach uses invariant polynomials for linearly reductive groups (Lines 2227-2228) to measure orbit distance.

[5] Mondal et al. Equivariant adaptation of large pretrained models (2023)

[6] Kim et al. Learning probabilistic symmetrization for architecture agnostic equivariance (2023)

[7] Zhang et al. SymDiff: Equivariant diffusion via stochastic symmetrisation (2024)

[8] Arora et al. Invariant physics-informed neural networks for ordinary differential equations (2024)

[9] Lagrave & Tron, Equivariant neural networks and differential invariants theory for solving partial differential equations (2022)

[10] Nguyen et al. Learning symmetrization for equivariance with orbit distance minimization (2023)

**Questions:**

Please see the weaknesses section.

---

> ### Author Response · Authors · 2024-11-19
> **[1/1]**
>
> We thank the reviewer for their thoughtful and detailed comments. We are glad that the reviewer found the contributions of our paper original and significant, and the experimental results comprehensive and informative. We would also like to thank the reviewer for the references provided, which have now been included in the paper. We address the weaknesses and minor comments below:
>
> * W1: We completely agree with the reviewer - in fact better constructions of energy functionals for such a task will likely be the main direction of improvement. Effectively we would wish to align minimas of such an energy with the training set - doing this directly is not that simple and should be a fruitful direction for future work.
> * W2: We agree that it could potentially be an interesting comparison - however ensuring fair conditions for such a comparison are unclear in relation to sizes of networks, dataset sizes and a multitude of other factors. Particularly, this falls slightly outside the scope of what canonicalization is meant to achieve, as the main benefit is the ability to turn already pre-trained models equivariant without retraining (or at most minor finetuning). Overall, definitely interesting, but may fall too far outside the scope of this paper.
>
> Minor comments and suggestions:
> * Typos have been fixed.
> * “On the orbit distance constraint (Line 357), the authors may find [10] relevant, as the approach uses invariant polynomials for linearly reductive groups (Lines 2227-2228) to measure orbit distance.”
>   * This paper [10] is very nice, and their method is in our language of weighted canonicalization. They are able to leverage the fact that they look at reductive linear groups acting algebraically on vector spaces, which allows them to use many classical results on invariant polynomials. Arbitrary lie group actions of the kind that we consider do not have such a nice theory, so it would be unfeasible to write it so cleanly. It is definitely a valuable comparison to make with our method though and we have added that.

---

> ### Author Response · Authors · 2024-11-25
>
> As the discussion period draws to a close, we'd like to check in on whether you've had a chance to review our responses and have any follow up questions?
>
> We hope that our reply clarifies and alleviates the reviewer’s concerns. If this is the case, we kindly ask the reviewer to consider raising their rating, given that they are acknowledging the novelty, the strengths and the contributions of our paper.

---

> ### Comment · Reviewer_VdBH · 2024-11-27
>
> Thank you for the response and incorporating the suggestions on typos and reference. I have read other reviews and responses and would like to retain my supportive rating.
>
> On W2, I believe it is reasonable to ask whether a new canonicalization method offers performance gains over existing equivariant networks (if available; like in Table 1) because, in the end of the day, the goal of canonicalization is equivariant learning, and equivariant networks are established methods that solve the same problem. But I agree that this would require substantial effort for the tasks in Section 4.3, especially if the argument in Line 48-51 applies.

---

> > ### Author Response · Authors · 2024-11-27
> >
> > Thank you for your support and insightful comments. We definitely agree re comparison, but indeed by 48-51, and particularly the fact that it does not act in a representation makes the problem very complicated. Instead, as suggested by one of the reviewers we will include (and already have for the heat equation) for ACE experiments a comparison against data-augmentation.

---

### Official Review · Reviewer_n4j8 · 2024-11-04

**Soundness:** 3
**Presentation:** 1
**Contribution:** 2
**Rating:** 6
**Confidence:** 3

**Summary:**

The paper proposes a Lie algebra canonicalization mechanism for achieving equivariance with respect to a variety of Lie groups. The authors propose an extension of the energy based canonicalization mechanism, analyzing limitations of existing work within this framework and describing how the methodology can be extended to work with more exotic (i.e. non-compact, non-abelian) Lie groups, enabling application to a large class of learning tasks where equivariance might be desired. The paper connects frames, canonicalization and frame averaging and the authors show that non-compact groups can be approached via the energy minimization framework where an optimization process makes use of the Lie algebra generators. The methodology is evaluated both on standard invariant image classification tasks as well as physics-informed learning problems dealing with Lie point symmetries.

**Strengths:**

- The paper is mathematically well-grounded. While I think the presentation could be reorganized (see below) I appreciate the focus that the authors have on providing derivations and proofs for their claims. The objective of unifying (weighted) canonicalization mechanisms and frame theory could prove useful for practitioners in this area.
- I am happy to see extensions to the canonicalization framework, especially ones which focus on making use of the underlying geometry and parametrization of the groups and spaces involved, as well as the treatment of non-compact groups which are under-explored.
- I find the potential use-case of applying this methodology to pre-trained models still under-explored and worthy of further investigation.

**Weaknesses:**

I think the current presentation lacks focus and the paper could be vastly improved with the objectives of:
- Highlighting clearly the limitations of past/current proposals and how these limitations are overcome, potentially alternating more concrete examples with more abstract limitations.
- Presenting a formalized methodology for the entire (extended) framework that could be understood by practitioners with choices and pitfalls for the spaces, groups, energy functions, etc. involved.

In regards to improving clarity:
- $WFra_{G}(X)$ should be defined before it is used on line 236.
- On lines 270-271 is $N$ simply some value $\in \mathbb{R}$?
- It should be made clearer that $C_{E}(x)$ (line 288-289) refers to the normalized counting measure on $M_{E}(x)$, since we are stating that $C_{E}$ is a probability measure, and then on line 296 we write $C_{E} = \mu_{x}$.
- It would be useful to make clearer for each proposed construction the limitation that exists with the current methodology, e.g. it seems from Definition 2.2 and the subsequent paragraphs one should understand that restricting the support to the orbits and equiping $WCAN_{G}$ with the coarsest topology was not proposed before? Similarly, it would be useful to make clearer what topological limitations appear for non-compact groups, and for which in particular (e.g. we are not just talking about translation).
- In the same spirit clarifying which non-compact Lie groups acting on which spaces (transitively or not) have non-closed orbits would highlight more clearly the settings where the framework should be considered.
- And similarly to the previous comment the cases where the energy $E$ induces a 'reasonable' probability measure on $M_{E}(x)$ could be contrasted with specific/concrete choices of groups and group actions.
- Considering the main proposal of the paper is a general framework I find the discussion in Section 3 much too general and unstructured. It is again highlighted what limitations could exist when one chooses an energy function, however I think it would be much more useful to present a summary of the entire canonicalization framework (and the limitations that were addressed) with a clear outline of potential choices for input spaces and groups (and their actions), potentially in increasing generality (e.g. finite -> compact Lie group -> non-compact Lie group). Once the methodology is clear one highlights criteria for choosing the energy function. Some form of presentation similar to Algorithm 1 in the appendix could potentially appear in the main manuscript given that the energy function is a key component of the generalized framework.
- "What makes moving away from the compactness assumption even worse, is the move away from the matrix view, as any compact Lie group is a matrix group (Knapp, 1996, Chapter 4)" - I don't quite understand what is being claimed here and in the next few sentences. A matrix Lie group is a Lie subgroup of $\textnormal{GL}(n, \mathbb{R})$ (or $\mathbb{C}$). Non-compact matrix Lie groups can also be decomposed/expressed as a product of exponentials, and one can optimize using their Lie algebra elements, see e.g. [3] and [4].
- I'm left not understanding what the authors mean when they state that their framework requires  less knowledge about the symmetry group. This is highlighted both in the presentation of the contributions as well as in the experiments, e.g. for invariant classification a comparison is done with [1], I assume as opposed to [1] and [2]? I don't have an issue with what results are cited, but it is not made clear what knowledge about the Lie group is needed for [2] that isn't needed for [1]?

I think the authors should focus on improving/rewriting parts of Sections 2 and 3 with the focus of highlighting both the limitations their framework overcomes and providing a clearer presentation of how the methodology can be applied in different cases.

Some of the expression/grammar in the appendix could also be improved, e.g. "Liealgebra descent", "why failure modes exists".

[1] Enabling equivariance for arbitrary lie groups, MacDonald et. al 2022
[2] Lie group decompositions for equivariant neural networks, Mironenco & Forre 2024
[3] Trivializations for Gradient-Based Optimization on Manifolds, Mario Lezcano-Casado, 2019
[4] Optimization algorithms on matrix manifolds, Absil et. al 2009

**Questions:**

Besides the questions in the weaknesses section:
- Can you clarify what role does the Haar measure play in the construction of weighted canonicalizations for the non-compact case? Does unimodularity play a role here? Is the case where the modular function is unbounded pose any additional obstructions?
- Is it not possible to make use of a riemannian structure on the group and have the energy minimizer be defined in terms of geodesic distance? It seems the current proposal already looks to work within the tangent space of the groups involved.
- Canonicalization methods IIUC deal with global invariance/equivariance, as opposed to e.g. lifting + regular/steerable convolutions which could deal with local equivariance (e.g. imagine 2 objects in an image rotated at different angles). I'm wondering if the authors would find this distinction worthy of highlighting.

---

> ### Author Response · Authors · 2024-11-19
> **[1/2]**
>
> * Lines 236, 270, 288, 296:
>   * All of these have now been fixed and clarified.
> * It would be useful to make clearer for each proposed construction the limitation that exists with the current methodology:
>   * In order to address exactly what limitations exist in current work we have added a table to summarise precisely what methods have what limitations. The consideration of the weak topology of canonicalizations acting on continuous functions comes from analogy with other weak topologies present in functional analysis. As far as we can tell no one else has used this sort of weak topology of the action on continuous maps, despite being the most natural choice.
> * "Clarifying which non-compact Lie groups acting on which spaces have non-closed orbits" and "cases where the energy induces a 'reasonable' probability measure":
>   * We appreciate the comments about the lack of clarity on these group actions. We have added a mathematical example to the theory section that highlights all of the problems that can occur when moving to non-compact Lie groups, both with regards to orbits and with our energy optimization framework.
> * “discussion in Section 3 much too general and unstructured”:
>   * We completely agree with the reviewers comment and have revised our manuscript accordingly to improve clarity and focus (as explained at the start of the response. We have now also moved one of the algorithms to the main text in order to make the approach explicit.
> * “I don't quite understand what is being claimed here”:
>   * We clarify that the goal of this paragraph was to emphasise that being able to globally parameterise the group is hopeless in the general non-compact lie group case. The noetherian property does not necessarily hold for these. Now we completely agree with the reviewer that knowing global structure (or even local via algebra solvability) can lead to global parameterisations - but this does not hold in general, and we may not have computational access to such global charts. We agree this was not clear, and have now rewritten this paragraph.
> * “I'm left not understanding what the authors mean when they state that their framework requires less knowledge about the symmetry group.“
>   * Yes, that is right - the global structure is assumed for [2], while [1] utilises being able to calculate $\exp( v )$ for some lie algebra vector $v$. Particularly note that theory of [1] only considers $v$ infinitesimal and the exact global exponential is not even needed, however it *does* require the exponential map for any $v$. For lie algebra canonicalization, when utilizing coordinate descent (algorithm 3) one only needs action of some basis $v_i$, which is the only thing normally derived when considering PDE symmetries.

---

> ### Author Response · Authors · 2024-11-19
> **[2/2]**
>
> * “Can you clarify what role does the Haar measure play in the construction of weighted canonicalizations for the non-compact case? Does unimodularity play a role here? Is the case where the modular function is unbounded pose any additional obstructions?”
>   * The Haar measure plays no role in constructing weighted canonicalizations. It plays a role in proving weighted closed canonicalizations are the sequential closure of weighted orbit canonicalizations, but we only really need it to have an (nice) $\sigma$-finite measure such that we can decompose a weighted canonicalization into easier to deal with terms. Replacing the left Haar measure with the right Haar measure in this proof doesn’t affect the result, though the constructed approximation will be different.
> * “Is it not possible to make use of a riemannian structure on the group and have the energy minimizer be defined in terms of geodesic distance? It seems the current proposal already looks to work within the tangent space of the groups involved.”
>   * We agree that utilizing more information of the group can prove beneficial. It is unclear however how exactly the geodesic distance can be helpful in minimizing the energy. In order for the method to be an orbit canonicalization, the problem has to be of the form eqn (2) or similar (see Kaba et al. 2023). Could you clarify how exactly you think this may be used?
>   * In addition, when moving away from the compact case, being able to put a reasonable Riemannian structure on a Lie group is not guaranteed. Namely, we would want a bi-invariant metric on the Lie group (i.e. preserved by left and right multiplication) because this means that the Riemannian exponential map defined by geodesics and the Lie algebra exponential map defined by the group structure will agree. This is not always possible for non-compact groups. Putting a non bi-invariant metric on them is always possible but this will not give us the information about the group structure that we might care about. In specific cases, when knowing the group one could imagine tailoring the Riemannian metric to give nice geodesic distances that play well with energy minimization but this would need to be done case-by-case if it is even possible.
> * “Canonicalization methods IIUC deal with global invariance/equivariance, as opposed to e.g. lifting + regular/steerable convolutions which could deal with local equivariance (e.g. imagine 2 objects in an image rotated at different angles). I'm wondering if the authors would find this distinction worthy of highlighting.”
>   * We agree that it is very interesting to consider local equivariance in addition to the global equivariance considered in our work, and we have added references to work in which this local equivariance of steerable convolutions is discussed and a discussion of the distinction between local and global equivariance. In the context of steerable convolutions, it is worth noting that the local equivariance is not unconditional, but depends on the spatial extent of the filters being sufficiently small compared to the distance between features of interest. We believe that a proper treatment of such local transformations would naturally lead to studying “global” equivariance with respect to infinite-dimensional subgroups of the diffeomorphism group, which may act differently at every point in the domain. We consider this a promising direction for future work on LieLAC and have noted this in the revised paper.

---

> ### Author Response · Authors · 2024-11-25
>
> As the discussion period draws to a close, we'd like to check in on whether you've had a chance to review our responses and have any follow up questions?
>
> We hope that our reply clarifies and alleviates the reviewer’s concerns. If this is the case, we kindly ask the reviewer to consider raising their rating, given that they are acknowledging the novelty, the strengths and the contributions of our paper.

---

> ### Comment · Reviewer_n4j8 · 2024-11-26
>
> Thank you for the clarifications. I would be happy to update my score; I would encourage the authors to upload the revised version of the manuscript so as to be able to see that the requested changes have been addressed.
>
> > Now we completely agree with the reviewer that knowing global structure (or even local via algebra solvability) can lead to global parameterisations - but this does not hold in general, and we may not have computational access to such global charts.
>
> I agree. Potentially a more precise minimal category of groups for which this (and your framework) holds are matrix Lie groups which are reductive/semi-simple (in the sense defined by Knapp).
>
> > Yes, that is right - the global structure is assumed for [2], while [1] utilises being able to calculate $exp(v)$ for some lie algebra vector $v$. Particularly note that theory of [1] only considers infinitesimal and the exact global exponential is not even needed, however it does require the exponential map for any.
>
> I understand now, thank you for clarifying. My confusion likely lied with the question of whether [1] actually achieves global equivariance given that they work only with the injectivity radius of the exponential map - a consequence of losing global structure.
>
> > We agree that utilizing more information of the group can prove beneficial. It is unclear however how exactly the geodesic distance can be helpful in minimizing the energy. In order for the method to be an orbit canonicalization, the problem has to be of the form eqn (2) or similar (see Kaba et al. 2023). Could you clarify how exactly you think this may be used?
>
> To clarify, I agree this is beyond the scope of the paper. IIUC the jumping off point where one heads towards weighed canonicalizations is given by the need to have a minimum on every orbit. The requirement is similar to one encountered under a framework I find similar to canonicalization - the deformable template model/random orbit model, and the connection seems unexplored.  If the authors are curious they can review [5] (see e.g. chapters 4,9), in short the group is endowed with a Riemannian metric and the geodesic distance acts as an additional regularization loss, such that one not only finds the group element taking a sample back to the 'canonical sample' (template) but the transformation is one of 'lowest magnitude'. Your concerns relating to the existence of a metric which is only left/right invariant rather than by-invariant also appear in this framework.
>
> [5] - Riemannian geometric statistics in medical image analysis, Pennec et. al 2020.
>
> edit after new revision has been uploaded:
>
> > In order to address exactly what limitations exist in current work we have added a table to summarise precisely what methods have what limitations.
>
> Is this referring to Figure 3 of the current revised version?

---

> > ### Author Response · Authors · 2024-11-27
> >
> > We appreciate the reviewer's insightful feedback and engagement.  We believe that thanks to the comments, the manuscript has now been significantly improved. We would appreciate if the reviewer updated their score and we welcome any further feedback.
> >
> > > I agree. Potentially a more precise minimal category of groups for which this (and your framework) holds are matrix Lie groups which are reductive/semi-simple (in the sense defined by Knapp).
> >
> > Yes, you're right. However our framework does not require the groups to fall into this category. We don't require the group to have those properties because doing steps on for energy minimization only needs the gradients of the action.  We don't need a global parameterization of the group since we're not searching for the frame itself, but rather its canonicalization.
> >
> > > To clarify, I agree this is beyond the scope of the paper. IIUC the jumping off point where one heads towards weighed canonicalizations is given by the need to have a minimum on every orbit. The requirement is similar to one encountered under a framework I find similar to canonicalization - the deformable template model/random orbit model, and the connection seems unexplored. If the authors are curious they can review [5] (see e.g. chapters 4,9), in short the group is endowed with a Riemannian metric and the geodesic distance acts as an additional regularization loss, such that one not only finds the group element taking a sample back to the 'canonical sample' (template) but the transformation is one of 'lowest magnitude'. Your concerns relating to the existence of a metric which is only left/right invariant rather than by-invariant also appear in this framework.
> >
> > We definitely agree with this - the connection seems to be completely unexplored and most definitely warranted. The most fundamental difference would be that in the problems above the group is the group of diffeomorphisms, being infinite dimensional generally. We will definitely include a note highlighting this connection and the relevant literature in the paper.
> >
> > > In order to address exactly what limitations exist in current work we have added a table to summarise precisely what methods have what limitations. Is this referring to Figure 3 of the current revised version?
> >
> > Apologies, when writing the response we noted down the limitations in a table only to realise there was actually very little information being carried in it. We instead reverted to constructing the spaces from the start, instead motivating our design choices in defining the spaces through limitations of previous concepts where necessary. Figure 3 now visually summarizes the limitations of previous methods, motivating our design choices for the new spaces, instead of stating the theorems themselves.

---

> > > ### Comment · Reviewer_n4j8 · 2024-11-27
> > >
> > > Thank you for the response and clarifications. I think the revised paper is in a better state, and am now leaning towards acceptance; I've updated my score accordingly.

---

### Comment · Area_Chair_hAWg · 2024-11-25

Dear reviewers,

If you haven’t done so already, please engage in the discussion as soon as possible. Specifically, please acknowledge that you have thoroughly reviewed the authors' rebuttal and indicate whether your concerns have been adequately addressed. Your input during this critical phase is essential—not only for the authors but also for your fellow reviewers and the Area Chair—to ensure a fair evaluation.
Best wishes,
AC

---

### Meta-Review · Area_Chair_hAWg · 2024-12-21

**Metareview:**

This paper proposes a novel extension of canonicalization-like approaches to non-compact Lie groups, a significant advancement in building symmetric neural networks. Frame averaging and canonicalization are foundational techniques, and extending these mechanisms to a broader class of groups addresses a critical gap in the literature. The approach is well-grounded theoretically, with rigorous justification, and the proposed method is both novel and impactful. Experimental results convincingly demonstrate its effectiveness. While initial concerns were raised regarding the clarity of the presentation, the authors resolved most issues during the rebuttal phase with thoughtful revisions, improving the manuscript's accessibility.

**Additional Comments On Reviewer Discussion:**

All the reviewers acknowledged the significance of the contribution, and most of the weaknesses raised were related to the presentation. After the rebuttal period, most of those concerns about clarity were resolved. The reviewers remained positive until the end of the discussion period.

---

### Decision · Program_Chairs · 2025-01-22

Accept (Poster)